# Southern California margin benthic foraminiferal assemblages record recent centennial-scale changes in oxygen minimum zone

Hannah M. Palmer[1,2], Tessa M. Hill[1,2], Peter D. Roopnarine[3], Sarah E. Myhre[4], Katherine R.
Reyes[1,5], and Jonas T. Donnenfield[1,6]

[1]Bodega Marine Laboratory, University of California, Davis, California, USA
[2]Department of Earth and Planetary Sciences, University of California, Davis, California, USA
[3]Department of Invertebrate Zoology and Geology, California Academy of Sciences, San Francisco, California,
USA
[4]School of Oceanography, University of Washington, Seattle, Washington, USA
[5]Department of Natural Sciences and Mathematics, Dominican University of California, San Rafael, California,
USA
[6]Geology Department, Carleton College, Northfield, Minnesota, USA

*Correspondence to:* Hannah M. Palmer (hmpalmer@ucdavis.edu)

**Abstract.** Microfossil assemblages provide valuable records to investigate variability in continental margin
biogeochemical cycles, including dynamics of the oxygen minimum zone (OMZ). Analyses of modern assemblages
across environmental gradients are necessary to understand relationships between assemblage characteristics and
environmental factors. Five cores were analyzed from the San Diego margin (32°42'00"N, 117°30'00"W, 300-1175
m water depth) for core top benthic foraminiferal assemblages to understand relationships between community
assemblages and spatial hydrographic gradients and for down core benthic foraminiferal assemblages to identify
changes in the OMZ through time. Comparisons of benthic foraminiferal assemblages from two size fractions (63-
150 and >150 μm) exhibit similar trends across the spatial/environmental gradient, or in some cases exhibit more
pronounced spatial trends in the >150 μm fraction. A range of species diversity exists within the modern OMZ
(1.910-2.586 H, Shannon Index), suggesting that diversity is not driven by oxygenation alone. We identify two
hypoxic associated species (*B. spissa* and *U. peregrina*), one oxic associated species (*G. subglobosa*) and one OMZ
edge-associated species (*B. argentea*). Down core analysis of indicator species reveal variability in the upper margin
of the OMZ (528 m water depth) while the core of the OMZ (800 m) and below the OMZ (1175 m) remained stable
in the last 1.5 ka. We document expansion of the upper margin of the OMZ beginning 400 ybp on the San Diego
margin that is synchronous with other regional records of oxygenation.

## 1 Introduction

Ocean oxygenation is declining globally; rising ocean temperatures decrease oxygen solubility at the sea surface and increased stratification inhibits ventilation, leading to decreased oxygen at depth (Breitburg et al., 2018; Levin et al., 2009; Stramma et al., 2010). Expansions of oxygen minimum zones (OMZ) have already been documented and further expansions are predicted (Bograd et al., 2008; Schmidtko et al., 2017; Stramma et al., 2010). Within the California Current system, a decline in dissolved oxygen (DO) concentration, shoaling of the hypoxic boundary, and decreased pH have been documented (Bakun, 2017; Bograd et al., 2008). The intensity and geographic extent of the CA margin OMZ has oscillated in response to past changes in climate and ocean temperatures on millennial timescales – weakening during cool periods and strengthening during warm periods (Cannariato and Kennett, 1999; Jaccard et al., 2014; Moffitt et al., 2014, 2015a; Ohkushi et al., 2013). Determination of timing and drivers of past expansions and contractions of OMZs is critical to developing accurate predictions of future change (Jaccard et al., 2014).

Continental margin biogeochemical dynamics structure shelf ecosystems across space and time (Levin et al., 2009; Levin and Dayton, 2009). In particular, oxygenation is a key determinant of benthic zonation; seafloor ecosystems are subject to major turnover in response to relatively minor inferred changes in oxygenation (Levin, 2003; Levin and Dayton, 2009; Moffitt et al., 2015b). Areas of low oxygen availability typically contain low abundance and diversity of organisms (Levin, 2003; Levin and Dayton, 2009). However, several species of benthic foraminifera are adapted to survive in low-oxygen conditions and are thus present, and often abundant, in such environments (Bernhard and Gupta, 1999; Gooday et al., 2000; Kaiho, 1994, 1999; Keating-Bitonti and Payne, 2016).

### 1.1 Benthic foraminifera record changes in coastal margin biogeochemistry

Microfossil records from the Southern California Borderlands are a critical tool for understanding changes in productivity (Cannariato and Kennett, 1999; Emmer and Thunell, 2000; Stott et al., 2000), orbital and millennial scale climate changes (Hendy, 2010; Hendy and Kennett, 2000; Taylor et al., 2015), and climate change through the Holocene (Balmaki et al., 2019; Fisler and Hendy, 2008; Friddell et al., 2003; Roark et al., 2003). Benthic foraminiferal assemblages are widely used as a proxy for changes in oxygenation through time (Balestra et al., 2018; Bernhard et al., 1997; Bernhard and Gupta, 1999; Cannariato and Kennett, 1999; Gooday, 2003; Jorissen et al., 2007; Moffitt et al., 2014; Ohkushi et al., 2013; Shibahara et al., 2007; Tetard et al., 2017). Previous work (through analysis of benthic foraminifera along environmental depth gradients and in laboratory culturing studies) documented relationships between benthic foraminiferal taxa and water depth, oxygen concentration, sediment substrate, position in the sediment matrix, nitrate availability, and organic matter availability (Bernhard et al., 1997; Bernhard and Bowser, 1999; Bernhard and Gupta, 1999; Cardich et al., 2019; Caulle et al., 2014; Douglas, 1981; Douglas and Heitman, 1979; Erdem et al., 2019; Jorissen et al., 2007; Kaiho, 1994, 1999; Mallon et al., 2012; Mazumder and Nigam, 2014; Mullins et al., 1985).

Generally, low oxygen environments contain high abundance and low diversity of benthic foraminifera; in these settings, infaunal, elongate, thin-walled species with high porosity dominate over porcelaneous and epifaunal taxa (Bernhard et al., 1997; Douglas, 1981; Jorissen et al., 1995, 2007; Kaiho, 1994, 1999; Mazumder and Nigam, 2014).

Further work has explored the relationship between foraminiferal size and oxygenation; generally volume to surface area ratios of foraminiferal tests are positively correlated with dissolved oxygen, yet studies of individual taxa on the Southern California margin do not consistently show this relationship (Keating-Bitonti and Payne, 2016; Keating-Bitonti and Payne, 2017; Rathburn et al., 2018). Often, individual taxa of foraminifera are classified into groups based on oxygen affinity or individually identified as oxygen indicator taxa (Jorissen et al., 1995; Kaiho, 1999; Moffitt et al., 2014). In particular, bolivinid taxa are noted as low oxygen indicator taxa (Cardich et al., 2015; Caulle

et al., 2014; Mallon et al., 2012; Mullins et al., 1985). However, these relationships between foraminiferal assemblages and environmental metrics are regionally defined and cannot be applied globally; regional calibrations of the benthic foraminifera oxygen proxy are required for accurate paleoceanographic analyses (Bernhard et al., 1997; Caulle et al., 2014; Kaiho, 1999; Mallon et al., 2012; Mazumder and Nigam, 2014). Similarly, the classification of oxygenation levels varies among paleoceanographic studies (Balestra et al., 2018; Kaiho, 1994;

Moffitt et al., 2015a). This study uses the following classification: oxic ($[O_2]$ >1.5 ml/L), intermediate hypoxia/suboxic ($[O_2]$ 1.5-0.5 ml/L), and severe hypoxia/dysoxic ($[O_2]$ <0.5 ml/L) (Moffitt et al., 2015a). Although oxygenation is a dominant driver of ecosystem zonation in marginal environments, sediment substrate, organic matter availability, and nitrate availability also play important roles in structuring benthic foraminiferal assemblages. Further analysis of the interacting environmental factors along depth/environmental gradients is needed (Jorissen et

al., 1995, 2007; Mullins et al., 1985; Venturelli et al., 2018).

Previous studies of oxygenation change over time from the Southern California margin focus largely on the Santa Barbara and Santa Monica Basins, due to their high sedimentation rates and regular laminations (Balestra et al., 2018; Cannariato and Kennett, 1999; Christensen et al., 1994; Kaiho, 1999; Moffitt et al., 2014; Schimmelmann et

al., 2013). Significantly fewer studies in the Northeast Pacific investigate sediments outside of those basin environments (McGann, 2002). Further analysis is therefore needed to constrain relationships between benthic foraminifera and environmental conditions in the open continental margin where biological and chemical gradients are more variable and to identify decadal to centennial changes in OMZ dynamics.

**1.2 Regional Setting**

The California margin is a well-studied system characterized by southward flow of the California Current, a strong seasonal upwelling regime bringing cold, nutrient rich waters to the surface, high coastal productivity, and a large OMZ occurring at intermediate water depths (Checkley and Barth, 2009). The San Diego margin is dominated by two surface currents: the southward flowing California Current and the seasonal, northward flowing surface

Davidson Current (Checkley and Barth, 2009).

An important feature of California margin coastal oceanography is the presence of a large, intermediate depth OMZ and carbon maximum zone (CMZ), from approximately 500 to 1000 m water depth (Helly and Levin, 2004; Stramma et al., 2010). The intensity and spatial extent of the modern California margin OMZ is driven by physical

mixing of well-oxygenated surface water, biological activity at the surface and at depth, and intrusion of lower oxygen bottom waters (Gilly et al., 2013). Both physical processes (temperature-dependent diffusion from atmosphere, mixing, stratification, deep water circulation) and biological processes (primary productivity at the surface and respiration at depth) control the intensity and extent of OMZs (Gilly et al., 2013; Helly and Levin, 2004).


Here we utilize the spatial variability of the continental margin as a natural laboratory to analyze relationships between core top, modern benthic foraminiferal assemblages and *in situ* measurements of environmental parameters (temperature, oxygenation, carbonate chemistry). We then quantify benthic foraminiferal assemblages down core to understand past environmental change.


## 2 Methods

### 2.1 Study site
The San Diego margin is located in the Southern California Borderlands in the Eastern North Pacific (Fig. 1). The margin slopes downward to a depth of approximately 1200 m at a distance of 30 km normal to shore (Fig. 1) and is

bounded by a bathymetric rise (600 m water depth) to the west. All hydrographic data and sediment cores used in this study were collected by the San Diego Coastal Expedition aboard the R/V *Melville* in December 2012 by a team from the Scripps Institution of Oceanography.

### 2.2 Oceanographic data collection
Bottom water temperature, salinity, and dissolved oxygen concentration were collected at each coring location using a Seabird CTD (SBE9), with a dissolved oxygen probe (Seabird Electronics Sensor SBE43). Carbonate chemistry was completed using *in situ* bottle sampling for pH and total dissolved inorganic carbon (DIC) and were previously published (Nam et al., 2015; Takeshita et al., 2015). Bottom depths were measured acoustically at each site.

### 2.3 Sediment cores
Five sediment cores collected along a depth transect from 300 to 1175 m depth were selected for use in this study (Figure 1; Table 1). Short (15-31cm) surface sediment cores were collected along the coastal margin using a deep-sea multicore with 9 cm diameter cores. Each core was divided into 1 cm intervals aboard the ship and immediately frozen. Sediments were not stained upon retrieval; thus, we could not carry out an analysis of live vs. dead or depth

habitat of these species. Total organic matter of core top sediments was measured as percent ash-free dry weight and previously published in Grupe et al. 2015 (Grupe et al., 2015). Subsamples were each disaggregated in deionized water, washed over a 63 μm sieve and oven dried at 50° C.

**2.3 Foraminiferal assemblages**

Assemblages were counted from the > 150 µm and 63 - 150 µm fraction for comparison between the size fractions. Sediments were split using a sediment splitter and dry sieved; a minimum of 300 foraminifera per sediment sample (in the >150 µm fraction) were identified and counted for all core top samples to provide a representative assemblage, unless fewer than 300 specimens were present in the entire sample. Identification of benthic foraminiferal taxa was based on previously published descriptions and images of benthic foraminifera (see Supporting Information, Figure S4 for images of dominant taxa and taxonomic reference list) (Balestra et al., 2018; Erdem and Schönfeld, 2017; Keating-Bitonti and Payne, 2017; Moffitt et al., 2014; Setoyama and Kaminski, 2015). Assemblages in the 63-150 µm fraction were quantified from the same sediment fraction as the > 150 µm to allow for comparison of abundance between the two groups. Down core assemblages were quantified in the > 150 µm size fraction (see discussion below) and a minimum of 150 foraminifera per sediment sample were counted in all down core samples unless fewer than 150 specimens were present in the sample (Mallon et al., 2012). Assemblage counts are standardized to the volume (63.62 cm$^3$) of the sampled cylinder of the sediment (core). Core top sediment (0-1, 1-2 cm) calcareous benthic foraminiferal assemblages were described for all cores. Complete down-core records of benthic foraminiferal assemblages were examined at 1 or 2 cm intervals for cores MV1217-2-3 (0-18 cm, sampled at 1 cm interval, 528 m water depth), MV1217-1-3 (0-10 cm, sampled at 1 cm interval; 10-20 cm, sampled at 2 cm interval, 800 m water depth). In order to build a longer temporal record at 1175 m we combine two multi cores; core MV1217-4-3 (0-10 cm, sampled at 1 cm interval) and MV1217-4-1 (sampled at 10-16 cm, 2 cm interval, 1175 m water depth).

Shannon Index of diversity was used to describe foraminiferal diversity (Peet, 1974). Analysis of variance (ANOVA) was used to analyze to determine differences between assemblages in two size fractions > 150 µm and 63 – 150 µm. Distance-based redundancy analysis (dbRDA) with Bray–Curtis distances was used to ordinate core top assemblages and examine relationship to environmental variables. Non-metric multidimensional scaling ordination, using square root transformation of assemblage species counts and Bray-Curtis similarities, were completed to identify relatedness between assemblages through time. All multivariate analyses were completed using the Vegan R package (Myhre et al., 2017; Oksanen et al., 2013).

**2.4 Radiocarbon based chronology**

Cores from two sites (528 m water depth, MV1217-2-3, 1175 m water depth, MV1217-4-1 and MV1217-4-3) were selected for radiocarbon ($^{14}$C) dating using mixed planktonic foraminifera. Core MV1217-2-3 (528 m) was sampled at three 1-cm intervals (11-12 cm, 16-17 cm, 25-26 cm). To generate an age model for the multicores at 1175m water depth, core MV1217-4-3 (1175 m) was sampled at one 1-cm interval (5-6 cm) and core MV1217-4-1 (1175 m) was sampled at two 1-cm intervals (10-11 cm, 20-21 cm) (Table 2). Radiocarbon analysis was completed at the Lawrence Livermore National Laboratory using δ$^{13}$C assumed values following the convention of Stuiver and Polach 1977 (Stuiver and Polach, 1977). The quoted age was given in radiocarbon years using the Libby half-life of

5568 years. The Calib7.1 calibration program was used to calibrate ages using a reservoir age of 220.0 +/- 40.0 (Ingram and Southon, 1996; Stuiver and Polach, 1977).

### 3.0 Results

### 3.1 Vertical profiles, sediment characterization of San Diego margin

Across the depth profile, bottom water temperature decreased steadily with depth (300 m to 1175 m) ranging from 8.6°C (300 m) to 3.8°C (1175 m) (Figure 2, Table 1). Salinity (not plotted here) had a mean of 34.4 psu ranging from 34.1 to 34.5 psu. Water column DO measurements collected directly above each coring site show oxic conditions at 300 m (1.54 ml/L) above the OMZ, intermediate hypoxia at 1175 m (0.58 ml/L) below the OMZ, and severe hypoxia at 528 m (0.35 ml/L) at the upper edge of the OMZ, and within the OMZ at 700 m (0.26 ml/L) and 800 m

(0.29 ml/L). Although not greatly variable, a pH minimum occurs at 700 m (7.55) and is higher at 300 m (7.65) and 1175 m, (7.59) (Figure 2, Table 1). Total organic matter increased with depth (6.8-14.7% AFDW, Table 1). These results are consistent with previous analyses of the California margin OMZ/CMZ (Helly and Levin, 2004).

### 3.2 Benthic foraminiferal assemblage across modern environmental gradient

Relative abundance of benthic foraminifera was quantified for all sites in the 0-1 and 1-2 cm intervals. We compared the 0-1 cm interval to the 1-2 cm interval to assess if depth habitat of any species determined their relative abundance in the core top assemblage. Specimens were not Rose Bengal stained, thus their presence in any interval does not indicate that they were living at the time of collection. We do not identify any significant relationship between relative abundance of a species and depth interval (ANOVA; in all cases $p > 0.05$ or $r^2$ is $<0.001$).

Foraminiferal abundance is low (<100 individuals) in some of the samples from 0-1 cm. Thus, in order to utilize sufficient numbers of individuals and because there were no significant differences in abundance of species between 0-1 cm and 1-2 cm, for the rest of the discussion we refer to the 0-2 cm fraction as the core top material (Figure 2 and 3). Calcareous taxa dominated the assemblage at every site; agglutinated foraminifera made up 0 (300 m) to 21% (700 m) of the assemblage. Due to their propensity for degradation and to remain consistent with other regional

studies, we exclude agglutinated taxa and all values are reported as percent of total calcareous taxa for the remainder of the text (Balestra et al., 2018; Kaiho, 1994; Moffitt et al., 2014; Venturelli et al., 2018).

Total abundance of foraminifera decreases with depth (Figure 2). Core top assemblages were dominated by *Bolivina argentea, Epistominella* sp.*, Uvigerina peregrina, Globocassidulina subglobosa, Cassidulina carinata,* and *Bolivina*

*spissa,* in order of decreasing abundance (see Figure S4 for images of dominant taxa). These dominant taxa make up 80% of all calcareous foraminifera counted across all core top samples. All other species each represent less than four percent of the total assemblage across all core tops. The following taxa are found at all five water depths: *Bolivina argentea, Bolivina spissa, Bulimina* spp., *Cibicidoides* sp.*, Epistominella* sp.*, Globobulimina pacifica, Globocassidulina subglobosa, Globobulimina ovata, Nonionella stella, Quinqueloculina* sp. and *Uvigerina*

*peregrina.*

First, we report the benthic foraminiferal assemblage from the >63 μm size fraction; we then report on a comparison between the 63 - 150 μm and >150 μm size fractions. The assemblage at 300 m is dominated by *G. subglobosa* (28%), *B. argentea* (25%), *U. peregrina* (10%), *Epistominella* sp. (8%), and *Bolivina spissa* (6%); species richness is 24 and diversity (H) is 2.133. The assemblage at 528 m is dominated by *B. argentea* (37%), *U. peregrina* (23%), *Epistominella* sp. (15%), *C. carinata* (6%) and *G. subglobosa* (5%); species richness is 25 and diversity (H) is 1.910. The assemblage at 700 m is dominated by *Epistominella* sp. (29%), *C. carinata* (15%), *U. peregrina* (13%), *G. subglobosa* (11%), *B. argentea* (10%), and *B. spissa* (7%); species richness is 23 and diversity (H) is 2.249. The assemblage at 800 m is dominated by *B. spissa* (16%), *U. peregrina* (16%), *Epistominella* sp. (13%), *C. carinata* (10%), *Cibicidoides* sp. (10%), and *Globobulimina ovata* (13%); species richness is 25 and diversity (H) is 2.586. The assemblage at 1175 m is dominated by *C. carinata* (25%), *Epistominella* sp. (20%), *G. subglobosa* (8%), *B. spissa* (6%); species richness is 25 and diversity (H) is 2.389 (Figure 2).

### 3.3 Comparison of benthic foraminifera in two size fractions

Comparison of foraminiferal abundance between the 63-150 μm and >150 μm shows higher abundance in the small fraction at 300 m, 700 m and 1175 m, and higher abundance in the large size fraction at 528 m and 800 m. Several taxa are found in both size fractions at all five water depths: *Globocassidulina subglobosa* and *Epistominella* sp. Four species have significantly different relative abundances between size classes; three are more likely to be found in the 63-150 μm (*C. carinata, Epistominella* sp. and *G. subglobosa*) and one species (*U. peregrina*) is significantly more likely to be found in the 150 μm size fraction (ANOVA, $p < 0.05$ for all, Figure 3).

In the >150 μm size fraction, species diversity (H) ranges from 1.316–2.700; minimum diversity (H) is found at 528 m (1.316) and maximum diversity (H) is found at 1175 m (2.700). In comparison, in the 63-150 μm size fraction, species diversity (H) ranges from 1.710–2.042; minimum diversity (H) is found at 700 m (1.710) and maximum diversity (H) is found at 800 m (2.042) (Figure 2). Species diversity is greater in the >150 μm size fraction relative to the 63-150 μm size fraction at all sites except the site at 528 m (Figure 2).

When we consider the complete assemblage (>63μm) we can classify the most abundant species into four groups based on their trends relative to the OMZ (Figure 3). Two species are more abundant within the OMZ: *B. spissa,* and *U. peregrina*; we identify these species as dysoxic indicator species. One species is less abundant within the OMZ relative to sites outside of the range of the OMZ: *G. subglobosa*; we identify this species as an oxic indicator species (Kaiho, 1999). Two species increase in abundance with water depth: *C. carinata* and *Epistominella* sp. One species is most abundant near the uppermost edge of the OMZ: *B. argentea*; this species may be edge-associated (Mullins et al., 1985). Importantly, when we consider only the >150 μm size fraction, we observe the same trends: high abundance in OMZ (*B. spissa, U. peregrina*), low abundance in OMZ (*G. subglobosa*), increased abundance with depth (*C. carinata* and *Epistominella* sp.), and OMZ edge-associated (*B. argentea*) (Figure 3). Generally, we find that trends across depth are similar between the complete (>63 μm) and large size fraction (>150 μm) or are more pronounced in the >150 μm size fraction compared to the 63-150 μm size fraction (Figure 3). In some taxa, trends in

both size fractions are similar across depth (*B. spissa, Epistominella* sp., *G. subglobosa, C. carinata*) (Figure 3). For other taxa, we observe a low relative abundance of a species in the (63-150 μm) fraction throughout all water depths, while for the same species in the >150 μm size fraction, we observe a pronounced trend through depth (*U. peregrina*, *B. argentea*) (Figure 3).

To further analyze these trends, we completed pairwise analysis of relative abundances of benthic foraminifera and environmental parameters. DO concentrations and pH are correlated at all water depths; here we chose to compare foraminiferal abundances to dissolved oxygen, yet we acknowledge that these affiliations may be driven by the combined effect of the OMZ/CMZ. When we analyze the complete assemblage (>63 μm) we identify a significant positive correlation between *G. subglobosa* and dissolved oxygen ($r^2$=0.76, $p<0.05$) and temperature ($r^2$=0.64, $p<0.05$) and a significant negative correlation between *G. subglobosa* and total organic matter ($r^2$=-0.72, $p<0.05$). If we analyze the >150 μm size fraction only, we identify the same significant positive correlation between DO and *G. subglobosa* ($r^2$=0.96, $p<0.05$) and also identify a positive correlation between *C. carinata* and water depth; abundance of this species increases with depth ($r^2$=0.93, $p<0.05$). When we analyze the 63-150 μm size fraction alone, we identify the same trends as observed in the >150 μm fraction: a significant positive correlation between *G. subglobosa* and dissolved oxygen ($r^2$=0.90, $p<0.05$). and a significant positive correlation between *C. carinata* and water depth ($r^2$=0.88, $p<0.05$). In the 63-150 μm fraction we also identify a significant negative correlation between *U. peregrina* and water depth ($r^2$=-0.95, $p<0.05$). We do not identify significant correlations between any other taxa and environmental parameter.

### 3.4 Multivariate analyses of benthic foraminiferal assemblage

Multivariate statistical analysis (using distance-based redundancy analysis) of core top assemblages identifies several taxa that contribute most strongly to the ordination of the assemblages (*G. subglobosa, U. peregrina, B. argentea*) (Figure S1). Oxygenation operates on an axis - separating sites at 300 m and 1175 m from the three OMZ sites 528 m, 700 m, and 800 m. Temperature operates on a second axis (Figure S1). Our findings support previous work that identify *G. subglobosa* with higher oxygen environments and *B. argentea* and *U. peregrina* with lower oxygen environments (Bernhard et al., 1997; Kaiho, 1994, 1999; Moffitt et al., 2014).

### 3.5 Age model development

Radiocarbon dating of two cores yielded variable sedimentation rates, from 6.1 to 42.2 cm/ka (error of +/- 11.3, see Table 2). An age model was developed for each core based on linear interpolation between radiocarbon dates (Figure S2). Core MV1217-3-3 (800 m water depth) was not radiocarbon dated; for this core we apply an average sedimentation rate (19.6 cm/ka) generated from the core above (528 m) and below (1175 m) this core. All following results will be discussed in age (years before present).

### 3.6 Temporal change in benthic foraminiferal assemblages

Down core assemblages were quantified in three cores (from 528, 800, and 1175 m water depth) in the >150 µm size fraction only, following results from core top analysis (see section 4.1). Down core assemblages contained similar species to core tops. Down core assemblages were dominated by (in descending order) *Uvigerina peregrina, Bolivina spissa, Bolivina argentea, Globobulimina* sp*., Cibicidoides* sp*.,* and *Epistominella* sp.. These dominant taxa make up more than 75% of all foraminifera counted across all cores and subsamples. All other species each account

for less than 5% of total assemblage across all cores and depths. The total number of species in each sample ranged from 11 to 26, comparable to the number of species found in the core tops.

Multivariate statistical analysis (using non-metric multidimensional scaling) of down core assemblages and core top assemblages shows that, through time, assemblage similarity within sites exceeds similarity to assemblages at any other site (Figure S3). In multivariate space, the difference between sites across space is greater than within any one

site through time. For this reason, we subsequently discuss change in assemblage through time at each site independently.

At 528 m water depth, foraminiferal assemblages vary through time, with a notable shift occurring at 400 ybp. Diversity decreases from 400 ybp to present, which is concurrent with a decrease in oxic indicator taxa *G. subglobosa*, an increase in dysoxic indicator *U. peregrina*, and a major increase in the proportion of *B. argentea*

(Figure 4).

At 800 m water depth, we do not document a significant shift in relative abundance of oxic indicators or dysoxic indicators over time, or a significant change in diversity over the interval examined (Figure 4). We interpret these assemblages to reflect environmental stability over the past 1.5ka (Figure 4). At 1175 m water depth, we document

little change in relative abundance of oxic and dysoxic indicator species from 200-800 ybp. Beginning at 200 ybp, we document an increase in *B. spissa*, but no change in *U. peregrina* (Figure 4)*.*

**4 Discussion**

**4.1 Benthic foraminiferal assemblages across modern environmental gradient**

Analysis of benthic foraminifera from two size fractions (63-150 µm and >150 µm) across a modern environmental gradient improves our understanding of benthic foraminifera as a proxy for past change. Total number of foraminifera in each size class varies with depth (Figure 2). A range of species diversity exists within the OMZ, suggesting that diversity is not driven by oxygenation alone (Figure 2).

In most cases, trends of relative abundance of benthic foraminifera across space are either similar between the complete assemblage (>63µm) and the >150 µm size fraction or trends in the >150 µm size fraction are more pronounced than in the complete assemblage (Figure 3). Trends across the OMZ gradient are similar in both size fractions (63-150 µm and >150 µm) in *G. subglobosa, Epistominella* sp., and *C. carinata.* If one were to interpret the combined assemblage or the >150 µm assemblage in these species, the results would be similar, despite these

species being higher in abundance in the small size fraction. Two species (*B. argentea* and *U. peregrina*) are present

in the 63-150 µm size fraction in all sites in similar (low) relative abundance, while in the >150 µm fraction, we document distinct trends in relative abundance of these two species across space (Figure 3). This is noteworthy as it may indicate that these species are able to tolerate a range of environmental conditions, and thus are present in small numbers and small shell sizes at all sites, but that in certain environments, these species are able to thrive and out-

compete other species, thus allowing them to grow to larger sizes (De Villiers, 2004; Gooday, 2003; Levin et al., 2010).  This is supported by previous work showing that environmental conditions do not play a role in determining volume of benthic foraminiferal proloculus (skeletal remains of initial cell), while volume and volume to surface area ratio of adult benthic foraminifera are controlled by dissolved oxygen within low oxygen environments (Keating-Bitonti and Payne, 2018).


In the Southern California Borderlands, the disciplinary convention has largely been to focus on the >150 µm size fraction, therefore quantifying this fraction is necessary for comparison to previously published studies (Balestra et al., 2018; Cannariato and Kennett, 1999; Moffitt et al., 2014). Our findings show that spatial trends in the >150 µm size fraction generally reflect those found in the >63 µm size fraction or are muted by the inclusion of the 63-150

µm (Figure 3). Results from a similar study in the Arabian Sea OMZ showed that assemblages were similar within the 63 – 125 µm fraction and >125 µm fraction and interpretation of the larger fraction was more useful to compare results to most paleoceanographic studies (Caulle et al., 2014). Thus, we recommend that workers utilize the >150 µm size fraction for analysis when targeting indicator taxa such as *B. argentea, B. spissa, U. peregrina, G. subglobosa* or when assessing trends across the OMZ gradient. However, it is useful to quantify the complete

>63µm assemblage in a subset of samples to ascertain whether there are important species or trends being missed. If the target of a project is to quantify changes in the ecology of a site or in specific metrics such as diversity, shell size, or shell weight, we recommend the inclusion of the complete assemblage (>63 µm). We acknowledge that the identification of microfossils as a tool for paleoceanographic interpretation contains inherent uncertainty due to variability in identification of species between observers and within single observers (Al-Sabouni et al., 2018; Fox et

al., 2018; Hsiang et al., 2019). Further, it has been shown that there is a correlation between size of specimen and accuracy of identification, meaning that the inclusion of the smaller specimens in the >63 µm fraction may reduce the accuracy of identification (Fox et al., 2018). Given this uncertainty, in subsequent text we focus only on spatial/environmental trends that change by a minimum of 10% relative abundance across the depth transect or through time.


In order to better compare to other similar studies from the Southern California Borderlands, for the remainder of the discussion we analyze the >150 µm fraction only. We identify two hypoxic-associated species (*B. spissa* and *U. peregrina*), one oxic-associated species (*G. subglobosa*) and one OMZ edge-associated species (*B. argentea*). These trends are shown in both the >150 µm assemblage and in the complete assemblage (>63 µm) (Figure 3). Not

surprisingly, these taxa are commonly used as indicator species in previous studies (Balestra et al., 2018; Cannariato and Kennett, 1999; Moffitt et al., 2014). The species we identify as dysoxic and edge-associated (*B. spissa, U. peregrina*, and *B. argentea*) are elongated in shape and are infaunal species. In comparison, *G. subglobosa* is more

abundant in higher oxygen environments and is a spherical, epifaunal species. This supports previous findings that infaunal vs. epifaunal habitat preference impacts the species distribution across the oxygenation gradient and that species in low oxygen zones have lower volume to surface area ratios relative to those in well oxygenated areas (Kaiho, 1999; Keating-Bitonti and Payne, 2016, 2018; Venturelli et al., 2018). In general, infaunal species are more common within the OMZ, while epifaunal are more common in well-oxygenated areas (Kaiho, 1999).

Previous studies have categorized benthic foraminifera into categories of oxygenation based upon similar work combining *in situ* environmental conditions and assemblage data (Cannariato and Kennett, 1999; Douglas and Heitman, 1979; Kaiho, 1994; Moffitt et al., 2014). Our findings indicate that region (or environment) specific oxygen species associations may be necessary, as our findings do not align directly with previous categorization of species. Several species that were previously recognized as low-oxygen indicators (*B. argentea, B. spissa,* and *U. peregrina*) were found at all water depths in this study, but we find only very low abundances of two well-documented low-oxygen indicator species, *Nonionella stella* and *Bolivina tumida* (Bernhard et al., 1997; Bernhard and Gupta, 1999; Cannariato and Kennett, 1999; Moffitt et al., 2014). These taxa have documented adaptations to extreme environments; *B. tumida* is associated with methane seep environments (Hill et al., 2003) and *N. stella* is known to sequester symbionts or plastids in extreme conditions (Bernhard and Bowser, 1999). We hypothesize that the marginal environment studied here does not reach the extreme hypoxic to anoxic conditions that are suitable for *B. tumida* or *N. stella.* Several species of documented oxic indicators (*Cibicidoides* sp. and *Quinqeloculina* sp. are found across all depths (300-1175 m) and oxygenation environments (0.26-1.54 ml/L) in the San Diego margin. Many past categorizations of these species were generated using species from very low oxygen basins (e.g., Santa Barbara Basin) where seasonal anoxia is present. The presence of oxic indicator species across all water depths on the San Diego margin may provide evidence for periodic flushing of high oxygen water or a selection for species that can tolerate a range of environmental conditions rather than a specific threshold of oxygenation. Alternatively, these species may be able to tolerate lower dissolved oxygen than previously thought if other environmental conditions (including substrate) are favorable or they may be able to tolerate short periods of low oxygen conditions (Burkett et al., 2016; Keating-Bitonti and Payne, 2018; Venturelli et al., 2018) (Figure 3). Further, habitat heterogeneity, including grain size (not measured here), may play a role in the determination of species assemblages at this site, particularly in low-oxygen areas in which the nature of the sediment matrix determines oxygenation of sediment porewater (Levin et al., 2010; Venturelli et al., 2018).

We document the presence of members of the *Bolivina* genera at all water depths and in some intervals described here, bolivinids make up more than fifty percent of the total assemblage. Therefore, any changes in abundance of this genus alone can drive changes in the assemblages as a whole. Within-species variation of morphologic traits have been correlated with affinities for certain environmental conditions (Lutze, 1964), yet congeneric gradations such as those observed here in the bolivinid genera merit further investigation. While bolivinids are widely accepted as low-oxygen dwelling species, *Bolivina argentea* specifically has been utilized as a low-oxygen indicator taxon

and analysis of their abundance and distribution requires careful scrutiny (Cardich et al., 2015; Caulle et al., 2014; Mallon et al., 2012).

We identify an anomalous assemblage at 528 m water depth; this assemblage is the least diverse (H=1.316, in >150 μm size fraction). Importantly, the low diversity at this site is driven by the dominance of a single species, *B. argentea*. We observe the dominance of *B. argentea* at 528 m water depth, near the modern upper margin of the OMZ and a lower relative abundance of this species at 700 and 800 m water depth, in the heart of the OMZ (Figure 3). This pattern of *B. argentea* at high abundances near the upper margin of the OMZ has been previously observed (Douglas, 1981; Mullins et al., 1985) and these species are often used as indicators of dysoxic environments (Bernhard et al., 1997; Kaiho, 1999). We attribute some of the unexpected variability in benthic communities that does not correlate with bottom water oxygenation to "edge effects" of the OMZ. Specifically, there is more biologically available nitrate and nitrifying bacteria at the edges of the OMZ as compared to the center, and thus we expect greater nutrient concentrations and larger food availability in these zones (Mullins et al., 1985). Multiple species of benthic foraminifera respire nitrate in low oxygen environments; in particular, *B. argentea* is particularly effective in utilizing nitrate as an alternative electron acceptor and this may contribute to its dominance at the upper edge of the OMZ (Bernhard et al., 2012; Glock et al., 2019; Kuhnt et al., 2013). Additionally, seasonal or annual variability in oxygenation of the upper margin of the OMZ causing a variable oxygenation regime at 528 m may drive selection for species that can tolerate a range of environmental conditions rather than a specific threshold of oxygenation. Further, environmental and ecological factors may combine to drive assemblage diversity; the interactive effects of competition and environmental adaptation may promote habitat specialization at this water depth (Fine et al., 2004). While the sites measured here document changes in the upper margin of the OMZ (528 m site), they may exclude the lower margin of the OMZ due to sampling depths of coring sites.

**4.2 Temporal change in benthic foraminiferal assemblages record environmental change through time**

Following results of analysis across modern core tops, all down core benthic foraminiferal assemblages were collected from the >150 μm size fraction. The combination of analysis from three distinct modern environments (upper margin of the OMZ (528 m), the center of the OMZ (800 m) and below the OMZ (1175 m)) allows for reconstruction of oceanographic change through the water column. We acknowledge that the environment is more variable than we can describe given the record available. Oxygenation varies on seasonal, annual, and decadal timescales - yet each interval of sediment analyzed represents >35 years, thus we are capturing a time-averaged signal. Further, several factors complicate our ability to interpret benthic assemblage records: relative preservation of various shell types, post-depositional changes in sediment, dominant fauna within some assemblages that dominate responses, and high frequency variability not captured in the record. Yet, we are still able to analyze benthic foraminiferal assemblages to identify environmental changes through the time interval described here.

Integrating analysis of cores from multiple depths reveals decadal to centennial variability in oxygenation at the upper margin of the OMZ (528 m) during the last 1.5 ka, but little to no change in the oxygenation at the center of

the OMZ (800 m) or below the OMZ (1175 m) (Figure 4). The change in assemblage at 528 m beginning at 400 ybp indicates a transition to the OMZ 'edge' environment in which *B. argentea* and *U. peregrina* species dominate in the modern. The formation of an assemblage that is similar to modern at 400 ybp implies the onset of modern conditions at this site which would include relatively low oxygen with variable oxygenation on seasonal to yearly timescales.

We interpret the transition in assemblage as a decrease in oxygenation at this depth and a shoaling of the upper margin of the OMZ beginning at 400 ybp and continuing to present. The combined suite of foraminiferal assemblages reveals shoaling of the upper margin of the OMZ in the last 400 years, while the center of the OMZ and below the OMZ remained stable (Figure 4). As such, we document an expansion of the upper margin of the OMZ beginning ~ 1600 CE and continuing to the present.


Our findings are consistent with observations from other regional records of oxygenation, including those from well-resolved records in nearby basins. Santa Monica Basin (SMB) and Santa Barbara Basin (SBB) are silled basins that experience periodic flushing; changes in the strength or oxygen content of North Pacific Intermediate Water, stratification or surface productivity can lead to changes in oxygenation within each basin (Balestra et al., 2018;

Cannariato and Kennett, 1999; Schimmelmann et al., 2013). Marine sediment records from Santa Monica Basin show non-annual laminations (indicating a hypoxic to anoxic environment) beginning 400 ybp and document shoaling of the low-oxygen zone within the basin from 400 ybp to present (Christensen et al., 1994). Santa Barbara Basin has well-documented sediment laminations through most of the Holocene indicating persistent low oxygen, but also shows gradual intensification of the OMZ within SBB since 1850 CE (Wang et al., 2017).


The synchronous decrease in oxygenation in Santa Monica Basin (Christensen et al., 1994) and San Diego margin (this study) from 400 ybp to present indicates that this deoxygenation is not driven by basinal changes alone; rather it is likely driven by regional scale phenomena. The decrease in oxygenation across the Southern California margin since 400 ybp could be attributed to 1) a change in oxygenation or strength of North Pacific Intermediate Water, 2)

an increase in organic carbon flux from sea surface to depth driven by changes in surface productivity, or 3) decrease in bottom water mixing or ventilation as a result of changes in surface water temperatures. We note that within the San Diego margin and Santa Monica Basin records, deoxygenation trends begin ~400 ybp but continue or intensify in the last 200 years. In the last 150 years, deoxygenation is synchronous across SBB, SMB and SDM. In this interval, decreases in oxygenation may be due to an increase in organic carbon supply from terrestrial sources

due to human land use change in the Southern CA region which has documented impacts on nearby benthic ecosystems (Tomasovych and Kidwell, 2017; Wang et al., 2017). Investigation of oxygenation change over time requires further research to identify forcing mechanisms for changes in the upper margin of the OMZ and to discern the relative impact of human and natural forcing in changing oxygenation across the last few centuries.

**5 Conclusion**
This spatial and temporal analysis of benthic foraminifera assemblages across a modern oxygen gradient on the San Diego margin improves our understanding of the relationship between assemblages and their environment.

Comparison of the relative abundance of benthic foraminifera in two size fractions (63-150 and >150 μm) across the modern OMZ shows that trends are either similar in both size fractions or are more pronounced in the larger size

fraction. As a result, we conclude that analysis of the >150 μm assemblage for this site provides the most useful record for interpreting benthic foraminifera as a proxy for past change.  We identify two hypoxic associated species (*B. spissa,* and *U. peregrina*), one oxic associated species (*G. subglobosa*) and one OMZ edge-associated species (*B. argentea*). Down core analysis of indicator species reveals variability in the upper margin of the OMZ while the center of the OMZ remained stable in the last 1.5ka. At 528 m, benthic foraminiferal assemblages indicate a

decrease in oxygenation at this depth and a shoaling of the upper margin of the OMZ beginning at 400 ybp and continuing to present. Expansion of the upper margin of OMZ beginning 400 ybp is synchronous with regional records of oxygenation.

**Data Availability**

All data collected for this paper is electronically archived by the NOAA Paleoclimatology Database as "San Diego Margin Benthic Foraminiferal Assemblages from Late Holocene" at www.ncdc.noaa.gov/paleo-search/study/28290 (Palmer et al., 2019).

**Author Contributions**

HMP, SEM, and TMH conceptualized and designed the project. HMP, SEM, JTD, and KRR completed data collection. HMP, TMH, and PDR completed data analysis. HMP wrote the manuscript. All authors contributed to editing of the manuscript.

**Acknowledgements**

We would like to acknowledge the San Diego Coastal Expedition team for their work in collecting the oceanographic data and sediment cores used in this study, particularly PIs Dr. Christina Frieder and Dr. Benjamin Grupe. We thank Dr. Seth Finnegan and Sara Kahanamoku for their support in microscopy and imaging. We acknowledge NSF support to T. Hill (OCE 1444451 and OCE 1832812), and Russell J. and Dorothy S. Bilinski Fellowship at Bodega Marine Laboratory, Geological Society of America Graduate Student Research Grant,

Cushman Foundation for Foraminiferal Research William V. Sliter Research Award, and UC Davis Earth and Planetary Sciences Durrell Grant for support to H. Palmer. We acknowledge two helpful reviews that improved this manuscript.

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

**Figures**

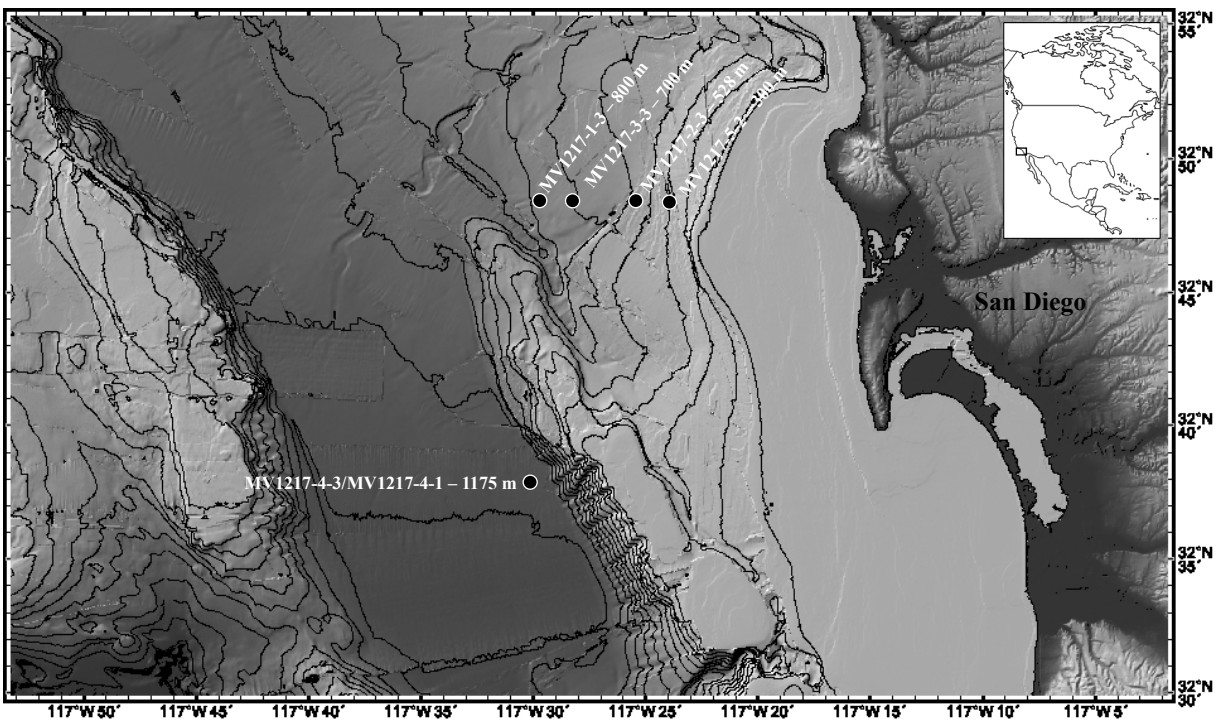


**Figure 1**. Map of cores used in this study. The cores were collected along a transect perpendicular to shore at the following water depths: 300, 528, 700, 800, 1175 m. Core top samples were analyzed for all cores. Cores MV1217-2-3, MV1217-1-3, and MV1217-4-3/MV1217-4-1 were analyzed down core.


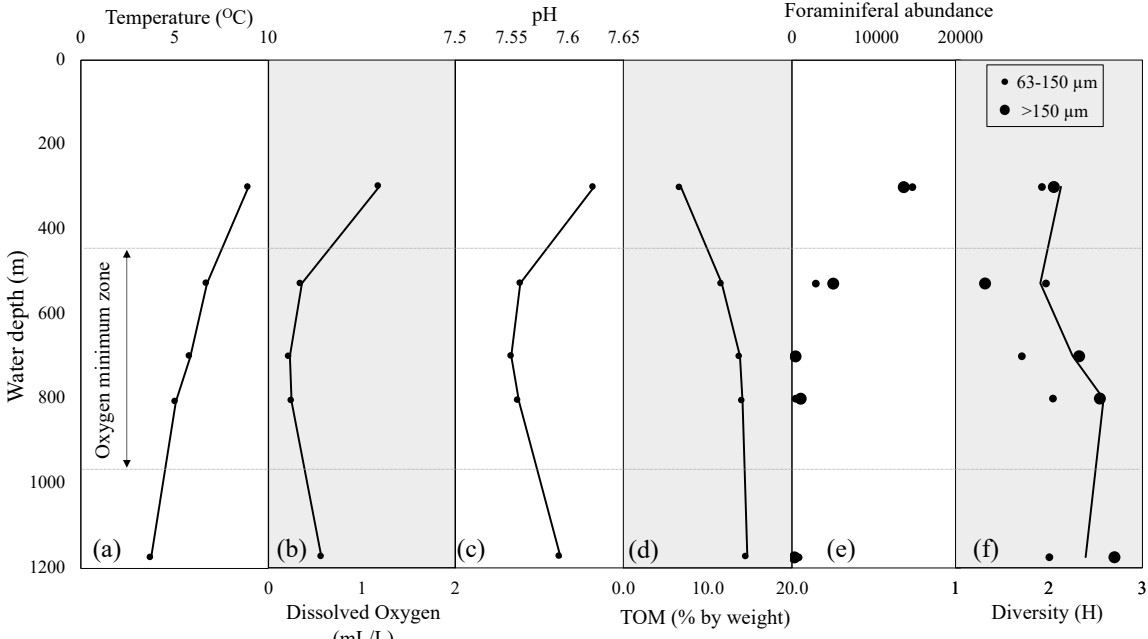

**Figure 2.** Profiles of temperature (a), dissolved oxygen (b), pH (c), and total organic matter (% by weight) (d) across depth transect. Foraminiferal abundance (total calcareous foraminifera) (e) and diversity (Shannon Index, H) (f) are shown for two size fractions. In panels (e) and (f), large black dots are >150 μm size fraction, small black dots are 63-150 μm and black line on diversity plot represents trends from the complete assemblage (>63 μm). Assemblage counts are standardized to the volume (63.62 cm³) of the sampled cylinder of the sediment (core). Gray dashed line shows approximate boundaries of the oxygen minimum zone.


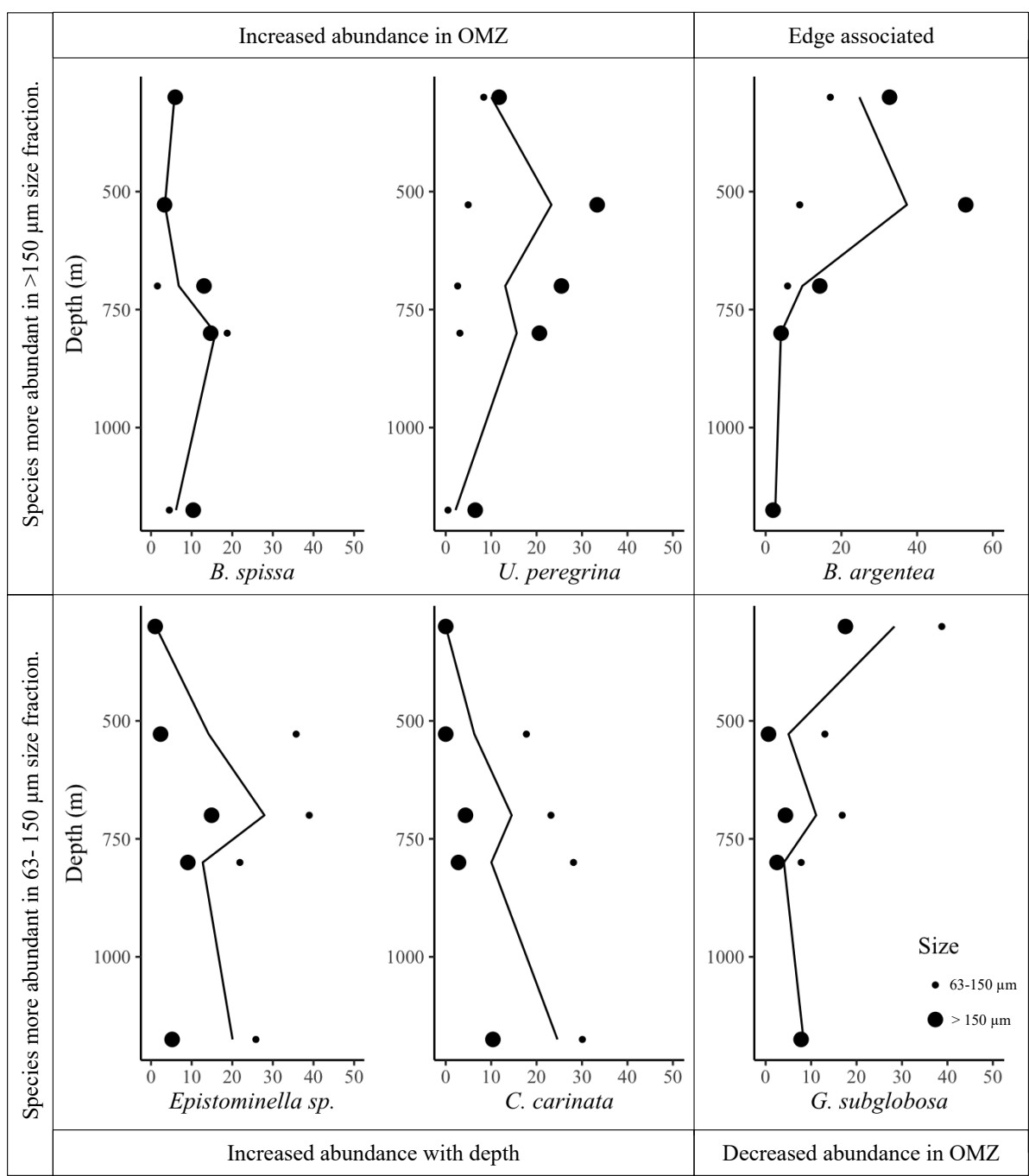


**Figure 3:** Relative abundance of foraminiferal species (percent of total calcareous taxa) in core top sample (0-2 cm) vs. water depth (m). Large black dots are >150 µm size fraction, small black dots are 63-150 µm size fraction and black lines represent trends considering the complete assemblage (>63 µm).


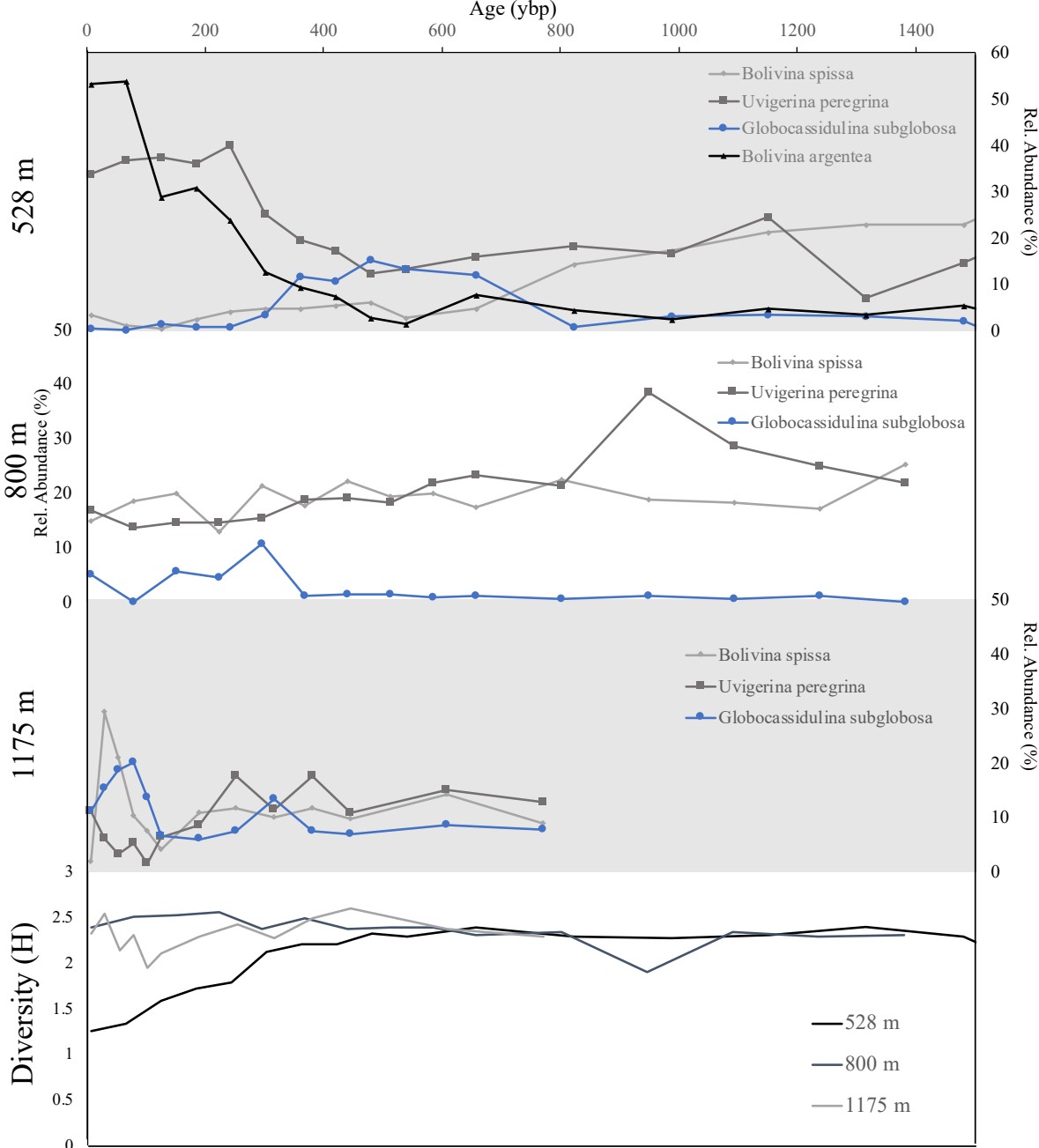

**Figure 4:** Top 3 panels show relative abundance of two species of hypoxia indicator foraminifera (*B. spissa* and *U. peregrina*, gray lines) and one species of oxic indicator foraminifera (*G. subglobosa*, blue lines) from the >150 µm size fraction down core through time, in years before present for cores from 3 water depths (528 m, 800 m, 1175 m). Top panel also includes relative abundance of *B. argentea* (edge-associated). Bottom panel shows diversity (Shannon's Index, H) through time for 3 cores.

**Tables**

| Core Name | Water Depth (m) | Latitude | Longitude | Core Length (cm) | Temperature (°C) | Dissolved Oxygen (mL/L) | pH | Salinity (psu) | Total organic matter (% wt) |
|---|---|---|---|---|---|---|---|---|---|
| MV1217-5-2 | 300 | 32.8100166 | 117.468100 | 16 | 8.614 | 1.54 | 7.65 | 34.145 | 6.8 |
| MV1217-2-3 | 528 | 32.8100333 | 117.416583 | 26 | 6.622 | 0.35 | 7.57 | 34.313 | 11.7 |
| MV1217-3-3 | 700 | 32.8099666 | 117.450966 | 20 | 5.898 | 0.26 | 7.56 | 34.348 | 13.9 |
| MV1217-1-3 | 800 | 32.8095166 | 117.506933 | 20 | 5.049 | 0.29 | 7.56 | 34.405 | 14.2 |
| MV1217-4-3/1 | 1175 | 32.6333333 | 117.499883 | 16 | 3.823 | 0.58 | 7.59 | 34.501 | 14.7 |

**Table 1:** Data for cores used in this study. Temperature, dissolved oxygen, pH and salinity were measured in bottom water directly above each coring site.

| Core | Sample Interval | Age ($^{14}$C years) | ± | 1 Sigma Maximum Calendar Age Range | 1 Sigma Minimum Calendar Age Range | Age in Calendar Years | Sedimentation Rate (cm/ka) | ± |
|---|---|---|---|---|---|---|---|---|
| MV1217-2-3 | 11-12 cm | 1230 | 30 | 1403 | 1319 | 1361 | 16.9 | 1.0 |
| MV1217-2-3 | 16-17 cm | 2085 | 30 | 602 | 474 | 538 | 6.1 | 0.2 |
| MV1217-2-3 | 25-26 cm | 2405 | 35 | 237 | 107 | 172 | 24.6 | 0.1 |
| MV1217-4-3 | 5-6 cm | 670 | 35 | 1950 | 1837 | 1893.5 | 42.2 | 11.3 |
| MV1217-4-1 | 10-11 cm | 960 | 30 | 1630 | 1518 | 1574 | 15.6 | 0.2 |
| MV1217-4-1 | 20-21 cm | 1840 | 35 | 817 | 698 | 757.5 | 12.3 | 0.4 |

**Table 2.** Radiocarbon ages of mixed planktonic foraminifera from MV1217-2-3 (528 m water depth), MV1217-4-3 (1175 m water depth) and MV1217-4-1 (1175 m water depth).