# Peer review of "Southern California margin benthic foraminiferal assemblages record recent centennial-scale changes in oxygen minimum zone"

_Biogeosciences, 2019_

## Referee Comment (RC1) · Anonymous Referee #1 · 9 Jan 2020

The paper titled, "Southern California margin benthic foraminiferal assemblages across a modern environmental gradient record recent centennial-scale changes in oxygen minimum zone" quantifies benthic foraminifera abundances and assemblages of surface multicore sections (0-2cm) and downcore (∼20cm). Overall, despite the shortcomings in the methodology, and lack of clarity in the writing style of this manuscript, the information offered in this study shows potential in its importance to current and future oxygen investigations using benthic foraminifera. I recommend this manuscript undergo Major Revisions including a significant re-write, standardization of data, and

incorporate a more thorough literature review.

This manuscript is well organized and contains most of the necessary information, but would benefit from re-writing and shortening of certain sections for clarity and readability. Suggested edits, such as changes in the present and past tenses and excluding the first-person point of view writing are included in the "Technical Corrections" section below. As is currently written the manuscript needs to be tied to established literature references and include a much more comprehensive discussion section justifying methods and interpretations. There also needs to be a section acknowledging the limitations of the regional interpretations made here. The authors state "Our findings indicate that region (or environment) specific oxygen species associations may be necessary, as our findings do not align directly with previous categorization of species." It is important that it is recognized that these results, although true in the Southern California Margin, may not show the same patterns elsewhere in the world's oceans and/or for the authors to compare their results with those previously published. Based on the current manuscript an understanding of how strong the interpretations are is unclear. What additional data would be needed to test or refine the applications and interpretations? All of these issues, limitations, and confounding factors may be addressed in a Major Revision.

Specific comments regarding scientific questions/issues:

1. How are raw abundance counts standardized? Are all abundances reported as raw numbers? If so, how do you account for inconsistencies in the amount of sediment examined in each interval? The data should be standardized to #/50cc or #/gram.

2. It is unfortunate that samples were not able to be stained using Rose Bengal, however, I do still believe the data to be extremely valuable in that it is necessary to examine what surficial fossil assemblages are when using them down core. Rose Bengal stained samples here would have completed the picture, but the information available from this data set is valuable and needed for fossil interpretations. I am not sure how

you make the assertion that examination of only the larger size fraction (>150micron) is suitable for paleooxygen investigations, at least, not with the text in its current form. Additional references and discussions are needed. Below is what I pulled from the manuscript and viewed as the line of thinking on the subject. Please reorganize and emphasize the statements to better clarify this argument. -Lines 200- Section 3.3: -Lines 291: "species richness is greater in the >150 $\mu$m relative to the 63-150 $\mu$m fraction, yet there is no consistent relationship between diversity and size fraction ". Is this your argument for looking only at the >150 micron size fraction? If so, you need to include a better discussion of the relevant literature which suggests the >63 micron fraction is necessary to examine in environments with low oxygen and high carbon inputs. -Line 297: "diversity is not driven by oxygen alone" And there is no clear pattern in diversity and size. So, what is the basis on which you are making the argument that only the >150 microns are necessary in oxygen related studies? I suggest referencing Keating-Bitonti and Payne, 2016 when writing this section. You include this as a reference, but you don't really discuss or justify referencing it. -Lines 314-316: "Our findings show that spatial trends in the >150 $\mu$m size fraction generally reflect those found in the >63 $\mu$m size fraction or are muted by the inclusion of the 63-150 $\mu$m." Have you presented this data for the readers to see? Is this Figure 3? If so, reference the figure here. If not, you need to provide supporting evidence. -Lines 324-325: "...correlation between size of specimen and accuracy of identification, meaning that the inclusion of the smaller specimens in the >63 $\mu$m fraction may reduce the accuracy of identification . . ." This is a good point and should be discussed sooner for better emphasis.

3. A better discussion of how agglutinated foraminifera are reported, or not, in the total foraminiferal counts should be included. I see no reason why not to include them in the abundances reported (is that what you are already doing?). Then simply report percentages of certain species within the calcareous population if that is what you are interested in (e.g., Agg sp. 1 comprises 10% of the total population; Cal sp. 1 comprises 30% of the total population and 50% of calcareous population). I think excluding agglutinated foraminifera populations will impede on future research of agglutinated

species.

4. A more complete literature review is needed in the discussion section in order to support the findings of the authors. I suggest two discussion/literature review sections: 1) examination of <150 micron (>63 or 63-150 micron) foraminifera in low oxygen/high carbon environments. Do an overview of who proposed it, what results have been seen, and how it relates to your data. 2) foraminifera test size in comparison with oxygen. Here would be a great chance to discuss Keating-Bitonti and Payne, 2016 in greater detail. Again, do an overview of who proposed it, what results have been seen, and how it relates to your data.

Technical Corrections

-Line 22: "San Diego Margin" is capitalized here and in the text, but not in the title. Choose one and be consistent. -Line 22: "Here, we" can be removed. Start the sentence with, "Five core tops were analysed . . ." -Line 24: Assemblage changes downcore? If so, please state. -First-person point of view should not be used in scientific writing. Please modify the text accordingly. -Line 29: "diversity is not driven by oxygenation" In the core top materials? Is this pattern observed with Shannon Index downcore? -Line 30: "variability in [the] upper margin" -Line 31: "stable in the last 1.5ka" add a space before ka to remain consistent with the formatting of other units. -Line 42: "declines in dissolved oxygen," awkward. Please rephrase. -Line 43: determination of -Line 49: "Low-oxygen zones typically contain both low abundance and low diversity of" change to "Areas of low oxygen availability typically contain low abundance and diversity of " -Line 73: "2013), yet significantly fewer" remove ", yet". Begin the sentence with "Significantly. . ." Do you have any examples to cite e.g.,? -Line 75: "biological and chemical gradients are more extensively distributed" or more extensively variable? -Line 82: remove "current, the" or something I just don't like how it is phrased. -Line 85: too many back-to-back parentheses. –This entire paragraph should be shortened. Along the California margin, a large, intermediate-depth oxygen minimum zone (OMZ) and carbon maximum zone (CMZ; approx.. 500 – 1000 m
water depth, Helly and Levin, 2004; Stramma et al., 2010) are formed and controlled through physical and biological processes including temperature-dependent diffusion from atmosphere, mixing, stratification, deep water circulation, primary productivity at the surface and respiration at depth (Gilly et al., 2013; Helly and Levin, 2004). Intensity and spatial extent of the modern California margin OMZ is influenced by physical mixing of well-oxygenated surface water, biological activity at the surface and at depth, and intrusion of lower oxygen bottom waters (Gilly et al., 2013). -Line 107: change core to coring -Lines 107-108: . . ."CTD (SBE9), with a dissolved oxygen probe (Seabird Electronics Sensor SBE43). . ."- CTDs typically have an oxygen probe. Is it necessary to state this? -Lines 110-111: "Bottom depths were measured acoustically at each site."- typically have an oxygen probe. Is it necessary to state this? -Line 117-118: "frozen. Sediments were not stained upon retrieval; thus, we could not carry out an analysis of live vs. dead or depth habitat of these species" This is too bad. It would have made for a much stronger study if living and fossil assemblages were able to be extracted from core top samples. I don't think the data presented here is useless, but having stained (recently living) samples to compare with the dead assemblages at the surface would have provided a much more powerful interpretation. -Line 119: (Grupe. . .2015) should not be italicized. -Lines 123-128: The word sample is used too many times (5 in one sentence). Try to replace with another word to avoid monotony. -Line 130: "present in the sample." I'm assuming it is in the entire sample? If so, add "entire" before "sample" -Line 132: I would not qualify 1-2cm as surface. It is shallow infaunal. -Lines 133-134: ". . . were examined at 1 cm intervals for cores MV1217-2-3 (528 m), MV1217-1-3 (800 m) and MV1217-4-3 (1175 m)." Ending at what depth in the cores? -Lines 132-134: These two sentences should be combined with the previous paragraph as 2 sentences is not a paragraph on its own. -Line 156: no units on salinity. Remove "with a total" replace "range of" with "ranging from" -Line 157: (DO) once you have introduced an abbreviation you do not have to reference it again and you can then use the shorthand in the text. I suggest you do a search of the manuscript and identify duplicates of instances such as this. I suggest a rewriting of this section for clarity. Either report based on water column depth or minimums. As is, it is confusing. –Rephrasing suggestion: —"Water column dissolved oxygen (DO) concentration documents a low oxygen zone, with a minimum occurring at 700 m water depth (0.26 ml/L; Figure 2, Table 1), compared to 1.54 ml/L at 300 m and 0.58 ml/L at 1175 m. Minimum pH is documented at 700 m (7.55) and is higher above (300 m, 7.65) and below (1175 m, 7.59) the intermediate depth low pH zone (Figure 2, Table 1)." —Water column DO measurements indicate areas of low oxygen availability from 300 m (1.54 ml/L) to 1175 m (0.58 ml/L) with lowest oxygen availability at 700 m (0.26 ml/L). Although not greatly variable, pH minima also occur at 700 m (7.55) and is higher at 300 m (7.65) and 1175 m, (7.59). In this section, why not report as hypoxic, anoxic, as outlined by Bernhard et al? -Section 3.2: this section is written in present tense while the rest is in past tense. Change all "was" to "were", "is" to "was", etc. –Where is this information reported? A graph a table? Reference it in this section. -Lines 169-172: "Foraminiferal abundance is low (<100 individuals) in some of the samples from 0-1 cm. Thus, in order to utilize sufficient numbers of individuals and because there were no significant differences in abundance of species between 0-1 cm and 1-2 cm, for the rest of the discussion we refer to the 0-2 cm fraction as the core top material." This needs to be identified in Figure 3. -Line 173: Remove "The site" and begin the sentence with "At". Change "at the core of the OMZ" to "within the core of the OMZ, the". Add "occurred" to the end of the sentence. Alternatively, this sentence could be shortened and combined with the previous sentence: Calcareous taxa dominated the assemblage at every site; agglutinated foraminifera made up 0 (e.g., XXX m) to 17.7% of the assemblage at 700 m, within the core of the OMZ. -Line 175: "Due to their propensity for degradation and to remain consistent with other regional studies, we exclude agglutinated taxa and all values are reported as percent of total calcareous taxa for the remainder of the text." I don't disagree with this statement, but I think that ignoring the agglutinates is a mistake we will regret in the future. Why not keep the data (they were a significant portion of the population) and just report Calcareous populations in the graphs etc? -Line 179: "sp1" can this just be written as "sp."? -Lines 180-181: "These dominant

taxa make up more than 84% of all foraminifera counted across all core top samples." Does this include the agglutinates or just 84% of the calcareous foraminiferal population? -Line 181: "All other species each account for <4% of total assemblage across all core tops." This sentence is confusing. Please rephrase. -Line 200: "of" repeated. Could streamline to, "Comparisons of foraminifera abundances between. . ." -Line 200: change 150 to >150 microns. -Lines 200- Section 3.3: Based on the results of this section why are you concluding that it is better to look at the >150micron size fraction as you state in the conclusion? -Lines 202-204: "Three species have significantly different relative abundances between size classes; three are more likely to be found in the 63-150 $\mu$m (Cassidulina carinata, Epistominella sp1 and G. subglobosa)". When you repeat "three" are you referring to the same 3 species? If so, please rephrase. -Line 207: I don't think it is necessary to repeat "Shannon Index" after you describe it in the methodology. -Line 217: When you say above and below the OMZ do you mean on the seafloor? Not in the water column? Please clarify. G. subglobosa as oxic indicator- This assertion needs a discussion or reference. Skipped to discussion -Line 295-296: I'm not sure this was the findings of Venturelli et al., 2018. Which section of the paper are you referring to here? The focus of this paper was on comparing sediment grain size, oxygen, and foraminiferal abundances in the Southern California Bight. It was not an OMZ survey and therefore I am not sure they proposed seeing variations of foraminiferal abundances in size fractions "within" the OMZ vs outside the OMZ. If you just mean to say that 63-150 micron foraminifera were more abundant than >150 micron reported by Venturelli et al., 2018 I think this is true, but how different were the populations and would grain size difference influence these abundances? -Line 296: Shannon Index -Line 341: remove DO as it is implied -Lines 355-338: "The presence of oxic indicator species across all water depths may provide evidence for periodic flushing of high oxygen water or a selection for species that can tolerate a range of environmental conditions rather than a specific threshold of oxygenation." It may also be that the physiological tolerances of indicator species are not fully understood. They may be able to tolerate lower oxygen than previously thought provided

another incentive such as substrate (e.g., Venturelli et al., 2018, Burkett et al., 2016), and or they may be able to tolerate short periods of low oxygen or inhospitable conditions (Bernhard et al., 2010). -Line 262: "see Discussion section" can you refer to a specific section number e.g., 4.3? -Line 263: "core top[s]" -Lines 268-269: please rephrase this sentence for clarity. It is hard to understand your meaning. -Line 270: What do you mean by equal in magnitude? Can you elaborate on that? Perhaps by giving total abundance or percent abundance examples? -Lines 273- 279: shorten this section to make your findings more impactful. The word "document" is repeated and could be eliminated completely. -Lines 300-302: Where can the readers see this stated relationship? "...and the >150 $\mu$m size fraction or 2) trends in the >150 $\mu$m size fraction are more pronounced than in the complete assemblage" -Lines 300-310: YES! I totally agree with some of the things you are saying in this section, but you have to do a complete discussion of the literature. I would suggest splitting it up into two sections 1) examination of <150 micron foraminifera in low oxygen/high carbon environments. Do an overview of who proposed it, what results have been seen, and how it relates to your data. The second literature comparison should be 2) foraminifera test size in comparison with oxygen. Here would be a great chance to discuss Keating-Bitonti and Payne, 2016 in greater detail. Again, do an overview of who proposed it, what results have been seen, and how it relates to your data. -Lines 338-339: "Infaunal species are more common within the OMZ, while epifaunal are more common in well-oxygenated areas." True, but there are studies that illustrate that epifaunal abundances may be limited by substrate rather than a physiological limitation to oxygen availability (see comment above). You just alluded to this in the previous sentence, but I suggest incorporating something of this statement into this sentence or starting the sentence with, "In general". -Line 440: remove "the"; change "oxygenation" to "oxygen" -Lines 442 and 443: change "classes" to "fractions" -Lines 444-445: "we conclude that analysis of the >150 $\mu$m assemblage for this site provides the most useful record for interpreting benthic foraminifera as a proxy for past change" This assertion needs a discussion or reference -Line 447: "variability in upper margin of the OMZ" should be "variability in

BGD

Interactive
comment

[the] upper margin of the OMZ" -Line 447: by "core" do you mean center? Is there another word you can use here so as not to confuse it with sediment cores? -Lines 448-449: "We document expansion of upper margin of OMZ beginning 400 ybp on San Diego Margin that is synchronous with regional records of oxygenation." Should be re-written. Perhaps, "In this study, upper margin OMZ expansion beginning 400 ybp on San Diego Margin is synchronous with regional records of oxygenation."? -Data availability: is the section with data files raw counts? Or is this standardized per volume? See discussion in the "scientific questions/issues" section. -Figure 2: Extra ")"; -Figure 3: "General observations discussed in the text are noted here, e.g., species that increase in abundance in the OMZ, appear associated with the "edge" of the OMZ, etc. Note difference in x-axis in B. argentea plot" is not very useful information for a figure caption. Please describe the structure of the graph and summarize what you observed or reference to the section of the paper where it is discussed. Please also clearly state in this figure that "core tops" are the 0-2cm intervals. -Based on the OMZ bounds of Figure 2 it seems the majority of the foraminiferal abundance plots resides in what you have defined at the OMZ. So how can you see increases if you have no "background" to compare it to? Please clarify. The key should be bounded by a box to better identify it. -Figure 4: The Bolivina spissa and U. peregrina lines are very similar in color and hard to distinguish. Can the point shapes be changed to better facilitate reading? -Table 1: Include salinity in the table. -Table 2: Contains a core not "used in this study" MV1217-4-1. If it is to be included in this table it would be helpful to also include depth and lat-long so readers have an idea of why you are including this in the manuscript. 14 in the 14C needs to be superscripted

---

## Referee Comment (RC2) · Anonymous Referee #2 · 24 Jan 2020

This manuscript entitled "Southern California margin benthic foraminiferal assemblages across a modern environmental gradient record recent centennial-scale changes in oxygen minimum zone" by Palmer et al. presents a valuable dataset from a gradient of one of the most prominent OMZs in the world. It presents the calcareous benthic foraminiferal assemblages focusing in size fractions from core tops along a depth transect. Later the authors investigate these assemblages in short cores in order to investigate the recent history of the OMZ and the benthic foraminiferal assemblages. The information provided here is an important input for the ongoing investigations re-

garding the relationship between OMZs, ecosystem and climate. Overall, I found some major details missing in this study and I believe it can be improved significantly.

1. This study is based on benthic foraminifera taxonomy work which should include references to species nomenclature also preferably a plate showing the major species mentioned. In case it is not possible to provide figures, there should be a section where list of species observed is given with references used for identification. For example: Bolivina spissa = Bolivina subadvaena Cushman var. spissa Cushman 1926a. [Figures 10.7, 11.4]. This is essential for taxonomy based papers where the reader will be able to evaluate the information provided. The number of publications without any reference material is increasing and this leads the misinterpretations regarding the foraminifera research. The authors mentions their concerns in the discussion therefor I highly encourage this MS to have section dedicated to nomenclature.

2. Introduction and discussion should be improved in terms of using literature and previous work from different OMZs. For instance there are significant amount of work from the Peruvian and Arabian OMZs focusing on similar oxygen gradient and benthic foraminiferal assemblages. These studies should be included in terms of benthic foraminifera habitat in relation with oxygen and nitrate availability etc. This will improve the MS significantly. It is a pity that the species are not stained limiting the comparison with previous studies, and yet I believe the information presented here is really valuable.

3. Presentation of environmental parameters is confusing. Are these values from measurements of bottom water conditions? deepest depth of CTD? Figure 2 should be improved accordingly where stations can be shown.

4. Definition of an OMZ: please introduce OMZ already in introduction. This MS uses certain terms such as OMZ edge, suboxia, hypoxic boundary and so on; to eliminate the confusion, edge or boundary of an OMZ considered here should be introduced as early in the text as possible.

Notes for specific parts: page 2, line 60: please check Tetard et al., 2017.

paragraph starting with line 67: this section could be improved significantly by including previous observations from other OMZs which should be included in discussion where Bolivinids and nitrate availability are discussed.

page 3, line 107: should be "dissolved oxygen concentration"

page 4, section under 2.3. needs to be rewritten considering the steps taken to reach the species counts. first, material sieved, dried and count in different fractions. Which references were considered for 300 and 150 specimens? Why did the authors decide these numbers? I am not an expert for statistical methods but what is the reason behind using dbRDA but not component analysis (CCA?) to test the relationship between foraminifera and environmental parameters?

line 145: "...mixed planktonic foraminifera species" please remove bulk

section 3.1.: this section is confusing. please be clear with what is presented here. I assume these are the deepest points CTD measured? is there any pore water measurements or are these only water column? and salinity should be included as well in the table.

line 205: is ANOVA introduced already in methods?

line 222: the term edge dominant.. what does this actually mean? According to which previous work edge of the OMZ is considered?

line 224: sentence starting with "in some taxa,.." needs rephrasing. paragraph starting from line 268: this could be written much simpler, I am not sure I understand the information provided here.

page 9 first paragraph: we know today oxygen limited high organic matter input regions are characterised by high population low diversity of benthic foraminifera. it is interesting to see this is not the case at these sites. Nevertheless, I am not fully convinced

the evidence provided in this study is enough to come to this conclusion. What do the authors think, if stained species were considered only the results would show any difference or not?

line 301: what does "...size fraction or 2" mean? paragraph starting at line 312: for such a discussion based on specific species, authors should provide a reference list including species names as mentioned earlier.

line 341: this is the first time specific oxygen concentration and terminology is given. This should come earlier. paragraph starting at line 365: discussion on Bolivinids: there are so many studies on bolivinids at similar setups, those should be mentioned and discussed here. Some examples are: Mallon et al., 2012; Cardich et al., (several papers), Glock et al., (several papers), Caulle et al., 2014; Jannink et al., 1998. the list goes on.

line 387: Please rephrase the last sentence. paragraph starting with line 424: it would be nice to compare results with previous observations from the region.

comment on data availability: will it be open access upon publication?

Title: I think title could be simplified and improved.

Figure 1: please give more information in the figure caption including which sites have what kind of results in the text. what are the depths of these sites?

Figure 2: water depth on y axis? station names could be implemented.

Figure 3 caption: "General observations .... " this is not needed here. Figure should be cited in the text more often.

Figure 4: y axis please mention Rel. Abundance (%) instead.

Table 1: please add salinity and the captions should be more informative including where this information comes from. CTD? porewater? what is TOM?

Table 2 caption: mixed planktonic foraminifera species. please remove bulk.

[Figure]

---

## Author Response (AR1)

Author's response for:

**Southern California margin benthic foraminiferal assemblages record recent centennial-scale changes in oxygen minimum zone**

Hannah M. Palmer, Tessa M. Hill, Peter D. Roopnarine, Sarah E. Myhre, Katherine R. Reyes, and Jonas T. Donnenfield

This document contains:

Response to Referee Comment 1
Response to Referee Comment 2
A marked-up manuscript with track-changes included

**Response to Referee Comment 1**

RC1: Referee comment 1, in gray
AC: Author comment, in black

AC: We thank Referee 1 for their helpful comments and suggestions to improve the manuscript. We have addressed each major comment below. We will incorporate all technical corrections listed by Referee 1 unless a comment is made below.

RC1: The paper titled, "Southern California margin benthic foraminiferal assemblages across a modern environmental gradient record recent centennial-scale changes in oxygen minimum zone" quantifies benthic foraminifera abundances and assemblages of surface multicore sections (0-2cm) and downcore (_20cm). Overall, despite the shortcomings in the methodology, and lack of clarity in the writing style of this manuscript, the information offered in this study shows potential in its importance to current and future oxygen investigations using benthic foraminifera. I recommend this manuscript undergo Major Revisions including a significant re-write, standardization of data, and incorporate a more thorough literature review.

This manuscript is well organized and contains most of the necessary information, but would benefit from re-writing and shortening of certain sections for clarity and readability. Suggested edits, such as changes in the present and past tenses and excluding the first-person point of view writing are included in the "Technical Corrections" section below. As is currently written the manuscript needs to be tied to established literature references and include a much more comprehensive discussion section justifying methods and interpretations. There also needs to be a section acknowledging the limitations of the regional interpretations made here. The authors state "Our findings indicate that region (or environment) specific oxygen species associations may be necessary, as our findings do not align directly with previous categorization of species." It is important that it is recognized that these results, although true in the Southern California Margin, may not show the same patterns elsewhere in the world's oceans and/or for the authors to compare their results with those previously published. Based on the current manuscript an understanding of how strong the interpretations are is unclear. What additional data would be needed to test or refine the applications and interpretations? All of these issues, limitations, and confounding factors may be addressed in a Major Revision.

AC: We thank the reviewer for this feedback; as shown below, we will adapt the writing of the manuscript, address data standardization, and incorporate a more thorough literature review.

RC1: Specific comments regarding scientific questions/issues:
RC1: 1. How are raw abundance counts standardized? Are all abundances reported as raw numbers? If so, how do you account for inconsistencies in the amount of sediment examined in each interval? The data should be standardized to #/50cc or #/gram.

AC: Raw abundance counts
The abundance counts are done by volume of sediment relative to total, then multiplied to equal the "total" volume. Thus, these should represent the amount of foraminifera in the originally

sampled cylinder of sediment (core) 1cm height x 9 cm diameter = 63.62 cm^3. This will be updated in the Methods section of the paper and in Figure 2.

RC1: 2. It is unfortunate that samples were not able to be stained using Rose Bengal, however, I do still believe the data to be extremely valuable in that it is necessary to examine what surficial fossil assemblages are when using them down core. Rose Bengal stained samples here would have completed the picture, but the information available from this data set is valuable and needed for fossil interpretations. I am not sure how you make the assertion that examination of only the larger size fraction (>150micron) is suitable for paleooxygen investigations, at least, not with the text in its current form. Additional references and discussions are needed. Below is what I pulled from the manuscript and viewed as the line of thinking on the subject. Please reorganize and emphasize the statements to better clarify this argument. -Lines 200- Section 3.3:
-Lines 291: "species richness is greater in the >150 μm relative to the 63-150 μm fraction, yet there is no consistent relationship between diversity and size fraction ". Is this your argument for looking only at the >150 micron size fraction? If so, you need to include a better discussion of the relevant literature which suggests the >63 micron fraction is necessary to examine in environments with low oxygen and high carbon inputs.
-Line 297: "diversity is not driven by oxygen alone" And there is no clear pattern in diversity and size. So, what is the basis on which you are making the argument that only the >150 microns are necessary in oxygen related studies? I suggest referencing Keating-Bitonti and Payne, 2016 when writing this section. You include this as a reference, but you don't really discuss or justify referencing it. -Lines 314-316: "Our findings show that spatial trends in the >150 μm size fraction generally reflect those found in
the >63 μm size fraction or are muted by the inclusion of the 63-150 μm." Have you presented this data for the readers to see? Is this Figure 3? If so, reference the figure here. If not, you need to provide supporting evidence.
-Lines 324-325: "...correlation between size of specimen and accuracy of identification, meaning that the inclusion of the smaller specimens in the >63 um fraction may reduce the accuracy of identification
. . ." This is a good point and should be discussed sooner for better emphasis.

AC: After submission of the manuscript, we realized that there was a graphical error that affected the plotting of some of the data in Figure 3. The interpretations of the figure in the text were accurate, it was the figure itself that did not display correctly. We have uploaded an updated version of Figure 3 in response to this comment and the updated figure should help to address the questions posed here by Referee #1. In this figure, it is apparent that "spatial trends in the >150 μm size fraction generally reflect those found in the >63 μm size fraction or are muted by the inclusion of the 63-150 μm." This was unclear in the previous iteration of the figure but has now been made clearer. We will cite this figure in the text more heavily to reference for understanding.

RC1: 3. A better discussion of how agglutinated foraminifera are reported, or not, in the total foraminiferal counts should be included. I see no reason why not to include them in the abundances reported (is that what you are already doing?). Then simply report percentages of certain species within the calcareous population if that is what you are interested in (e.g., Agg sp. 1 comprises 10% of the total population; Cal sp. 1 comprises 30% of the total population and

50% of calcareous population). I think excluding agglutinated foraminifera populations will impede on future research of agglutinated species.

AC: We appreciate the interest in agglutinated species. While they were counted in this study, they were not speciated. Additionally, as is discussed further below, they are not well preserved so not utilized for investigations that reconstruct environments through time. For this reason, it is the industry standard to only utilize calcareous species in relative abundance analyses.
We noted other papers have reported on agglutinated taxa in the same way (Venturelli et al 2018) or have excluded examination of all agglutinated (Balestra et al 2017, Kaiho 1994, and Myhre et al 2014). In an effort for our study to be comparable to other regional studies we chose to report relative abundance as percent of total calcareous taxa. We acknowledge that due to the lack of preservation of agglutinated foraminifera, fossil assemblages may capture an incomplete reconstruction of diversity and ecosystem function.

RC1: 4. A more complete literature review is needed in the discussion section in order to support the findings of the authors. I suggest two discussion/literature review sections: 1) examination of <150 micron (>63 or 63-150 micron) foraminifera in low oxygen/high carbon environments. Do an overview of who proposed it, what results have been seen, and how it relates to your data. 2) foraminifera test size in comparison with oxygen. Here would be a great chance to discuss Keating-Bitonti and Payne, 2016 in greater detail. Again, do an overview of who proposed it, what results have been seen, and how it relates to your data.

AC: We will incorporate a more thorough literature review of the distribution of foraminifera in low oxygen/high carbon environments and of foraminiferal test size in relation to oxygen. We will include Keating-Bitonti and Payne 2016, Keating-Bitonti and Payne 2018, Kaiho et al 1994, Kaiho et al 1999, Venturelli et al 2019, and others.
* * *
RC1: Technical Corrections
AC: We will incorporate **all** technical corrections listed by Referee 1 that are listed below

-Line 22: "San Diego Margin" is capitalized here and in the text, but not in the title. Choose one and be consistent.
-Line 22: "Here, we" can be removed. Start the sentence with, "Five core tops were analysed ."
-Line 24: Assemblage changes downcore? If so, please state. -First-person point of view should not be used in scientific writing. Please modify the text accordingly.
-Line 30: "variability in [the] upper margin"
-Line 31: "stable in the last 1.5ka" add a space before ka to remain consistent with the formatting of other units.
-Line 42: "declines in dissolved oxygen," awkward. Please rephrase.
-Line 43: determination of
-Line 49: "Low-oxygen zones typically contain both low abundance and low diversity of" change to "Areas of low oxygen availability typically contain low abundance and diversity of "

-Line 73: "2013), yet significantly fewer" remove ", yet". Begin the sentence with "Significantly. . ." Do you have any examples to cite e.g.,? -

-Line 82: remove "current, the" or something I just don't like how it is phrased.

-Line 85: too many back-to-back parentheses. –This entire paragraph should be shortened. Along the California margin, a large, intermediate-depth oxygen minimum zone (OMZ) and carbon maximum zone (CMZ; approx.. 500 – 1000 m water depth, Helly and Levin, 2004; Stramma et al., 2010) are formed and controlled through physical and biological processes including temperature-dependent diffusion from atmosphere, mixing, stratification, deep water circulation, primary productivity at the surface and respiration at depth (Gilly et al., 2013; Helly and Levin, 2004). Intensity and spatial extent of the modern California margin OMZ is influenced by physical mixing of well-oxygenated surface water, biological activity at the surface and at depth, and intrusion of lower oxygen bottom waters (Gilly et al., 2013).

-Line 107: change core to coring

-Line 119: (Grupe. . .2015) should not be italicized.

-Lines 123-128: The word sample is used too many times (5 in one sentence). Try to replace with another word to avoid monotony.

-Line 130: "present in the sample." I'm assuming it is in the entire sample? If so, add "entire" before "sample"

-Lines 132-134: These two sentences should be combined with the previous paragraph as 2 sentences is not a paragraph on its own.

-Section 3.2: this section is written in present tense while the rest is in past tense. Change all "was" to "were", "is" to "was", etc. –Where is this information reported? A graph a table? Reference it in this section.

-Lines 169-172: "Foraminiferal abundance is low (<100 individuals) in some of the samples from 0-1 cm. Thus, in order to utilize sufficient numbers of individuals and because there were no significant differences in abundance of species between 0-1 cm and 1-2 cm, for the rest of the discussion we refer to the 0-2 cm fraction as the core top material." This needs to be identified in Figure 3.

-Line 173: Remove "The site" and begin the sentence with "At". Change "at the core of the OMZ" to "within the core of the OMZ, the". Add "occurred" to the end of the sentence. Alternatively, this sentence could be shortened and combined with the previous sentence: Calcareous taxa dominated the assemblage at every site; agglutinated foraminifera made up 0 (e.g., XXX m) to 17.7% of the assemblage at 700 m, within the core of the OMZ.

-Line 179: "sp1" can this just be written as "sp."?

-Line 181: "All other species each account for <4% of total assemblage across all core tops." This sentence is confusing. Please rephrase.

-Line 200: "of" repeated. Could streamline to, "Comparisons of foraminifera abundances between. . ."

-Line 200: change 150 to >150 microns.

-Lines 202-204: "Three species have significantly different relative abundances between size classes; three are more likely to be found in the 63-150 μm (Cassidulina carinata, Epistominella sp1 and G. subglobosa)". When you repeat "three" are you referring to the same 3 species? If so, please rephrase.

-Line 207: I don't think it is necessary to repeat "Shannon Index" after you describe it in the methodology.

-Line 296: Shannon Index -Line 341: remove DO as it is implied
-Line 262: "see Discussion section" can you refer to a specific section number e.g., 4.3?
-Line 263: "core top[s]"
-Lines 268-269: please rephrase this sentence for clarity. It is hard to understand your meaning.
-Lines 273- 279: shorten this section to make your findings more impactful. The word "document" is repeated and could be eliminated completely.
-Lines 338-339: "Infaunal species are more common within the OMZ, while epifaunal are more common in well-oxygenated areas." True, but there are studies that illustrate that epifaunal abundances may be limited by substrate rather than a physiological limitation to oxygen availability (see comment above). You just alluded to this in the previous sentence, but I suggest incorporating something of this statement into this sentence or starting the sentence with, "In general".
-Line 440: remove "the"; change "oxygenation" to "oxygen"
-Lines 442 and 443: change "classes" to "fractions" -Lines 444-445: "we conclude that analysis of the >150 μm assemblage for this site provides the most useful record for interpreting benthic foraminifera as a proxy for past change" This assertion needs a discussion or reference
-Line 447: "variability in upper margin of the OMZ" should be "variability in [the] upper margin of the OMZ"
-Lines 448-449: "We document expansion of upper margin of OMZ beginning 400 ybp on San Diego Margin that is synchronous with regional records of oxygenation." Should be re-written. Perhaps, "In this study, upper margin OMZ expansion beginning 400 ybp on San Diego Margin is synchronous with regional records of oxygenation."?
-Figure 2: Extra ")";
-Figure 4: The Bolivina spissa and U. peregrina lines are very similar in color and hard to distinguish. Can the point shapes be changed to better facilitate reading?
-Table 1: Include salinity in the table
-Table 2: Contains a core not "used in this study" MV1217-4-1. If it is to be included in this table it would be helpful to also include depth and lat-long so readers have an idea of why you are including this in the manuscript. 14 in the 14C needs to be superscripted
* * *
RC1: Technical Corrections
AC: We have addressed or commented on the following suggested changes:

-Line 29: "diversity is not driven by oxygenation" In the core top materials? Is this pattern observed with Shannon Index downcore?

AC: Diversity is not driven by oxygenation in the core top samples. This statement is not referring to the downcore changes in diversity. We will clarify this in the text.

Line 75: "biological and chemical gradients are more extensively distributed" or more extensively variable?

AC: We will change the text to "more variable."

-Lines 107-108: . . ."CTD (SBE9), with a dissolved oxygen probe (Seabird Electronics Sensor SBE43). . ."- CTDs typically have an oxygen probe. Is it necessary to state this?

AC: We included this to show that values for dissolved oxygen were acquired using a sensor rather than by bottle sampling and Winkler titration.

-Lines 110-111: "Bottom depths were measured acoustically at each site." Is it necessary to state this?

AC: For completeness, we included how each of the environmental parameters were measured.

-Line 117-118: "frozen. Sediments were not stained upon retrieval; thus, we could not carry out an analysis of live vs. dead or depth habitat of these species" This is too bad. It would have made for a much stronger study if living and fossil assemblages were able to be extracted from core top samples. I don't think the data presented here is useless, but having stained (recently living) samples to compare with the dead assemblages at the surface would have provided a much more powerful interpretation.

AC: We agree. This study would be stronger if the sediments had been stained at the time of collection.

-Line 132: I would not qualify 1-2cm as surface. It is shallow infaunal.

AC: We utilized both the 0-1 and 1-2 cm fraction as the "surface" for several reasons. These are discussed in section 3.2 Lines 168-172. We completed a comparison of the 0-1cm and 1-2cm "We do not identify any significant relationship between relative abundance of a species and depth interval (in all cases p>0.05 or r2 is <0.001). Foraminiferal abundance is low (<100 individuals) in some of the samples from 0-1 cm. Thus, in order to utilize sufficient numbers of individuals and because there were no significant differences in abundance of species between 0-1 cm and 1-2 cm, for the rest of the discussion we refer to the 0-2 cm fraction as the core top material." (Lines 168-172).

-Lines 133-134: ". . . were examined at 1 cm intervals for cores MV1217-2-3 (528 m), MV1217-1-3 (800 m) and MV1217-4-3 (1175 m)." Ending at what depth in the cores?

AC: We will add in the depth (in cm) that we analyzed from each core.

-Line 156: no units on salinity. Remove "with a total" replace "range of" with "ranging from"

AC: We chose to report salinity in "practical salinity units." We will make the additional language change.

-Line 157: (DO) once you have introduced an abbreviation you do not have to reference it again and you can then use the shorthand in the text. I suggest you do a search of the manuscript and identify duplicates of instances such as this. I suggest a rewriting of this section for clarity. Either report based on water column depth or minimums. As is, it is confusing. –Rephrasing

suggestion: —"Water column dissolved oxygen (DO) concentration documents a low oxygen zone, with a minimum occurring at 700 m water depth (0.26 ml/L; Figure 2, Table 1), compared to 1.54 ml/L at 300 m and 0.58 ml/L at 1175 m. Minimum pH is documented at 700 m (7.55) and is higher above (300 m, 7.65) and below (1175 m, 7.59) the intermediate depth low pH zone (Figure 2, Table 1)." —Water column DO measurements indicate areas of low oxygen availability from 300 m (1.54 ml/L) to 1175 m (0.58 ml/L) with lowest oxygen availability at 700 m (0.26 ml/L). Although not greatly variable, pH minima also occur at 700 m (7.55) and is higher at 300 m (7.65) and 1175 m, (7.59). In this section, why not report as hypoxic, anoxic, as outlined by Bernhard et al?

AC: We will improve upon the introduction of abbreviations earlier on in the text as suggested by both reviewers. We chose not to report oxygen as hypoxic, anoxic, as outlined by Bernhard et al because we later argue that in the environment we studied, these categories of foraminifera by oxygenation do not accurately reflect the foraminifera we sampled. Thus, it is more useful to describe each site using the measured dissolved oxygen.

-Line 175: "Due to their propensity for degradation and to remain consistent with other regional studies, we exclude agglutinated taxa and all values are reported as percent of total calcareous taxa for the remainder of the text." I don't disagree with this statement, but I think that ignoring the agglutinates is a mistake we will regret in the future. Why not keep the data (they were a significant portion of the population) and just report Calcareous populations in the graphs etc?

AC: In order to remain consistent with other studies (Kaiho 1994, Balestra et al 2017, Myhre et al 2014) that focus only on calcareous taxa, we exclude the agglutinated taxa from further discussion.

-Lines 180-181: "These dominant taxa make up more than 84% of all foraminifera counted across all core top samples." Does this include the agglutinates or just 84% of the calcareous foraminiferal population?

AC: This only takes into account calcareous taxa. We will make this clear in the text.

-Lines 200- Section 3.3: Based on the results of this section why are you concluding that it is better to look at the >150micron size fraction as you state in the conclusion?

AC: We conclude that it is effective to look at the > 150 micron size fraction because we find that trends across depth are similar between the complete (>63 μm) and large size fraction (>150 μm) or are more pronounced in the >150 μm size fraction compared to the 63-150 μm size fraction. Further, we utilized this size fraction to remain consistent with other regional studies. These trends are further elucidated by the corrected Figure 3 we have uploaded as part of this response.

-Line 217: When you say above and below the OMZ do you mean on the seafloor? Not in the water column? Please clarify. G. subglobosa as oxic indicator- This assertion needs a discussion or reference. Skipped to discussion

AC: When we refer to above and below the OMZ we are referring to within the water column, not above/below the sediment surface. We are indicating that *G. subglobosa* is more abundant at the sites above (300m) and below (1175m) the OMZ. As such, it is from our own data that we identify this species as an oxic indicator, rather than from previous work. But, additionally, we can add citations showing this species as an oxic indicator (Kaiho 1999).

-Line 295-296: I'm not sure this was the findings of Venturelli et al., 2018. Which section of the paper are you referring to here? The focus of this paper was on comparing sediment grain size, oxygen, and foraminiferal abundances in the Southern California Bight. It was not an OMZ survey and therefore I am not sure they proposed seeing variations of foraminiferal abundances in size fractions "within" the OMZ vs outside the OMZ. If you just mean to say that 63-150 micron foraminifera were more abundant than >150 micron reported by Venturelli et al., 2018 I think this is true, but how different were the populations and would grain size difference influence these abundances?

AC: We will remove the citation for Venturelli et al 2018 and instead incorporate a more thorough discussion of the relationship between size and oxygenation in benthic foraminifera following the discussion in Keating-Bitonti and Payne 2016. See comment below on adding more thorough literature review and discussion of size.

-Lines 355-338: "The presence of oxic indicator species across all water depths may provide evidence for periodic flushing of high oxygen water or a selection for species that can tolerate a range of environmental conditions rather than a specific threshold of oxygenation." It may also be that the physiological tolerances of indicator species are not fully understood. They may be able to tolerate lower oxygen than previously thought provided another incentive such as substrate (e.g., Venturelli et al., 2018, Burkett et al., 2016), and or they may be able to tolerate short periods of low oxygen or inhospitable conditions (Bernhard et al., 2010).

AC: This is an interesting point. We will add further discussion and incorporate the references cited by Referee 1 (Venturelli et al., 2018, Burkett et al., 2016, Bernhard et al., 2010).

-Line 270: What do you mean by equal in magnitude? Can you elaborate on that? Perhaps by giving total abundance or percent abundance examples?

AC: In multivariate space, the difference between sites across space is greater than within any one site through time. We will clarify this in the text.

-Lines 300-302: Where can the readers see this stated relationship? "...and the >150 μm size fraction or 2) trends in the >150 μm size fraction are more pronounced than in the complete assemblage"

AC: This relationship can be seen in the updated Figure 3; due to the graphical error in Figure 3, this was not clear in the original submission.

-Lines 300-310: YES! I totally agree with some of the things you are saying in this section, but you have to do a complete discussion of the literature. I would suggest splitting it up into two

sections 1) examination of <150 micron foraminifera in low oxygen/high carbon environments. Do an overview of who proposed it, what results have been seen, and how it relates to your data. The second literature comparison should be 2) foraminifera test size in comparison with oxygen. Here would be a great chance to discuss Keating-Bitonti and Payne, 2016 in greater detail. Again, do an overview of who proposed it, what results have been seen, and how it relates to your data.

AC: We will incorporate a more thorough literature review of the distribution of foraminifera in low oxygen/high carbon environments and of foraminiferal test size in relation to oxygen.

 -Line 447: by "core" do you mean center? Is there another word you can use here so as not to confuse it with sediment cores?

AC: In this sentence we are referring to the center of the OMZ, we will change the language so that it is not confused with sediment cores.

-Data availability: is the section with data files raw counts? Or is this standardized per volume? See discussion in the "scientific questions/issues" section.

AC: We have addressed this question in a comment above. The raw data are given as raw abundance by volume of original sediment core.

-Figure 3: "General observations discussed in the text are noted here, e.g., species that increase in abundance in the OMZ, appear associated with the "edge" of the OMZ, etc. Note difference in x-axis in B. argentea plot" is not very useful information for a figure caption. Please describe the structure of the graph and summarize what you observed or reference to the section of the paper where it is discussed. Please also clearly state in this figure that "core tops" are the 0-2cm intervals. -Based on the OMZ bounds of Figure 2 it seems the majority of the foraminiferal abundance plots resides in what you have defined at the OMZ. So how can you see increases if you have no "background" to compare it to? Please clarify. The key should be bounded by a box to better identify it.

AC: We will make the suggested graphical improvements. Further, by updating Figure 3 to eliminate the graphical error, we will fix these issues. We will more heavily cite Figure 3 in the text so that it is clear when interpretations are based on this data.

**Response to Referee Comment 2**

RC2: Referee comment 2, in gray
AC: Author comment, in black

AC: We thank Referee 2 for their helpful comments and suggestions to improve the manuscript. We have addressed each major comment below. We will incorporate all "notes for specific parts" listed by Referee 2 unless a comment is made below.

RC2: This manuscript entitled "Southern California margin benthic foraminiferal assemblages across a modern environmental gradient record recent centennial-scale changes in oxygen minimum zone" by Palmer et al. presents a valuable dataset from a gradient of one of the most prominent OMZs in the world. It presents the calcareous benthic foraminiferal assemblages focusing in size fractions from core tops along a depth transect. Later the authors investigate these assemblages in short cores in order to investigate the recent history of the OMZ and the benthic foraminiferal assemblages. The information provided here is an important input for the ongoing investigations regarding the relationship between OMZs, ecosystem and climate. Overall, I found some major details missing in this study and I believe it can be improved significantly.

1. This study is based on benthic foraminifera taxonomy work which should include references to species nomenclature also preferably a plate showing the major species mentioned. In case it is not possible to provide figures, there should be a section where list of species observed is given with references used for identification. For example: Bolivina spissa = Bolivina subadvaena Cushman var. spissa Cushman 1926a. [Figures 10.7, 11.4]. This is essential for taxonomy based papers where the reader will be able to evaluate the information provided. The number of publications without any reference material is increasing and this leads the misinterpretations regarding the foraminifera research. The authors mentions their concerns in the discussion therefor I highly encourage this MS to have section dedicated to nomenclature.

AC: We will update the list of species observed with the references used for identification. We will also add a plate with images of the 6 species that are discussed in depth in the paper.

RC2:  2. Introduction and discussion should be improved in terms of using literature and previous work from different OMZs. For instance there are significant amount of work from the Peruvian and Arabian OMZs focusing on similar oxygen gradient and benthic foraminiferal assemblages. These studies should be included in terms of benthic foraminifera habitat in relation with oxygen and nitrate availability etc. This will improve the MS significantly. It is a pity that the species are not stained limiting the comparison with previous studies, and yet I believe the information presented here is really valuable.

AC: We will improve our literature review and include more literature from OMZs outside of the North Pacific (Erdem et al 2019, Caulle et al 2014, Enge et al 2016, Mallon et al 2011, Mazumder et al 2014). Further, per the comments of referee 1, we will incorporate a more thorough literature review on the distribution of foraminifera in low oxygen/high carbon environments and of foraminiferal test size in relation to oxygen.

RC2: 3. Presentation of environmental parameters is confusing. Are these values from measurements of bottom water conditions? deepest depth of CTD? Figure 2 should be improved accordingly where stations can be shown.

AC: The environmental parameters listed are from measurements of bottom water conditions taken at the same time as sediment core sampling.

RC2:  4. Definition of an OMZ: please introduce OMZ already in introduction. This MS uses certain terms such as OMZ edge, suboxia, hypoxic boundary and so on; to eliminate the confusion, edge or boundary of an OMZ considered here should be introduced as early in the text as possible.

AC: We will incorporate an introduction to nomenclature in the introduction that will make the entire paper more readable and streamlined. We received similar comments from referee #1 and we will address them both.

Notes for specific parts:

page 2, line 60: please check Tetard et al., 2017.

AC: We will add the suggested reference to this section of the paper.

paragraph starting with line 67: this section could be improved significantly by including previous observations from other OMZs which should be included in discussion where Bolivinids and nitrate availability are discussed.

AC: Based upon this suggestion and that of Reviewer 1, additional observations from OMZs will be added.

page 3, line 107: should be "dissolved oxygen concentration"

AC: We will incorporate the suggested change.

page 4, section under 2.3. needs to be rewritten considering the steps taken to reach the species counts. first, material sieved, dried and count in different fractions. Which references were considered for 300 and 150 specimens? Why did the authors decide these numbers? I am not an expert for statistical methods but what is the reason behind using dbRDA but not component analysis (CCA?) to test the relationship between foraminifera and environmental parameters?

AC: We chose to use a dbRDA instead of a CCA because drRDA allows for the use of Bray-Curtis dissimilarity rather than Euclidean distance in quantifying differences between groups and is able to integrate data from drivers (environmental factors) as well as assemblages.

line 145: "...mixed planktonic foraminifera species" please remove bulk

AC: We will incorporate the suggested change.

section 3.1.: this section is confusing. please be clear with what is presented here. I assume these are the deepest points CTD measured? is there any pore water measurements or are these only water column? and salinity should be included as well in the table.

AC: The data shown here are the deepest CTD points measured, not pore water measurements. We will add salinity to the table.

Line 205: is ANOVA introduced already in methods?

AC: ANOVA was not introduced in the methods. We will add ANOVA to section 2.3 on foraminifera assemblages.

line 222: the term edge dominant.. what does this actually mean? According to which previous work edge of the OMZ is considered?

AC: Incorporating the referee's earlier comment about adding some clarifying language and nomenclature to the introduction would be helpful here as well. In this case, we are referring to a species (*B. argentea*) that is most abundant at 528m water depth, near the upper margin of the modern OMZ. A previous study that we cite in the paper, Mullins et al 1985, also finds a high abundance of some species of benthic foraminifera at similar depths and attributes this to the higher concentrations of biologically available nitrate and nitrifying bacteria at the edges of the OMZ as compared to the center. We will cite this paper here to show support.

line 224: sentence starting with "in some taxa,.." needs rephrasing. paragraph starting from line 268: this could be written much simpler, I am not sure I understand the information provided here.

AC: We will improve these lines for clarity and simplicity.

page 9 first paragraph: we know today oxygen limited high organic matter input regions are characterised by high population low diversity of benthic foraminifera. it is interesting to see this is not the case at these sites. Nevertheless, I am not fully convinced the evidence provided in this study is enough to come to this conclusion. What do the authors think, if stained species were considered only the results would show any difference or not?

AC: The referee poses an important question here. Unfortunately, there are not many studies in this region comparing live/dead assemblages so it is difficult to speculate on this point. Further, some studies (Bernhard et al 2006) have shown that Rose-Bengal staining does not accurately capture the live foraminiferal fauna. Other studies that have examined stained vs. unstained including Jorissen and Wittling 1999 document that some epifaunal and superficial infaunal species may reproduce opportunistically and thus have higher seasonal variability in comparison to infaunal species which they document as having stable densities through time. Thus, the assemblage we quantified may oversample epifaunal taxa relative to infaunal taxa in comparison to what may have been found if the samples were stained.

Line 301: what does "...size fraction or 2" mean? paragraph starting at line 312: for such a discussion based on specific species, authors should provide a reference list including species names as mentioned earlier.

AC: We will add a species list and reference list for the species that we discuss in detail. "Size fraction or 2)" was part of a larger list, we will remove the 1) and 2) for clarity.

line 341: this is the first time specific oxygen concentration and terminology is given. This should come earlier.

AC: This will appear earlier, starting in the introduction.

paragraph starting at line 365: discussion on Bolivinids: there are so many studies on bolivinids at similar setups, those should be mentioned and discussed here. Some examples are: Mallon et al., 2012; Cardich et al., (several papers), Glock et al., (several papers), Caulle et al., 2014; Jannink et al., 1998. the list goes on.

AC: We will add these references.

line 387: Please rephrase the last sentence.

AC: We will incorporate the suggested change.

paragraph starting with line 424: it would be nice to compare results with previous observations from the region.

AC: We will incorporate the suggested change.

comment on data availability: will it be open access upon publication?

AC: Yes, the data are already available and open access on NOAA Paleoclimate Database.

Figure 1: please give more information in the figure caption including which sites have what kind of results in the text. what are the depths of these sites?

AC: We will incorporate the suggested change. The depths are 300, 528, 700, 800, and 1200 m, this can be added to Figure 1.

Figure 2: water depth on y axis? station names could be implemented.

AC: Water depth is on the y axis in this plot. We will clarify this in the figure.

Figure 3 caption: "General observations .... " this is not needed here. Figure should be cited in the text more often.

AC: We will cite Figure 3 in the text to improve clarification.  Referee #1 also suggested the same change. Further, we have updated Figure 3, this will add clarification to this point.

Figure 4: y axis please mention Rel. Abundance (%) instead.

AC: We will incorporate the suggested change.

Table 1: please add salinity and the captions should be more informative including where this information comes from. CTD? porewater? what is TOM?

AC: We will incorporate the suggested change. This data comes from a CTD (as answered above) of bottom water, not porewater. TOM is total organic matter, methods for this are listed in the methods section of the paper.

Table 2 caption: mixed planktonic foraminifera species. please remove bulk.

AC: We will incorporate the suggested change.

**Southern California margin benthic foraminiferal assemblages record recent centennial-scale changes in oxygen minimum zone**

Hannah M. Palmer[1,2], Tessa M. Hill[1,2], Peter D. Roopnarine[3], Sarah E. Myhre[4], Katherine R. Reyes[1,5], and Jonas T. Donnenfield[1,6]

[1]Bodega Marine Laboratory, University of California, Davis, California, USA
[2]Department of Earth and Planetary Sciences, University of California, Davis, California, USA
[3]Department of Invertebrate Zoology and Geology, California Academy of Sciences, San Francisco, California, USA
[4]School of Oceanography, University of Washington, Seattle, Washington, USA
[5]Department of Natural Sciences and Mathematics, Dominican University of California, San Rafael, California, USA
[6]Geology Department, Carleton College, Northfield, Minnesota, USA

*Correspondence to:* Hannah M. Palmer (hmpalmer@ucdavis.edu)

**Abstract.** Microfossil assemblages provide valuable records to investigate variability in continental margin biogeochemical cycles, including dynamics of the oxygen minimum zone (OMZ). Analyses of modern assemblages across environmental gradients are necessary to understand relationships between assemblage characteristics and environmental factors. Five cores were analyzed from the San Diego margin (32°42'00"N, 117°30'00"W, 300-1175 m water depth) for core top benthic foraminiferal assemblages to understand relationships between community assemblages and spatial hydrographic gradients and for down core benthic foraminiferal assemblages to identify changes in the oxygen minimum zone through time. Comparisons of benthic foraminiferal assemblages from two size fractions (63-150 and >150 μm) exhibit similar trends across the spatial/environmental gradient, or in some cases exhibit more pronounced spatial trends in the >150 μm fraction. A range of species diversity exists within the modern OMZ (1.910-2.586 H, Shannon Index), suggesting that diversity is not driven by oxygenation alone. We identify two hypoxic associated species (*B. spissa* and *U. peregrina*), one oxic associated species (*G. subglobosa*) and one OMZ edge-associated species (*B. argentea*). Down core analysis of indicator species reveal variability in the upper margin of the OMZ (528 m water depth) while the core of the OMZ (800 m) and below the OMZ (1175 m) remained stable in the last 1.5 ka. We document expansion of the upper margin of the OMZ beginning 400 ybp on the San Diego margin that is synchronous with other regional records of oxygenation.

**1 Introduction**

Ocean oxygenation is declining globally; rising ocean temperatures decrease oxygen solubility at the sea surface and increased stratification inhibits ventilation, leading to decreased oxygen at depth (Breitburg et al., 2018; Levin et al., 2009; Stramma et al., 2010). Expansions of oxygen minimum zones (OMZ) have already been documented and further expansions are predicted (Bograd et al., 2008; Schmidtko et al., 2017; Stramma et al., 2010). Within the California Current system, a decline in dissolved oxygen (DO) concentration, shoaling of the hypoxic boundary, and decreased pH have been documented (Bakun, 2017; Bograd et al., 2008). The intensity and geographic extent of the CA margin oxygen minimum zone has oscillated in response to past changes in climate and ocean temperatures on millennial timescales, weakening during cool periods and strengthening during warm periods (Cannariato and Kennett, 1999; Jaccard et al., 2014; Moffitt et al., 2014, 2015a; Ohkushi et al., 2013). Determination of timing and drivers of past expansions and contractions of oxygen minimum zones is critical to developing accurate predictions of future change (Jaccard et al., 2014).

Continental margin biogeochemical dynamics structure shelf ecosystems across space and time (Levin et al., 2009; Levin and Dayton, 2009). In particular, oxygenation is a key determinant of benthic zonation; seafloor ecosystems are subject to major turnover in response to relatively minor inferred changes in oxygenation (Levin, 2003; Levin and Dayton, 2009; Moffitt et al., 2015b). Areas of low oxygen availability typically contain low abundance and diversity of organisms (Levin, 2003; Levin and Dayton, 2009). However, several species of benthic foraminifera are adapted to survive in low-oxygen conditions and are thus present, and often abundant, in such environments (Bernhard and Gupta, 1999; Gooday et al., 2000; Kaiho, 1994, 1999; Keating-Bitonti and Payne, 2016).

**1.1 Benthic foraminifera record changes in coastal margin biogeochemistry**

Microfossil records from the Southern California Borderlands are a critical tool for understanding changes in productivity (Cannariato and Kennett, 1999; Emmer and Thunell, 2000; Stott et al., 2000), orbital and millennial scale climate changes (Hendy, 2010; Hendy and Kennett, 2000; Taylor et al., 2015), and climate change through the Holocene (Balmaki et al., 2019; Fisler and Hendy, 2008; Friddell et al., 2003; Roark et al., 2003). Benthic foraminiferal assemblages are widely used as a proxy for changes in oxygenation through time (Balestra et al., 2018; Bernhard et al., 1997; Bernhard and Gupta, 1999; Cannariato and Kennett, 1999; Gooday, 2003; Jorissen et al., 2007; Moffitt et al., 2014; Ohkushi et al., 2013; Shibahara et al., 2007; Tetard et al., 2017). Previous work (through analysis of benthic foraminifera along environmental depth gradients and in laboratory culturing studies) documented relationships between benthic foraminiferal taxa and water depth, oxygen concentration, sediment substrate, position in the sediment matrix, nitrate availability, and organic matter availability (Bernhard et al., 1997; Bernhard and Bowser, 1999; Bernhard and Gupta, 1999; Caulle et al., 2014; Douglas, 1981; Douglas and Heitman, 1979; Erdem et al., 2019; Jorissen et al., 2007; Kaiho, 1994, 1999; Mallon et al., 2012; Mazumder and Nigam, 2014; Mullins et al., 1985).

Moved down [1]: 1.1 Microfossil record of coastal margin biogeochemical change ¶

Commented [h7]: RC1: -Line 42: "declines in dissolved oxygen," awkward. Please rephrase.

Moved (insertion) [3]

Commented [h8]: RC1: -Line 43: determination of

Commented [h9]: RC1: -Line 49: "Low-oxygen zones typically contain both low abundance and low diversity of" change to "Areas of low oxygen availability typically contain low abundance and diversity of"

Moved (insertion) [1]

Commented [h11]: RC2: paragraph starting with line 67: this section could be improved significantly by including previous observations from other OMZs which should be included in discussion where Bolivinids and nitrate availability are discussed.
RC2: RC2: 2. Introduction and discussion should be improved in terms of using literature and previous work from different OMZs. For instance there are significant amount of work from the Peruvian and Arabian OMZs focusing on similar oxygen gradient and benthic foraminiferal assemblages.

Field Code Changed

Commented [h12]: RC2: page 2, line 60: please check Tetard et al., 2017.

Commented [h13]: Added Tetard et al 2017

Commented [h14]: Added Caulle et al 2014, Erdem et al 2019, Mallon et al 2012, Mazumder and Nigam 2014

Moved (insertion) [2]

Deleted: Relationships between benthic assemblages and modern environmental conditions were previously developed through analysis of benthic foraminifera along environmental depth gradients and in laboratory culturing [1]

Generally, low oxygen environments contain high abundance and low diversity of benthic foraminifera; in these settings, infaunal, elongate, thin-walled species with high porosity dominate over porcelaneous and epifaunal taxa (Bernhard et al., 1997; Douglas, 1981; Jorissen et al., 1995, 2007; Kaiho, 1994, 1999; Mazumder and Nigam, 2014). Further work has explored the relationship between foraminiferal size and oxygenation; generally volume to surface area ratios of foraminiferal tests are positively correlated with dissolved oxygen, yet studies of individual taxa on the Southern California margin do not consistently show this relationship (Keating-Bitonti and Payne, 2016; Keating-Bitonti and Payne, 2017; Rathburn et al., 2018). Often, individual taxa of foraminifera are classified into groups based on oxygen affinity or individually identified as oxygen indicator taxa (Jorissen et al., 1995; Kaiho, 1999; Moffitt et al., 2014). In particular, bolivinid taxa are noted as low oxygen indicator taxa (Cardich et al., 2015; Caulle et al., 2014; Mallon et al., 2012; Mullins et al., 1985). However, these relationships between foraminiferal assemblages and environmental metrics are regionally defined and cannot be applied globally; regional calibrations of the benthic foraminifera oxygen proxy are required for accurate paleoceanographic analyses (Bernhard et al., 1997; Caulle et al., 2014; Kaiho, 1999; Mallon et al., 2012; Mazumder and Nigam, 2014). Similarly, the classification of oxygenation levels varies among paleoceanographic studies (Balestra et al., 2018; Kaiho, 1994; Moffitt et al., 2015a). This study uses the following classification: oxic ($[O_2]$ >1.5 ml/L), intermediate hypoxia/suboxic ($[O_2]$ 1.5-0.5 ml/L), and severe hypoxia/dysoxic ($[O_2]$ <0.5 ml/L) (Moffitt et al., 2015a). Although oxygenation is a dominant driver of ecosystem zonation in marginal environments, sediment substrate, organic matter availability, and nitrate availability also play important roles in structuring benthic foraminiferal assemblages. Further analysis of the interacting environmental factors along depth/environmental gradients is needed (Jorissen et al., 1995, 2007; Mullins et al., 1985; Venturelli et al., 2018).

Previous studies of oxygenation change over time from the Southern California margin focus largely on the Santa Barbara and Santa Monica Basins, due to their high sedimentation rates and regular laminations (Balestra et al., 2018; Cannariato and Kennett, 1999; Christensen et al., 1994; Kaiho, 1999; Moffitt et al., 2014; Schimmelmann et al., 2013). Significantly fewer studies investigate sediments outside of those basin environments (Erdem et al., 2019; Mallon et al., 2012; McGann, 2002). Further analysis is therefore needed to constrain relationships between benthic foraminifera and environmental conditions in the open continental margin where biological and chemical gradients are more variable, and to identify decadal to centennial changes in oxygen minimum zone dynamics.

**1.2 Regional Setting**

The California margin is a well-studied system characterized by southward flow of the California Current, a strong seasonal upwelling regime bringing cold, nutrient rich waters to the surface, high coastal productivity, and a large oxygen minimum zone occurring at intermediate water depths (Checkley and Barth, 2009). The San Diego margin is dominated by two surface currents: the southward flowing California Current and the seasonal, northward flowing surface Davidson Current (Checkley and Barth, 2009).

**Commented [h15]:** RC2: RC2: 2. Introduction and discussion should be improved in terms of using literature and previous work from different OMZs. For instance there are significant amount of work from the Peruvian and Arabian OMZs focusing on similar oxygen gradient and benthic foraminiferal assemblages. These studies should be included in terms of benthic foraminifera habitat in relation with oxygen and nitrate availability etc. This will improve the MS significantly.

**Commented [h16]:** RC2: line 341: this is the first time specific oxygen concentration and terminology is given. This should come earlier.
RC2: RC2: 4. Definition of an OMZ: please introduce OMZ already in introduction. This MS uses certain terms such as OMZ edge, suboxia, hypoxic boundary and so on; to eliminate the confusion, edge or boundary of an OMZ considered here should be introduced as early in the text as possible.

**Commented [h17]:** RC1: RC1: 4. A more complete literature review is needed in the discussion section in order to support the findings of the authors
RC2: paragraph starting with line 67: this section could be improved significantly by including previous observations from other OMZs which should be included in discussion where Bolivinids and nitrate availability are discussed.
RC2: 2. Introduction and discussion should be improved in terms of using literature and previous work from different OMZs. For instance there are significant amount of work from the Peruvian and Arabian OMZs focusing on similar oxygen gradient and benthic foraminiferal assemblages. These studies should be included in terms of benthic foraminifera habitat in relation with oxygen and nitrate ... [2]

**Moved up [2]:** Relationships between benthic assemblages and modern environmental conditions were previously developed through analysis of benthic foraminifera along

**Commented [h18]:** RC1: -Line 73: "2013), yet significantly fewer" remove ", yet". Begin the sentence with "Significantly. . ." Do you have any examples to cite e.g.,? -

**Commented [h20]:** RC1: Line 75: "biological and chemical gradients are more extensively distributed" or more extensively variable?

**Moved up [3]:** Previous studies utilizing benthic foraminiferal assemblages as a proxy for oxygenation demonstrated that the intensity and geographic extent of the

**Commented [h21]:** RC1: -Line 82: remove "current, the" or something I just don't like how it is phrased.

[revised manuscript text omitted]

**Commented [h28]:** RC1: -Lines 123-128: The word sample is used too many times (5 in one sentence). Try to replace with another word to avoid monotony

**Commented [h29]:** RC1: -Line 130: "present in the sample." I'm assuming it is in the entire sample? If so, add "entire" before "sample"

**Commented [h30]:** RC2: page 4, section under 2.3. needs to be rewritten considering the steps taken to reach the species counts. first, material sieved, dried and count in different fractions. Which references were considered for 300 and 150 specimens? Why did the authors decide these numbers?

**Commented [h31]:** RC2: 1. How are raw abundance counts standardized? Are all abundances reported as raw numbers? If so, how do you account for inconsistencies in the amount of sediment examined in each interval? The data should be standardized to #/50cc or #/gram.

**Commented [h32]:** RC1: -Lines 132-134: These two sentences should be combined with the previous paragraph as 2 sentences is not a paragraph on its own.

**Commented [h33]:** RC1: -Line 132: I would not qualify 1-2cm as surface. It is shallow infaunal.

**Commented [h34]:** RC1: -Lines 133-134: ". . . were examined at 1 cm intervals for cores MV1217-2-3 (528 m), MV1217-1-3 (800 m) and MV1217-4-3 (1175 m)." Ending at what depth in the cores?

**Commented [h35]:** RC1: -Table 2: Contains a core not "used in this study" MV1217-4-1. If it is to be included in this table it would be helpful to also include depth and lat-long so readers have an idea of why you are including this in the manuscript.

**Commented [h36]:** RC2: Line 205: is ANOVA introduced already in methods? ... [5]

**Commented [h37]:** RC2: line 145: "...mixed planktonic foraminifera species" please remove bulk

5568 years. The Calib7.1 calibration program was used to calibrate ages using a reservoir age of 220.0 +/- 40.0 (Ingram and Southon, 1996; Stuiver and Polach, 1977).

280

**3.0 Results**

**3.1 Vertical profiles, sediment characterization of San Diego margin**

Across the depth profile, bottom water temperature decreased steadily with depth (300 m to 1175 m) ranging from 8.6°C (300 m) to 3.8°C (1175 m) (Figure 2, Table 1). Salinity (not plotted here) had a mean of 34.4 psu ranging from 34.1 to 34.5 psu. Water column DO measurements collected directly above each coring site show oxic conditions at 300 m (1.54 ml/L) above the OMZ, intermediate hypoxia at 1175 m (0.58 ml/L) below the OMZ, and severe hypoxia at 528 m (0.35 ml/L) at the upper edge of the OMZ, and within the OMZ at 700 m (0.26 ml/L) and 800 m (0.29 ml/L). Although not greatly variable, a pH minimum occurs at 700 m (7.55) and is higher at 300 m (7.65) and 1175 m, (7.59) (Figure 2, Table 1). Total organic matter increased with depth (6.8-14.7% AFDW). These results are consistent with previous analyses of the California margin OMZ/CMZ (Helly and Levin, 2004).

290

**3.2 Benthic foraminiferal assemblage across modern environmental gradient**

Relative abundance of benthic foraminifera was quantified for all sites in the 0-1 and 1-2 cm intervals. We compared the 0-1 cm interval to the 1-2 cm interval to assess if depth habitat of any species determined their relative abundance in the core top assemblage. Specimens were not Rose Bengal stained, thus their presence in any interval does not indicate that they were living at the time of collection. We do not identify any significant relationship between relative abundance of a species and depth interval (ANOVA: in all cases p>0.05 or $r^2$ is <0.001). Foraminiferal abundance is low (<100 individuals) in some of the samples from 0-1 cm. Thus, in order to utilize sufficient numbers of individuals and because there were no significant differences in abundance of species between 0-1 cm and 1-2 cm, for the rest of the discussion we refer to the 0-2 cm fraction as the core top material (Figure 2 and 3). Calcareous taxa dominated the assemblage at every site; agglutinated foraminifera made up 0 (300 m) to 21% (700 m) of the assemblage. Due to their propensity for degradation and to remain consistent with other regional studies, we exclude agglutinated taxa and all values are reported as percent of total calcareous taxa for the remainder of the text (Balestra et al., 2018; Kaiho, 1994; Moffitt et al., 2014; Venturelli et al., 2018).

Total abundance of foraminifera decreases with depth (Figure 2). Core top assemblages were dominated by *Bolivina argentea, Uvigerina peregrina, Globocassidulina subglobosa, Epistominella* sp*., Cassidulina carinata,* and *Bolivina spissa,* in order of decreasing abundance (see supplemental data for images of dominant taxa). These dominant taxa make up 80% of all calcareous foraminifera counted across all core top samples. All other species each represent less than four percent of the total assemblage across all core tops. The following taxa are found at all five water depths: *Bolivina argentea, Bolivina spissa, Bulimina* sp., *Cibicidoides* sp., *Epistominella* sp*., Globobulimina pacifica, Globocassidulina subglobosa, Globobulimina ovata, Nonionella stella, Quinqueloculina* sp*.* and *Uvigerina peregrina.*

Field Code Changed

Commented [h38]: RC1: -Line 156: no units on salinity. Remove "with a total" replace "range of" with "ranging from"

AC: We chose to report salinity in "practical salinity units." We will make the additional language change.

Commented [h39]: Rc2: RC2: 4. Definition of an OMZ: please introduce OMZ already in introduction. This MS uses certain terms such as OMZ edge, suboxia, hypoxic boundary and so on; to eliminate the confusion, edge or boundary of an OMZ considered here should be introduced as early in the text as possible.

Commented [h40]: RC1: -Line 157: (DO) once you have introduced an abbreviation you do not have to reference it again and you can then use the shorthand in the text. I suggest you do a search of the manuscript and identify duplicates of instances such as this. I suggest a rewriting of this section for clarity. Either report based on water column depth or minimums. As is, it is confusing. –Rephrasing suggestion: —"Water column dissolved oxygen (DO) concentration documents a low oxygen zone, with a minimum occurring at 700 m water depth (0.26 ml/L; Figure 2, Table 1), compared to 1.54 ml/L at 300 m and 0.58 ml/L [6]

Commented [h41]: RC2: section 3.1.: this section is confusing. please be clear with what is presented here. I [7]

Commented [h42]: RC1: ." This needs to be identified in Figure 3.

Commented [h43]: RC1: -Line 173: Remove "The site" and begin the sentence with "At". Change "at the core of the [9]

Commented [h44]: RC1: 3. A better discussion of how agglutinated foraminifera are reported, or not, in the total [11]

Commented [h45]: RC1: -Lines 180-181: "These dominant taxa make up more than 84% of all foraminifera counted [12]

Commented [h46]: RC1: -Line 181: "All other species each account for <4% of total assemblage across all core tops [13]

Commented [h47]: RC1: -Line 179: "sp1" can this just be written as "sp."?

First, we report the benthic foraminiferal assemblage from the >63 µm size fraction; we then report on a comparison between the 63 - 150 µm and >150 µm size fractions. The assemblage at 300 m is dominated by *G. subglobosa* (28%), *B. argentea* (25%), *U. peregrina* (10%), *Epistominella* sp. (8%), and *Bolivina spissa* (6%); species richness is 24 and diversity (H) is 2.133. The assemblage at 528 m is dominated by *B. argentea* (37%), *U. peregrina* (23%), *Epistominella* sp. (15%), *C. carinata* (6%) and *G. subglobosa* (5%); species richness is 25 and diversity (H) is 1.910. The assemblage at 700 m is dominated by *Epistominella* sp. (29%), *C. carinata* (15%), *U. peregrina* (13%), *G. subglobosa* (11%), *B. argentea* (10%), and *B. spissa* (7%); species richness is 23 and diversity (H) is 2.249. The assemblage at 800 m is dominated by *B. spissa* (16%), *U. peregrina* (16%), *Epistominella* sp. (13%), *C. carinata* (10%), *Cibicidoides* sp. (10%), and *Globobulimina ovata* (13%); species richness is 25 and diversity (H) is 2.586. The assemblage at 1175 m is dominated by *C. carinata* (25%), *Epistominella* sp. (20%), *G. subglobosa* (8%), *B. spissa* (6%); species richness is 25 and diversity (H) is 2.389 (Figure 2).

**3.3 Comparison of benthic foraminifera in two size fractions**

Comparison of foraminiferal abundance between the 63-150 µm and >150 µm shows higher abundance in the small fraction at 300 m, 700 m and 1175 m, and higher abundance in the large size fraction at 528 m and 800 m. Several taxa are found in both size fractions at all five water depths: *Globocassidulina subglobosa* and *Epistominella* sp. Four species have significantly different relative abundances between size classes; three are more likely to be found in the 63-150 µm (*Cassidulina carinata, Epistominella* sp. and *G. subglobosa*) and one species (*U. peregrina* ) is significantly more likely to be found in the 150 µm size fraction (ANOVA, p<0.05 for all, Figure 3).

In the >150 µm size fraction, species diversity (H) ranges from 1.316 - 2.700; minimum diversity (H) is found at 528 m (1.316) and maximum diversity (H) is found at 1175 m (2.700). In comparison, in the 63-150 µm size fraction, species diversity (H) ranges from 1.710 - 2.042; minimum diversity (H) is found at 700 m (1.710) and maximum diversity (H) is found at 800 m (2.042) (Figure 2). Species diversity is greater in the >150 µm size fraction relative to the 63-150 µm size fraction at all sites except the site at 528 m (Figure 2).

When we consider the complete assemblage (>63µm) we can classify the most abundant species into four groups based on their trends relative to the oxygen minimum zone (Figure 3). Two species are more abundant within the OMZ: *B. spissa,* and *U. peregrina*; we identify these species as dysoxic indicator species. One species is less abundant within the OMZ relative to sites outside of the range of the OMZ: *G. subglobosa*; we identify this species as an oxic indicator species (Kaiho, 1999). Two species increase in abundance with water depth: *C. carinata* and *Epistominella* sp. One species is most abundant near the uppermost edge of the OMZ: *B. argentea*; this species may be edge-associated (Mullins et al., 1985). Importantly, when we consider only the >150 µm size fraction, we observe the same trends: high abundance in OMZ (*B. spissa, U. peregrina*), low abundance in OMZ (*G. subglobosa*), increased abundance with depth (*C. carinata* and *Epistominella* sp.), and OMZ edge-associated (*B. argentea*) (Figure 3). Generally, we find that trends across depth are similar between the complete (>63 µm) and large size fraction (>150 µm) or are more pronounced in the >150 µm size fraction compared to the 63-150 µm size fraction

**Commented [h48]:** RC1: -Line 200: "of" repeated. Could streamline to, "Comparisons of foraminifera abundances between. . ."

**Commented [h49]:** RC1: -Line 200: change 150 to >150 microns.

**Commented [h50]:** RC1: -Lines 202-204: "Three species have significantly different relative abundances between size classes; three are more likely to be found in the 63-150 µm (Cassidulina carinata, Epistominella sp1 and G. subglobosa)". When you repeat "three" are you referring to the same 3 species? If so, please rephrase.

**Commented [h51]:** RC1: -Line 207: I don't think it is necessary to repeat "Shannon Index" after you describe it in the methodology.

**Commented [h52]:** RC1: -Line 217: When you say above and below the OMZ do you mean on the seafloor? Not in the water column?

**Commented [h53]:** RC1: Please clarify. G. subglobosa as oxic indicator- This assertion needs a discussion or reference. Skipped to discussion

AC: When we refer to above and below the OMZ we are referring to within the water column, not above/below the sediment surface. We are indicating that G. subglobosa is more abundant at the sites above (300m) and below (1175m) the OMZ. As such, it is from our own data that we identify [14]

**Commented [h54]:** RC2: line 222: the term edge domina[15]

**Commented [h55]:** RC1: -Lines 200- Section 3.3: Based [16]

[revised manuscript text omitted]

**Commented [h57]:** RC1: -Line 262: "see Discussion section" can you refer to a specific section number e.g., 4.3?

**Commented [h58]:** RC1: -Line 263: "core top[s]"

**Commented [h59]:** RC1: -Line 270: What do you mean by equal in magnitude? Can you elaborate on that? Perhaps by giving total abundance or percent abundance examples?

-Lines 268-269: please rephrase this sentence for clarity. It is hard to understand your meaning.

**Commented [h60]:** RC2: paragraph starting from line 268: this could be written much simpler, I am not sure I understand the information provided here.

**Commented [h61]:** RC1: -Lines 273- 279: shorten this section to make your findings more impactful. The word [17]

**Commented [h62]:** RC2: Line 301: what does "...size [19]

**Commented [h63]:** RC1: -Lines 300-302: Where can the [20]

**Moved (insertion) [4]**

[revised manuscript text omitted]

**Commented [h76]:** RC2: Figure 1: please give more information in the figure caption including which sites have what kind of results in the text. what are the depths of these sites?

[Figure]

[Figure]

**Figure 2.** Profiles of temperature (a), dissolved oxygen (b), pH (c), and total organic matter (% by weight) (d) across depth transect. Foraminiferal abundance (total calcareous foraminifera) (e) and diversity (Shannon Index, H) (f) are shown for two size fractions. In panels (e) and (f), large black dots are >150 µm size fraction, small black dots are 63-150 µm and black line on diversity plot represents trends from the complete assemblage (>63 µm). Assemblage counts are standardized to the volume (63.62 cm$^3$) of the sampled cylinder of the sediment (core). Gray dashed line shows approximate boundaries of the oxygen minimum zone.

925

**Commented [h77]:** RC1: -Based on the OMZ bounds of Figure 2 it seems the majority of the foraminiferal abundance plots resides in what you have defined at the OMZ. So how can you see increases if you have no "background" to compare it to? Please clarify.
AC: Updated Figure 2 to show defined OMZ.

RC2: Figure 2: water depth on y axis? station names could be implemented.

**Commented [h78]:** AC: Figure 2 Diversity panel updated to alleviate graphical error (same error that occurred in Figure 3)

**Commented [h79]:** RC1: -Figure 2: Extra ")";

[Figure]

**Figure 3:** Relative abundance of foraminiferal species (percent of total calcareous taxa) in core top sample (0-2 cm) vs. water depth (m). Large black dots are >150 µm size fraction, small black dots are 63-150 µm size fraction and black lines represent trends considering the complete assemblage (>63 µm).

**Commented [h80]:** RC1: ." This needs to be identified in Figure 3.

**Commented [h81]:** RC1: -Figure 3: "General observations discussed in the text are noted here, e.g., species that increase in abundance in the OMZ, appear associated with the "edge" of the OMZ, etc. Note difference in x-axis in B. argentea plot" is not very useful information for a figure caption. Please describe the structure of the graph and summarize what you observed or reference to the section ... [21]

**Commented [h82]:** RC2: Figure 3 caption: "General observations .... " this is not needed here. Figure should be cited in the text more often.

**Commented [h83]:**

[Figure]

[Figure]

**Figure 4:** Top 3 panels show relative abundance of two species of low-oxygen foraminifera (*B. spissa* and *U. peregrina*, gray lines) and one species of oxic foraminifera (*G. subglobosa*, blue lines) from the >150 μm size fraction down core through time, in years before present for cores from 3 water depths (528 m, 800 m, 1175 m). Top panel also includes relative abundance of *B. argentea*. Bottom panel shows diversity (Shannon's Index, H) through time for 3 cores.

**Commented [h84]:** RC1: -Figure 4: The Bolivina spissa and U. peregrina lines are very similar in color and hard to distinguish. Can the point shapes be changed to better facilitate reading?
AC: changed color of line

RC2: Figure 4: y axis please mention Rel. Abundance (%) instead.

**Tables**

| Core Name | Water Depth (m) | Latitude | Longitude | Core Length (cm) | Temperature (ºC) | Dissolved Oxygen (mL/L) | pH | Salinity (psu) | Total organic matter (% wt) |
|---|---|---|---|---|---|---|---|---|---|
| MV1217-5-2 | 300 | 32.8100166 | 117.4681 | 16 | 8.6137 | 1.54 | 7.65 | 34.145 | 6.8 |
| MV1217-2-3 | 528 | 32.8100333 | 117.416583 | 26 | 6.6217 | 0.35 | 7.57 | 34.313 | 11.7 |
| MV1217-3-3 | 700 | 32.8099666 | 117.450966 | 20 | 5.8975 | 0.26 | 7.56 | 34.348 | 13.9 |
| MV1217-1-3 | 800 | 32.8095166 | 117.506933 | 20 | 5.0491 | 0.29 | 7.56 | 34.405 | 14.2 |
| MV1217-4-3/1 | 1175 | 32.6333333 | 117.499883 | 16 | 3.8231 | 0.58 | 7.59 | 34.501 | 14.7 |

**Table 1:** Data for cores used in this study. Temperature, dissolved oxygen, pH and salinity were measured in bottom water directly above each coring site.

| Core | Sample Interval | Age ($^{14}$C years) | ± | 1 Sigma Maximum Calendar Age Range | 1 Sigma Minimum Calendar Age Range | Age in Calendar Years | Sedimentation Rate (cm/ka) | ± |
|---|---|---|---|---|---|---|---|---|
| MV1217-2-3 | 11-12 cm | 1230 | 30 | 1403 | 1319 | 1361 | 16.9 | 1.0 |
| MV1217-2-3 | 16-17 cm | 2085 | 30 | 602 | 474 | 538 | 6.1 | 0.2 |
| MV1217-2-3 | 25-26 cm | 2405 | 35 | 237 | 107 | 172 | 24.6 | 0.1 |
| MV1217-4-3 | 5-6 cm | 670 | 35 | 1950 | 1837 | 1893.5 | 42.2 | 11.3 |
| MV1217-4-1 | 10-11 cm | 960 | 30 | 1630 | 1518 | 1574 | 15.6 | 0.2 |
| MV1217-4-1 | 20-21 cm | 1840 | 35 | 817 | 698 | 757.5 | 12.3 | 0.4 |

 **Table 2.** Radiocarbon ages of mixed planktonic foraminifera from MV1217-2-3 (528 m water depth), MV1217-4-3 (1175 m water depth) and MV1217-4-1 (1175 m water depth).

| Page 2: [1] Deleted | hmpalmer4@gmail.com | 2/25/20 10:25:00 AM |
|---|---|---|

| Page 3: [2] Commented [h17] | hmpalmer4@gmail.com | 2/25/20 10:36:00 AM |
|---|---|---|

RC1: RC1: 4. A more complete literature review is needed in the discussion section in order to support the findings of the authors

RC2: paragraph starting with line 67: this section could be improved significantly by including previous observations from other OMZs which should be included in discussion where Bolivinids and nitrate availability are discussed.

RC2: 2. Introduction and discussion should be improved in terms of using literature and previous work from different OMZs. For instance there are significant amount of work from the Peruvian and Arabian OMZs focusing on similar oxygen gradient and benthic foraminiferal assemblages. These studies should be included in terms of benthic foraminifera habitat in relation with oxygen and nitrate availability etc. This will improve the MS significantly. It is a pity that the species are not stained limiting the comparison with previous studies, and yet I believe the information presented here is really valuable.

| Page 3: [3] Deleted | hmpalmer4@gmail.com | 2/25/20 9:13:00 AM |
|---|---|---|

| Page 5: [4] Deleted | hmpalmer4@gmail.com | 2/24/20 10:04:00 AM |
|---|---|---|

| Page 5: [5] Commented [h36] | hmpalmer4@gmail.com | 2/24/20 12:35:00 PM |
|---|---|---|

RC2: Line 205: is ANOVA introduced already in methods?

| Page 6: [6] Commented [h40] | hmpalmer4@gmail.com | 2/24/20 11:24:00 AM |
|---|---|---|

RC1: -Line 157: (DO) once you have introduced an abbreviation you do not have to reference it again and you can then use the shorthand in the text. I suggest you do a search of the manuscript and identify duplicates of instances such as this. I suggest a rewriting of this section for clarity. Either report based on water column depth or minimums. As is, it is confusing. –Rephrasing suggestion: —"Water column dissolved oxygen (DO) concentration documents a low oxygen zone, with a minimum occurring at 700 m water depth (0.26 ml/L; Figure 2, Table 1), compared to 1.54 ml/L at 300 m and 0.58 ml/L at 1175 m. Minimum pH is documented at 700 m (7.55) and is higher above (300 m, 7.65) and below (1175 m, 7.59) the intermediate depth low pH zone (Figure 2, Table 1)." —Water column DO measurements indicate areas of low oxygen availability from 300 m (1.54 ml/L) to 1175 m (0.58 ml/L) with lowest oxygen availability at 700 m (0.26 ml/L). Although not greatly variable, pH minima also occur at 700 m (7.55) and is higher at 300 m (7.65) and 1175 m, (7.59). In this section, why not report as hypoxic, anoxic, as outlined by Bernhard et al?

AC: We will improve upon the introduction of abbreviations earlier on in the text as suggested by both reviewers. We chose not to report oxygen as hypoxic, anoxic, as outlined by Bernhard et al because we later argue that in the environment we studied, these categories of foraminifera by oxygenation do not accurately reflect the foraminifera we sampled. Thus, it is more useful to describe each site using the measured dissolved oxygen.

| **Page 6: [7] Commented [h41]** | hmpalmer4@gmail.com | 2/24/20 12:29:00 PM |

RC2: section 3.1.: this section is confusing. please be clear with what is presented here. I assume these are the deepest points CTD measured? is there any pore water measurements or are these only water column?

| **Page 6: [8] Deleted** | hmpalmer4@gmail.com | 2/24/20 11:24:00 AM |

▼

| **Page 6: [9] Commented [h43]** | hmpalmer4@gmail.com | 2/24/20 10:08:00 AM |

RC1:  -Line 173: Remove "The site" and begin the sentence with "At". Change "at the core of the OMZ" to "within the core of the OMZ, the". Add "occurred" to the end of the sentence. Alternatively, this sentence could be shortened and combined with the previous sentence: Calcareous taxa dominated the assemblage at every site; agglutinated foraminifera made up 0 (e.g., XXX m) to 17.7% of the assemblage at 700 m, within the core of the OMZ.

| **Page 6: [10] Deleted** | hmpalmer4@gmail.com | 2/24/20 10:07:00 AM |

▼

| **Page 6: [11] Commented [h44]** | hmpalmer4@gmail.com | 2/24/20 9:24:00 AM |

RC1: 3. A better discussion of how agglutinated foraminifera are reported, or not, in the total foraminiferal counts should be included.

| **Page 6: [12] Commented [h45]** | hmpalmer4@gmail.com | 2/24/20 11:25:00 AM |

RC1: -Lines 180-181: "These dominant taxa make up more than 84% of all foraminifera counted across all core top samples." Does this include the agglutinates or just 84% of the calcareous foraminiferal population?

| **Page 6: [13] Commented [h46]** | hmpalmer4@gmail.com | 2/24/20 10:14:00 AM |

RC1: -Line 181: "All other species each account for <4% of total assemblage across all core tops." This sentence is confusing. Please rephrase.

| **Page 7: [14] Commented [h53]** | hmpalmer4@gmail.com | 2/24/20 11:29:00 AM |

RC1: Please clarify. G. subglobosa as oxic indicator- This assertion needs a discussion or reference. Skipped to discussion

AC: When we refer to above and below the OMZ we are referring to within the water column, not above/below the sediment surface. We are indicating that *G. subglobosa* is more abundant at the sites above (300m) and below (1175m) the OMZ. As such, it is from our own data that we identify this species as an oxic indicator, rather than from previous work. But, additionally, we can add citations showing this species as an oxic indicator (Kaiho 1999).

| **Page 7: [15] Commented [h54]** | hmpalmer4@gmail.com | 2/25/20 10:43:00 AM |

RC2: line 222: the term edge dominant.. what does this actually mean? According to which previous work edge of the OMZ is considered?

| **Page 7: [16] Commented [h55]** | hmpalmer4@gmail.com | 2/24/20 11:27:00 AM |

RC1: -Lines 200- Section 3.3: Based on the results of this section why are you concluding that it is better to look at the >150micron size fraction as you state in the conclusion?

AC: We conclude that it is effective to look at the > 150 micron size fraction because we find that trends across depth are similar between the complete (>63 μm) and large size fraction (>150 μm) or are more pronounced in the >150 μm size fraction compared to the 63-150 μm size fraction. Further, we utilized this size fraction to remain consistent with other regional studies. These trends are further elucidated by the corrected Figure 3 we have uploaded as part of this response.

| Page 9: [17] Commented [h61] | hmpalmer4@gmail.com | 2/25/20 11:41:00 AM |
|---|---|---|

RC1: -Lines 273- 279: shorten this section to make your findings more impactful. The word "document" is repeated and could be eliminated completely.

| Page 9: [18] Deleted | hmpalmer4@gmail.com | 2/25/20 11:41:00 AM |
|---|---|---|

| Page 9: [19] Commented [h62] | hmpalmer4@gmail.com | 2/25/20 10:47:00 AM |
|---|---|---|

RC2: Line 301: what does "...size fraction or 2" mean? paragraph starting at line 312: for such a discussion based on specific species, authors should provide a reference list including species names as mentioned earlier.

| Page 9: [20] Commented [h63] | hmpalmer4@gmail.com | 2/24/20 11:46:00 AM |
|---|---|---|

RC1: -Lines 300-302: Where can the readers see this stated relationship? "...and the >150 μm size fraction or 2) trends in the >150 μm size fraction are more pronounced than in the complete assemblage"
AC: This relationship can be seen in the updated Figure 3; due to the graphical error in Figure 3, this was not clear in the original submission.

| Page 23: [21] Commented [h81] | hmpalmer4@gmail.com | 2/24/20 11:50:00 AM |
|---|---|---|

RC1: -Figure 3: "General observations discussed in the text are noted here, e.g., species that increase in abundance in the OMZ, appear associated with the "edge" of the OMZ, etc. Note difference in x-axis in B. argentea plot" is not very useful information for a figure caption. Please describe the structure of the graph and summarize what you observed or reference to the section of the paper where it is discussed. Please also clearly state in this figure that "core tops" are the 0-2cm intervals.

---

## Referee Report (RR1)

**Comments on revised version of "Southern California margin benthic foraminiferal assemblages record recent centennial-scale changes in oxygen minimum zone" by Palmer et al.**

**Referee #2**

Overall, I am happy with the revised version answering and revisiting all the comments I had made. There are final minor details that I would recommend authors to have another look. Below comments indicating line numbers are based on the file showing track changes "**bg-2019-446-author_response-version1**".

- Please check usage of abbreviation OMZ throughout the text. Sometimes within the same paragraph it is used and followed by oxygen minimum zone (e.g., first paragraph of introduction; paragraph starting with line 595).

- Relationship between nitrate and benthic foraminifera could still be improved. Current stage of the discussion serves the purpose of the manuscript and I am aware nitrate was npt measured. Nevertheless, I think adding the potential impact of nitrate availability on certain species would only make the MS better. In case Authors decide to improve this part, I would highly recommend them to check publications from Kuhnt et al., 2013 (http://dx.doi.org/10.1016/j.dsr.2012.11.013) and Glock et al., 2019 (www.pnas.org/cgi/doi/10.1073/pnas.1813887116) and the references therein.

- Paragraph starting with line 124 is rather confusing. It says studies focusing on benthic foraminifera in Southern California are limited to the basins and there are few other from outside. Both Mallon and Erdem reported foraminifera from the Peruvian margin. I would recommend either remove these citations or change the description to Eastern Pacific Ocean continental margin, or something like that. In this case, please check Cardich et al., 2019 (https://doi.org/10.3389/fmars.2019.00270).

- Line 419. Short names can be given instead of the full names of the species.

- Table 1. Four digit after decimal points needed?

- I am glad to see a plate included in the supplementary material. When I mentioned about a species list with references I had something like this I prepared below. These are species mentioned in the manuscript. Hopefully, I did not miss any. This could be included after the plate as done in Balestra et al., 2018; Erdem and Schoenfeld, 2017.

**Example taxonomic reference list for benthic foraminifera species mentioned in the MS:**

*Bolivina argentea* Cushman 1926. (Figure S4)

*Bolivina spissa Bolivina subadvaena* Cushman var. *spissa* Cushman 1926 (Figure S4)

*Bulimina sp.* (any note on why species is not identified?)

*Cassidulina carinata Cassidulina laevigata* d'Orbigny var. *carinata* Silvestri, 1896 (Figure S4)

*Cibicidoides sp.* (any note on why species is not identified?)

*Epistominella sp.* (Figure S4)

*Globobulimina pacifica* …. and so on…

*Globobulimina ovata*

*Globocassidulina subglobosa* (Figure S4)

*Nonionella stella*

*Quinqueloculina sp* (any note on why species is not identified?)

*Uvigerina peregrina*  (Figure S4)

---

## Author Response (AR3)

Author's Response for:

**Southern California margin benthic foraminiferal assemblages record recent centennial-scale changes in oxygen minimum zone**

Hannah M. Palmer[1,2], Tessa M. Hill[1,2], Peter D. Roopnarine[3], Sarah E. Myhre[4], Katherine R. Reyes[1,5], and Jonas T. Donnenfield[1,6]

At this stage, the manuscript underwent very small changes.

A data citation for the data used in this manuscript was added to the references section and updated in the Data Availability section of the paper.

All other changes were made during the Major Revision and Minor Revision stages and can be seen below.

Minor Revisions:
Response to Referee Comment 2
A marked-up manuscript with track-changes included

Major Revisions:
Response to Referee Comment 1
Response to Referee Comment 2
A marked-up manuscript with track-changes included

**Response to RC2 Minor Revisions**

RC2: Reviewer comment (in black)
AC: Author comment (in gray)

RC2: Comments on revised version of "Southern California margin benthic foraminiferal assemblages record recent centennial-scale changes in oxygen minimum zone" by Palmer et al. Referee #2
Overall, I am happy with the revised version answering and revisiting all the comments I had made. There are final minor details that I would recommend authors to have another look. Below comments indicating line numbers are based on the file showing track changes "bg-2019-446-author_response-version1".
- AC: We thank the reviewer for their additional reviews and comments. We have responded to each comment below.

RC2: Please check usage of abbreviation OMZ throughout the text. Sometimes within the same paragraph it is used and followed by oxygen minimum zone (e.g., first paragraph of introduction; paragraph starting with line 595).
- AC: We have made the suggested correction and streamlined the use of the OMZ abbreviation throughout the manuscript.

RC2: Relationship between nitrate and benthic foraminifera could still be improved. Current stage of the discussion serves the purpose of the manuscript and I am aware nitrate was not measured. Nevertheless, I think adding the potential impact of nitrate availability on certain species would only make the MS better. In case Authors decide to improve this part, I would highly recommend them to check publications from Kuhnt et al., 2013 (http://dx.doi.org/10.1016/j.dsr.2012.11.013) and Glock et al., 2019 (www.pnas.org/cgi/doi/10.1073/pnas.1813887116) and the references therein.
- AC: We have added an additional sentence on benthic foraminiferal nitrate respiration in the discussion and included the suggested citations.

RC2: Paragraph starting with line 124 is rather confusing. It says studies focusing on benthic foraminifera in Southern California are limited to the basins and there are few other from outside. Both Mallon and Erdem reported foraminifera from the Peruvian margin. I would recommend either remove these citations or change the description to Eastern Pacific Ocean continental margin, or something like that. In this case, please check Cardich et al., 2019 (https://doi.org/10.3389/fmars.2019.00270).
- AC: We have updated the language as recommended and removed the citations. Further, we have included a citation of Cardich et al., 2019 in the manuscript.

RC2: Line 419. Short names can be given instead of the full names of the species.
- AC: We made the suggested correction.

RC2: Table 1. Four digit after decimal points needed?
- AC: We updated the Temperature column on Table 1 to reflect the same number of digits after decimal points to other reported variables (Salinity).

RC2: I am glad to see a plate included in the supplementary material. When I mentioned about a species list with references, I had something like this I prepared below. These are species mentioned in the manuscript. Hopefully, I did not miss any. This could be included after the plate as done in Balestra et al., 2018; Erdem and Schoenfeld, 2017.

Example taxonomic reference list for benthic foraminifera species mentioned in the MS:
*Bolivina argentea* Cushman 1926. (Figure S4)
*Bolivina spissa Bolivina subadvaena* Cushman var. *spissa* Cushman 1926 (Figure S4)
*Bulimina* sp. (any note on why species is not identified?)
*Cassidulina carinata Cassidulina laevigata* d'Orbigny var. *carinata* Silvestri, 1896 (Figure S4)
*Cibicidoides* sp. (any note on why species is not identified?)
*Epistominella* sp. (Figure S4)
*Globobulimina pacifica* …. and so on…
*Globobulimina ovata*
*Globocassidulina subglobosa* (Figure S4)
*Nonionella stella*
*Quinqueloculina* sp (any note on why species is not identified?)
*Uvigerina peregrina* (Figure S4)
- AC: Thank you for the suggestion. We have included a Taxonomic Reference List for the species discussed in this manuscript according to the outline given above. The list can be found in the Supplemental Information following Supplemental Figure 4.

[revised manuscript text omitted]
 four supplemental figures and a taxonomic reference list for the benthic foraminiferal species included in the text of the manuscript. All materials and methods for data collection are described in the main text of this article.

[Figure]

**Figure S1.** Distance Based Redundancy Analysis of core-top benthic foraminiferal assemblages including both size fractions (>63 μm). Species are listed in red – species discussed in manuscript are labeled. Sites are listed as water depth in meters. Blue arrows show ordination of environmental factors T= temperature, DO = dissolved oxygen, TOM= total organic matter). Eigenvalues for two axes are CAP1 (0.4575) and CAP2 (0.2024).

[Figure]

**Figure S2.** Age model for two cores. Age shown in years before present. Linear interpolation between radiocarbon dates (+). MV1217-2-3 (528 m water depth) is black line. MV1217-4-3 and MV1217-4-1 (1175 m) is blue line. Dashed lined represent +/- 1 sigma years.

[Figure]

**Figure S3.** Non-metric multidimensional scaling plot of all benthic foraminiferal assemblages through time. Species are plotted as small gray dots. All other points represent assemblages from 1 cm intervals through time at each of the 5 cores. Legend shows which dots represent each water depth/site (Red triangle = 300 m, blue square = 528 m, orange crosshair = 700 m, Grey box with x = 800 m, black circle = 1175 m).

[Figure]

*Bolivina spissa*

*Uvigerina peregrina*

*Bolivina argentea*

*Epistominella* sp.

*Cassidulina carinata*

*Globocassidulina subglobosa*

**Figure S4:** Light microscope images of benthic foraminifera (*Bolivina spissa, Uvigerina peregrina, Bolivina argentea, Epistominella* sp., *Cassidulina carinata, Globocassidulina subglobosa*) evaluated in this study. All scale bars are 100 µm.

**Taxonomic reference list of benthic foraminifera species included in manuscript**

*Bolivina argentea* Cushman 1926. (Figure S4)
*Bolivina spissa = Bolivina subadvaena* Cushman var. *spissa* Cushman 1926. (Figure S4)
*Bulimina* spp.* (includes *Bulimina mexicana = Bulimina inflata* Seguenza var. *mexicana* Cushman, 1922 and
           *Bulimina exilis = Bulimina elegans* d'Orbigny var. *exilis* Brady, 1884.)
*Cassidulina carinata = Cassidulina laevigata* d'Orbigny var. *carinata* Silvestri, 1896. (Figure S4)
*Cibicidoides* sp. **
*Epistominella* sp. (Figure S4)
*Globobulimina pacifica* Cushman, 1927.
*Globobulimina ovata* d'Orbigny 1846.
*Globocassidulina subglobosa = Cassidulina subglobosa* Brady, 1881. (Figure S4)
*Nonionella stella = Nonionella miocenica* Cushman var. *stella* Cushman and Moyer, 1930.
*Quinqueloculina* sp.**
*Uvigerina peregrina* (Figure S4) Cushman 1923.

\* Individuals in this genus were identified to species but considered together for the interpretations of this paper.
\*\*Due to low abundance, individuals in this genus were not identified to species level.

Author's response for:

**Southern California margin benthic foraminiferal assemblages record recent centennial-scale changes in oxygen minimum zone**

Hannah M. Palmer, Tessa M. Hill, Peter D. Roopnarine, Sarah E. Myhre, Katherine R. Reyes, and Jonas T. Donnenfield

This document contains:

Response to Referee Comment 1
Response to Referee Comment 2
A marked-up manuscript with track-changes included

**Response to Referee Comment 1**

RC1: Referee comment 1, in gray
AC: Author comment, in black

AC: We thank Referee 1 for their helpful comments and suggestions to improve the manuscript. We have addressed each major comment below. We will incorporate all technical corrections listed by Referee 1 unless a comment is made below.

RC1: The paper titled, "Southern California margin benthic foraminiferal assemblages across a modern environmental gradient record recent centennial-scale changes in oxygen minimum zone" quantifies benthic foraminifera abundances and assemblages of surface multicore sections (0-2cm) and downcore (_20cm). Overall, despite the shortcomings in the methodology, and lack of clarity in the writing style of this manuscript, the information offered in this study shows potential in its importance to current and future oxygen investigations using benthic foraminifera. I recommend this manuscript undergo Major Revisions including a significant re-write, standardization of data, and incorporate a more thorough literature review.

This manuscript is well organized and contains most of the necessary information, but would benefit from re-writing and shortening of certain sections for clarity and readability. Suggested edits, such as changes in the present and past tenses and excluding the first-person point of view writing are included in the "Technical Corrections" section below. As is currently written the manuscript needs to be tied to established literature references and include a much more comprehensive discussion section justifying methods and interpretations. There also needs to be a section acknowledging the limitations of the regional interpretations made here. The authors state "Our findings indicate that region (or environment) specific oxygen species associations may be necessary, as our findings do not align directly with previous categorization of species." It is important that it is recognized that these results, although true in the Southern California Margin, may not show the same patterns elsewhere in the world's oceans and/or for the authors to compare their results with those previously published. Based on the current manuscript an understanding of how strong the interpretations are is unclear. What additional data would be needed to test or refine the applications and interpretations? All of these issues, limitations, and confounding factors may be addressed in a Major Revision.

AC: We thank the reviewer for this feedback; as shown below, we will adapt the writing of the manuscript, address data standardization, and incorporate a more thorough literature review.

RC1: Specific comments regarding scientific questions/issues:
RC1: 1. How are raw abundance counts standardized? Are all abundances reported as raw numbers? If so, how do you account for inconsistencies in the amount of sediment examined in each interval? The data should be standardized to #/50cc or #/gram.

AC: Raw abundance counts
The abundance counts are done by volume of sediment relative to total, then multiplied to equal the "total" volume. Thus, these should represent the amount of foraminifera in the originally

sampled cylinder of sediment (core) 1cm height x 9 cm diameter = 63.62 cm^3. This will be updated in the Methods section of the paper and in Figure 2.

RC1: 2. It is unfortunate that samples were not able to be stained using Rose Bengal, however, I do still believe the data to be extremely valuable in that it is necessary to examine what surficial fossil assemblages are when using them down core. Rose Bengal stained samples here would have completed the picture, but the information available from this data set is valuable and needed for fossil interpretations. I am not sure how you make the assertion that examination of only the larger size fraction (>150micron) is suitable for paleooxygen investigations, at least, not with the text in its current form. Additional references and discussions are needed. Below is what I pulled from the manuscript and viewed as the line of thinking on the subject. Please reorganize and emphasize the statements to better clarify this argument. -Lines 200- Section 3.3:
-Lines 291: "species richness is greater in the >150 μm relative to the 63-150 μm fraction, yet there is no consistent relationship between diversity and size fraction ". Is this your argument for looking only at the >150 micron size fraction? If so, you need to include a better discussion of the relevant literature which suggests the >63 micron fraction is necessary to examine in environments with low oxygen and high carbon inputs.
-Line 297: "diversity is not driven by oxygen alone" And there is no clear pattern in diversity and size. So, what is the basis on which you are making the argument that only the >150 microns are necessary in oxygen related studies? I suggest referencing Keating-Bitonti and Payne, 2016 when writing this section. You include this as a reference, but you don't really discuss or justify referencing it. -Lines 314-316: "Our findings show that spatial trends in the >150 μm size fraction generally reflect those found in
the >63 μm size fraction or are muted by the inclusion of the 63-150 μm." Have you presented this data for the readers to see? Is this Figure 3? If so, reference the figure here. If not, you need to provide supporting evidence.
-Lines 324-325: "...correlation between size of specimen and accuracy of identification, meaning that the inclusion of the smaller specimens in the >63 um fraction may reduce the accuracy of identification
. . ." This is a good point and should be discussed sooner for better emphasis.

AC: After submission of the manuscript, we realized that there was a graphical error that affected the plotting of some of the data in Figure 3.  The interpretations of the figure in the text were accurate, it was the figure itself that did not display correctly. We have uploaded an updated version of Figure 3 in response to this comment and the updated figure should help to address the questions posed here by Referee #1. In this figure, it is apparent that "spatial trends in the >150 μm size fraction generally reflect those found in the >63 μm size fraction or are muted by the inclusion of the 63-150 μm." This was unclear in the previous iteration of the figure but has now been made clearer. We will cite this figure in the text more heavily to reference for understanding.

RC1: 3. A better discussion of how agglutinated foraminifera are reported, or not, in the total foraminiferal counts should be included. I see no reason why not to include them in the abundances reported (is that what you are already doing?). Then simply report percentages of certain species within the calcareous population if that is what you are interested in (e.g., Agg sp. 1 comprises 10% of the total population; Cal sp. 1 comprises 30% of the total population and

50% of calcareous population). I think excluding agglutinated foraminifera populations will impede on future research of agglutinated species.

AC: We appreciate the interest in agglutinated species. While they were counted in this study, they were not speciated. Additionally, as is discussed further below, they are not well preserved so not utilized for investigations that reconstruct environments through time. For this reason, it is the industry standard to only utilize calcareous species in relative abundance analyses.
We noted other papers have reported on agglutinated taxa in the same way (Venturelli et al 2018) or have excluded examination of all agglutinated (Balestra et al 2017, Kaiho 1994, and Myhre et al 2014). In an effort for our study to be comparable to other regional studies we chose to report relative abundance as percent of total calcareous taxa. We acknowledge that due to the lack of preservation of agglutinated foraminifera, fossil assemblages may capture an incomplete reconstruction of diversity and ecosystem function.

RC1: 4. A more complete literature review is needed in the discussion section in order to support the findings of the authors. I suggest two discussion/literature review sections: 1) examination of <150 micron (>63 or 63-150 micron) foraminifera in low oxygen/high carbon environments. Do an overview of who proposed it, what results have been seen, and how it relates to your data. 2) foraminifera test size in comparison with oxygen. Here would be a great chance to discuss Keating-Bitonti and Payne, 2016 in greater detail. Again, do an overview of who proposed it, what results have been seen, and how it relates to your data.

AC: We will incorporate a more thorough literature review of the distribution of foraminifera in low oxygen/high carbon environments and of foraminiferal test size in relation to oxygen. We will include Keating-Bitonti and Payne 2016, Keating-Bitonti and Payne 2018, Kaiho et al 1994, Kaiho et al 1999, Venturelli et al 2019, and others.
* * *
RC1: Technical Corrections
AC: We will incorporate **all** technical corrections listed by Referee 1 that are listed below

-Line 22: "San Diego Margin" is capitalized here and in the text, but not in the title. Choose one and be consistent.
-Line 22: "Here, we" can be removed. Start the sentence with, "Five core tops were analysed ."
-Line 24: Assemblage changes downcore? If so, please state. -First-person point of view should not be used in scientific writing. Please modify the text accordingly.
-Line 30: "variability in [the] upper margin"
-Line 31: "stable in the last 1.5ka" add a space before ka to remain consistent with the formatting of other units.
-Line 42: "declines in dissolved oxygen," awkward. Please rephrase.
-Line 43: determination of
-Line 49: "Low-oxygen zones typically contain both low abundance
and low diversity of" change to "Areas of low oxygen availability typically contain low abundance and diversity of "

-Line 73: "2013), yet significantly fewer" remove ", yet". Begin the sentence with "Significantly. . ." Do you have any examples to cite e.g.,? -

-Line 82: remove "current, the" or something I just don't like how it is phrased.

-Line 85: too many back-to-back parentheses. –This entire paragraph should be shortened. Along the California margin, a large, intermediate-depth oxygen minimum zone (OMZ) and carbon maximum zone (CMZ; approx.. 500 – 1000 m water depth, Helly and Levin, 2004; Stramma et al., 2010) are formed and controlled through physical and biological processes including temperature-dependent diffusion from atmosphere, mixing, stratification, deep water circulation, primary productivity at the surface and respiration at depth (Gilly et al., 2013; Helly and Levin, 2004). Intensity and spatial extent of the modern California margin OMZ is influenced by physical mixing of well-oxygenated surface water, biological activity at the surface and at depth, and intrusion of lower oxygen bottom waters (Gilly et al., 2013).

-Line 107: change core to coring

-Line 119: (Grupe. . .2015) should not be italicized.

-Lines 123-128: The word sample is used too many times (5 in one sentence). Try to replace with another word to avoid monotony.

-Line 130: "present in the sample." I'm assuming it is in the entire sample? If so, add "entire" before "sample"

-Lines 132-134: These two sentences should be combined with the previous paragraph as 2 sentences is not a paragraph on its own.

-Section 3.2: this section is written in present tense while the rest is in past tense. Change all "was" to "were", "is" to "was", etc. –Where is this information reported? A graph a table? Reference it in this section.

-Lines 169-172: "Foraminiferal abundance is low (<100 individuals) in some of the samples from 0-1 cm. Thus, in order to utilize sufficient numbers of individuals and because there were no significant differences in abundance of species between 0-1 cm and 1-2 cm, for the rest of the discussion we refer to the 0-2 cm fraction as the core top material." This needs to be identified in Figure 3.

 -Line 173: Remove "The site" and begin the sentence with "At". Change "at the core of the OMZ" to "within the core of the OMZ, the". Add "occurred" to the end of the sentence. Alternatively, this sentence could be shortened and combined with the previous sentence: Calcareous taxa dominated the assemblage at every site; agglutinated foraminifera made up 0 (e.g., XXX m) to 17.7% of the assemblage at 700 m, within the core of the OMZ.

-Line 179: "sp1" can this just be written as "sp."?

-Line 181: "All other species each account for <4% of total assemblage across all core tops." This sentence is confusing. Please rephrase.

 -Line 200: "of" repeated. Could streamline to, "Comparisons of foraminifera abundances between. . ."

-Line 200: change 150 to >150 microns.

-Lines 202-204: "Three species have significantly different relative abundances between size classes; three are more likely to be found in the 63-150 μm (Cassidulina carinata, Epistominella sp1 and G. subglobosa)". When you repeat "three" are you referring to the same 3 species? If so, please rephrase.

-Line 207: I don't think it is necessary to repeat "Shannon Index" after you describe it in the methodology.

-Line 296: Shannon Index -Line 341: remove DO as it is implied
-Line 262: "see Discussion section" can you refer to a specific section number e.g., 4.3?
-Line 263: "core top[s]"
-Lines 268-269: please rephrase this sentence for clarity. It is hard to understand your meaning.
-Lines 273- 279: shorten this section to make your findings more impactful. The word "document" is repeated and could be eliminated completely.
-Lines 338-339: "Infaunal species are more common within the OMZ, while epifaunal are more common in well-oxygenated areas." True, but there are studies that illustrate that epifaunal abundances may be limited by substrate rather than a physiological limitation to oxygen availability (see comment above). You just alluded to this in the previous sentence, but I suggest incorporating something of this statement into this sentence or starting the sentence with, "In general".
-Line 440: remove "the"; change "oxygenation" to "oxygen"
-Lines 442 and 443: change "classes" to "fractions" -Lines 444-445: "we conclude that analysis of the >150 μm assemblage for this site provides the most useful record for interpreting benthic foraminifera as a proxy for past change" This assertion needs a discussion or reference
-Line 447: "variability in upper margin of the OMZ" should be "variability in [the] upper margin of the OMZ"
-Lines 448-449: "We document expansion of upper margin of OMZ beginning 400 ybp on San Diego Margin that is synchronous with regional records of oxygenation." Should be re-written. Perhaps, "In this study, upper margin OMZ expansion beginning 400 ybp on San Diego Margin is synchronous with regional records of oxygenation."?
-Figure 2: Extra ")";
-Figure 4: The Bolivina spissa and U. peregrina lines are very similar in color and hard to distinguish. Can the point shapes be changed to better facilitate reading?
-Table 1: Include salinity in the table
-Table 2: Contains a core not "used in this study" MV1217-4-1. If it is to be included in this table it would be helpful to also include depth and lat-long so readers have an idea of why you are including this in the manuscript. 14 in the 14C needs to be superscripted
* * *
RC1: Technical Corrections
AC: We have addressed or commented on the following suggested changes:

-Line 29: "diversity is not driven by oxygenation" In the core top materials? Is this pattern observed with Shannon Index downcore?

AC: Diversity is not driven by oxygenation in the core top samples. This statement is not referring to the downcore changes in diversity. We will clarify this in the text.

Line 75: "biological and chemical gradients are more extensively distributed" or more extensively variable?

AC: We will change the text to "more variable."

-Lines 107-108: . . ."CTD (SBE9), with a dissolved oxygen probe (Seabird Electronics Sensor SBE43). . ."- CTDs typically have an oxygen probe. Is it necessary to state this?

AC: We included this to show that values for dissolved oxygen were acquired using a sensor rather than by bottle sampling and Winkler titration.

-Lines 110-111: "Bottom depths were measured acoustically at each site." Is it necessary to state this?

AC: For completeness, we included how each of the environmental parameters were measured.

-Line 117-118: "frozen. Sediments were not stained upon retrieval; thus, we could not carry out an analysis of live vs. dead or depth habitat of these species" This is too bad. It would have made for a much stronger study if living and fossil assemblages were able to be extracted from core top samples. I don't think the data presented here is useless, but having stained (recently living) samples to compare with the dead assemblages at the surface would have provided a much more powerful interpretation.

AC: We agree. This study would be stronger if the sediments had been stained at the time of collection.

-Line 132: I would not qualify 1-2cm as surface. It is shallow infaunal.

AC: We utilized both the 0-1 and 1-2 cm fraction as the "surface" for several reasons. These are discussed in section 3.2 Lines 168-172. We completed a comparison of the 0-1cm and 1-2cm "We do not identify any significant relationship between relative abundance of a species and depth interval (in all cases p>0.05 or r2 is <0.001). Foraminiferal abundance is low (<100 individuals) in some of the samples from 0-1 cm. Thus, in order to utilize sufficient numbers of individuals and because there were no significant differences in abundance of species between 0-1 cm and 1-2 cm, for the rest of the discussion we refer to the 0-2 cm fraction as the core top material." (Lines 168-172).

-Lines 133-134: ". . . were examined at 1 cm intervals for cores MV1217-2-3 (528 m), MV1217-1-3 (800 m) and MV1217-4-3 (1175 m)." Ending at what depth in the cores?

AC: We will add in the depth (in cm) that we analyzed from each core.

-Line 156: no units on salinity. Remove "with a total" replace "range of" with "ranging from"

AC: We chose to report salinity in "practical salinity units." We will make the additional language change.

-Line 157: (DO) once you have introduced an abbreviation you do not have to reference it again and you can then use the shorthand in the text. I suggest you do a search of the manuscript and identify duplicates of instances such as this. I suggest a rewriting of this section for clarity. Either report based on water column depth or minimums. As is, it is confusing. –Rephrasing

suggestion: —"Water column dissolved oxygen (DO) concentration documents a low oxygen zone, with a minimum occurring at 700 m water depth (0.26 ml/L; Figure 2, Table 1), compared to 1.54 ml/L at 300 m and 0.58 ml/L at 1175 m. Minimum pH is documented at 700 m (7.55) and is higher above (300 m, 7.65) and below (1175 m, 7.59) the intermediate depth low pH zone (Figure 2, Table 1)." —Water column DO measurements indicate areas of low oxygen availability from 300 m (1.54 ml/L) to 1175 m (0.58 ml/L) with lowest oxygen availability at 700 m (0.26 ml/L). Although not greatly variable, pH minima also occur at 700 m (7.55) and is higher at 300 m (7.65) and 1175 m, (7.59). In this section, why not report as hypoxic, anoxic, as outlined by Bernhard et al?

AC: We will improve upon the introduction of abbreviations earlier on in the text as suggested by both reviewers. We chose not to report oxygen as hypoxic, anoxic, as outlined by Bernhard et al because we later argue that in the environment we studied, these categories of foraminifera by oxygenation do not accurately reflect the foraminifera we sampled. Thus, it is more useful to describe each site using the measured dissolved oxygen.

-Line 175: "Due to their propensity for degradation and to remain consistent with other regional studies, we exclude agglutinated taxa and all values are reported as percent of total calcareous taxa for the remainder of the text." I don't disagree with this statement, but I think that ignoring the agglutinates is a mistake we will regret in the future. Why not keep the data (they were a significant portion of the population) and just report Calcareous populations in the graphs etc?

AC: In order to remain consistent with other studies (Kaiho 1994, Balestra et al 2017, Myhre et al 2014) that focus only on calcareous taxa, we exclude the agglutinated taxa from further discussion.

-Lines 180-181: "These dominant taxa make up more than 84% of all foraminifera counted across all core top samples." Does this include the agglutinates or just 84% of the calcareous foraminiferal population?

AC: This only takes into account calcareous taxa. We will make this clear in the text.

-Lines 200- Section 3.3: Based on the results of this section why are you concluding that it is better to look at the >150micron size fraction as you state in the conclusion?

AC: We conclude that it is effective to look at the > 150 micron size fraction because we find that trends across depth are similar between the complete (>63 μm) and large size fraction (>150 μm) or are more pronounced in the >150 μm size fraction compared to the 63-150 μm size fraction. Further, we utilized this size fraction to remain consistent with other regional studies. These trends are further elucidated by the corrected Figure 3 we have uploaded as part of this response.

-Line 217: When you say above and below the OMZ do you mean on the seafloor? Not in the water column? Please clarify. G. subglobosa as oxic indicator- This assertion needs a discussion or reference. Skipped to discussion

AC: When we refer to above and below the OMZ we are referring to within the water column, not above/below the sediment surface. We are indicating that *G. subglobosa* is more abundant at the sites above (300m) and below (1175m) the OMZ. As such, it is from our own data that we identify this species as an oxic indicator, rather than from previous work. But, additionally, we can add citations showing this species as an oxic indicator (Kaiho 1999).

-Line 295-296: I'm not sure this was the findings of Venturelli et al., 2018. Which section of the paper are you referring to here? The focus of this paper was on comparing sediment grain size, oxygen, and foraminiferal abundances in the Southern California Bight. It was not an OMZ survey and therefore I am not sure they proposed seeing variations of foraminiferal abundances in size fractions "within" the OMZ vs outside the OMZ. If you just mean to say that 63-150 micron foraminifera were more abundant than >150 micron reported by Venturelli et al., 2018 I think this is true, but how different were the populations and would grain size difference influence these abundances?

AC: We will remove the citation for Venturelli et al 2018 and instead incorporate a more thorough discussion of the relationship between size and oxygenation in benthic foraminifera following the discussion in Keating-Bitonti and Payne 2016. See comment below on adding more thorough literature review and discussion of size.

-Lines 355-338: "The presence of oxic indicator species across all water depths may provide evidence for periodic flushing of high oxygen water or a selection for species that can tolerate a range of environmental conditions rather than a specific threshold of oxygenation." It may also be that the physiological tolerances of indicator species are not fully understood. They may be able to tolerate lower oxygen than previously thought provided another incentive such as substrate (e.g., Venturelli et al., 2018, Burkett et al., 2016), and or they may be able to tolerate short periods of low oxygen or inhospitable conditions (Bernhard et al., 2010).

AC: This is an interesting point. We will add further discussion and incorporate the references cited by Referee 1 (Venturelli et al., 2018, Burkett et al., 2016, Bernhard et al., 2010).

-Line 270: What do you mean by equal in magnitude? Can you elaborate on that? Perhaps by giving total abundance or percent abundance examples?

AC: In multivariate space, the difference between sites across space is greater than within any one site through time. We will clarify this in the text.

-Lines 300-302: Where can the readers see this stated relationship? "...and the >150 μm size fraction or 2) trends in the >150 μm size fraction are more pronounced than in the complete assemblage"

AC: This relationship can be seen in the updated Figure 3; due to the graphical error in Figure 3, this was not clear in the original submission.

-Lines 300-310: YES! I totally agree with some of the things you are saying in this section, but you have to do a complete discussion of the literature. I would suggest splitting it up into two

sections 1) examination of <150 micron foraminifera in low oxygen/high carbon environments. Do an overview of who proposed it, what results have been seen, and how it relates to your data. The second literature comparison should be 2) foraminifera test size in comparison with oxygen. Here would be a great chance to discuss Keating-Bitonti and Payne, 2016 in greater detail. Again, do an overview of who proposed it, what results have been seen, and how it relates to your data.

AC: We will incorporate a more thorough literature review of the distribution of foraminifera in low oxygen/high carbon environments and of foraminiferal test size in relation to oxygen.

 -Line 447: by "core" do you mean center? Is there another word you can use here so as not to confuse it with sediment cores?

AC: In this sentence we are referring to the center of the OMZ, we will change the language so that it is not confused with sediment cores.

-Data availability: is the section with data files raw counts? Or is this standardized per volume? See discussion in the "scientific questions/issues" section.

AC: We have addressed this question in a comment above. The raw data are given as raw abundance by volume of original sediment core.

-Figure 3: "General observations discussed in the text are noted here, e.g., species that increase in abundance in the OMZ, appear associated with the "edge" of the OMZ, etc. Note difference in x-axis in B. argentea plot" is not very useful information for a figure caption. Please describe the structure of the graph and summarize what you observed or reference to the section of the paper where it is discussed. Please also clearly state in this figure that "core tops" are the 0-2cm intervals. -Based on the OMZ bounds of Figure 2 it seems the majority of the foraminiferal abundance plots resides in what you have defined at the OMZ. So how can you see increases if you have no "background" to compare it to? Please clarify. The key should be bounded by a box to better identify it.

AC: We will make the suggested graphical improvements. Further, by updating Figure 3 to eliminate the graphical error, we will fix these issues. We will more heavily cite Figure 3 in the text so that it is clear when interpretations are based on this data.

**Response to Referee Comment 2**

RC2: Referee comment 2, in gray
AC: Author comment, in black

AC: We thank Referee 2 for their helpful comments and suggestions to improve the manuscript. We have addressed each major comment below. We will incorporate all "notes for specific parts" listed by Referee 2 unless a comment is made below.

RC2: This manuscript entitled "Southern California margin benthic foraminiferal assemblages across a modern environmental gradient record recent centennial-scale changes in oxygen minimum zone" by Palmer et al. presents a valuable dataset from a gradient of one of the most prominent OMZs in the world. It presents the calcareous benthic foraminiferal assemblages focusing in size fractions from core tops along a depth transect. Later the authors investigate these assemblages in short cores in order to investigate the recent history of the OMZ and the benthic foraminiferal assemblages. The information provided here is an important input for the ongoing investigations regarding the relationship between OMZs, ecosystem and climate. Overall, I found some major details missing in this study and I believe it can be improved significantly.

1. This study is based on benthic foraminifera taxonomy work which should include references to species nomenclature also preferably a plate showing the major species mentioned. In case it is not possible to provide figures, there should be a section where list of species observed is given with references used for identification. For example: Bolivina spissa = Bolivina subadvaena Cushman var. spissa Cushman 1926a. [Figures 10.7, 11.4]. This is essential for taxonomy based papers where the reader will be able to evaluate the information provided. The number of publications without any reference material is increasing and this leads the misinterpretations regarding the foraminifera research. The authors mentions their concerns in the discussion therefor I highly encourage this MS to have section dedicated to nomenclature.

AC: We will update the list of species observed with the references used for identification. We will also add a plate with images of the 6 species that are discussed in depth in the paper.

RC2: 2. Introduction and discussion should be improved in terms of using literature and previous work from different OMZs. For instance there are significant amount of work from the Peruvian and Arabian OMZs focusing on similar oxygen gradient and benthic foraminiferal assemblages. These studies should be included in terms of benthic foraminifera habitat in relation with oxygen and nitrate availability etc. This will improve the MS significantly. It is a pity that the species are not stained limiting the comparison with previous studies, and yet I believe the information presented here is really valuable.

AC: We will improve our literature review and include more literature from OMZs outside of the North Pacific (Erdem et al 2019, Caulle et al 2014, Enge et al 2016, Mallon et al 2011, Mazumder et al 2014). Further, per the comments of referee 1, we will incorporate a more thorough literature review on the distribution of foraminifera in low oxygen/high carbon environments and of foraminiferal test size in relation to oxygen.

RC2: 3. Presentation of environmental parameters is confusing. Are these values from measurements of bottom water conditions? deepest depth of CTD? Figure 2 should be improved accordingly where stations can be shown.

AC: The environmental parameters listed are from measurements of bottom water conditions taken at the same time as sediment core sampling.

RC2: 4. Definition of an OMZ: please introduce OMZ already in introduction. This MS uses certain terms such as OMZ edge, suboxia, hypoxic boundary and so on; to eliminate the confusion, edge or boundary of an OMZ considered here should be introduced as early in the text as possible.

AC: We will incorporate an introduction to nomenclature in the introduction that will make the entire paper more readable and streamlined. We received similar comments from referee #1 and we will address them both.

Notes for specific parts:

page 2, line 60: please check Tetard et al., 2017.

AC: We will add the suggested reference to this section of the paper.

paragraph starting with line 67: this section could be improved significantly by including previous observations from other OMZs which should be included in discussion where Bolivinids and nitrate availability are discussed.

AC: Based upon this suggestion and that of Reviewer 1, additional observations from OMZs will be added.

page 3, line 107: should be "dissolved oxygen concentration"

AC: We will incorporate the suggested change.

page 4, section under 2.3. needs to be rewritten considering the steps taken to reach the species counts. first, material sieved, dried and count in different fractions. Which references were considered for 300 and 150 specimens? Why did the authors decide these numbers? I am not an expert for statistical methods but what is the reason behind using dbRDA but not component analysis (CCA?) to test the relationship between foraminifera and environmental parameters?

AC: We chose to use a dbRDA instead of a CCA because drRDA allows for the use of Bray-Curtis dissimilarity rather than Euclidean distance in quantifying differences between groups and is able to integrate data from drivers (environmental factors) as well as assemblages.

line 145: "...mixed planktonic foraminifera species" please remove bulk

AC: We will incorporate the suggested change.

section 3.1.: this section is confusing. please be clear with what is presented here. I assume these are the deepest points CTD measured? is there any pore water measurements or are these only water column? and salinity should be included as well in the table.

AC: The data shown here are the deepest CTD points measured, not pore water measurements. We will add salinity to the table.

Line 205: is ANOVA introduced already in methods?

AC: ANOVA was not introduced in the methods. We will add ANOVA to section 2.3 on foraminifera assemblages.

line 222: the term edge dominant.. what does this actually mean? According to which previous work edge of the OMZ is considered?

AC: Incorporating the referee's earlier comment about adding some clarifying language and nomenclature to the introduction would be helpful here as well. In this case, we are referring to a species (*B. argentea*) that is most abundant at 528m water depth, near the upper margin of the modern OMZ. A previous study that we cite in the paper, Mullins et al 1985, also finds a high abundance of some species of benthic foraminifera at similar depths and attributes this to the higher concentrations of biologically available nitrate and nitrifying bacteria at the edges of the OMZ as compared to the center. We will cite this paper here to show support.

line 224: sentence starting with "in some taxa,.." needs rephrasing. paragraph starting from line 268: this could be written much simpler, I am not sure I understand the information provided here.

AC: We will improve these lines for clarity and simplicity.

page 9 first paragraph: we know today oxygen limited high organic matter input regions are characterised by high population low diversity of benthic foraminifera. it is interesting to see this is not the case at these sites. Nevertheless, I am not fully convinced the evidence provided in this study is enough to come to this conclusion. What do the authors think, if stained species were considered only the results would show any difference or not?

AC: The referee poses an important question here. Unfortunately, there are not many studies in this region comparing live/dead assemblages so it is difficult to speculate on this point. Further, some studies (Bernhard et al 2006) have shown that Rose-Bengal staining does not accurately capture the live foraminiferal fauna. Other studies that have examined stained vs. unstained including Jorissen and Wittling 1999 document that some epifaunal and superficial infaunal species may reproduce opportunistically and thus have higher seasonal variability in comparison to infaunal species which they document as having stable densities through time. Thus, the assemblage we quantified may oversample epifaunal taxa relative to infaunal taxa in comparison to what may have been found if the samples were stained.

Line 301: what does "...size fraction or 2" mean? paragraph starting at line 312: for such a discussion based on specific species, authors should provide a reference list including species names as mentioned earlier.

AC: We will add a species list and reference list for the species that we discuss in detail. "Size fraction or 2)" was part of a larger list, we will remove the 1) and 2) for clarity.

line 341: this is the first time specific oxygen concentration and terminology is given. This should come earlier.

AC: This will appear earlier, starting in the introduction.

paragraph starting at line 365: discussion on Bolivinids: there are so many studies on bolivinids at similar setups, those should be mentioned and discussed here. Some examples are: Mallon et al., 2012; Cardich et al., (several papers), Glock et al., (several papers), Caulle et al., 2014; Jannink et al., 1998. the list goes on.

AC: We will add these references.

line 387: Please rephrase the last sentence.

AC: We will incorporate the suggested change.

paragraph starting with line 424: it would be nice to compare results with previous observations from the region.

AC: We will incorporate the suggested change.

comment on data availability: will it be open access upon publication?

AC: Yes, the data are already available and open access on NOAA Paleoclimate Database.

Figure 1: please give more information in the figure caption including which sites have what kind of results in the text. what are the depths of these sites?

AC: We will incorporate the suggested change. The depths are 300, 528, 700, 800, and 1200 m, this can be added to Figure 1.

Figure 2: water depth on y axis? station names could be implemented.

AC: Water depth is on the y axis in this plot. We will clarify this in the figure.

Figure 3 caption: "General observations .... " this is not needed here. Figure should be cited in the text more often.

AC: We will cite Figure 3 in the text to improve clarification. Referee #1 also suggested the same change. Further, we have updated Figure 3, this will add clarification to this point.

Figure 4: y axis please mention Rel. Abundance (%) instead.

AC: We will incorporate the suggested change.

Table 1: please add salinity and the captions should be more informative including where this information comes from. CTD? porewater? what is TOM?

AC: We will incorporate the suggested change. This data comes from a CTD (as answered above) of bottom water, not porewater. TOM is total organic matter, methods for this are listed in the methods section of the paper.

Table 2 caption: mixed planktonic foraminifera species. please remove bulk.

AC: We will incorporate the suggested change.

**Southern California margin benthic foraminiferal assemblages record recent centennial-scale changes in oxygen minimum zone**

Hannah M. Palmer[1,2], Tessa M. Hill[1,2], Peter D. Roopnarine[3], Sarah E. Myhre[4], Katherine R. Reyes[1,5], and Jonas T. Donnenfield[1,6]

[1]Bodega Marine Laboratory, University of California, Davis, California, USA
[2]Department of Earth and Planetary Sciences, University of California, Davis, California, USA
[3]Department of Invertebrate Zoology and Geology, California Academy of Sciences, San Francisco, California, USA
[4]School of Oceanography, University of Washington, Seattle, Washington, USA
[5]Department of Natural Sciences and Mathematics, Dominican University of California, San Rafael, California, USA
[6]Geology Department, Carleton College, Northfield, Minnesota, USA

*Correspondence to:* Hannah M. Palmer (hmpalmer@ucdavis.edu)

**Abstract.** Microfossil assemblages provide valuable records to investigate variability in continental margin biogeochemical cycles, including dynamics of the oxygen minimum zone (OMZ). Analyses of modern assemblages across environmental gradients are necessary to understand relationships between assemblage characteristics and environmental factors. Five cores were analyzed from the San Diego margin (32°42'00"N, 117°30'00"W, 300-1175 m water depth) for core top benthic foraminiferal assemblages to understand relationships between community assemblages and spatial hydrographic gradients and for down core benthic foraminiferal assemblages to identify changes in the oxygen minimum zone through time. Comparisons of benthic foraminiferal assemblages from two size fractions (63-150 and >150 μm) exhibit similar trends across the spatial/environmental gradient, or in some cases exhibit more pronounced spatial trends in the >150 μm fraction. A range of species diversity exists within the modern OMZ (1.910-2.586 H, Shannon Index), suggesting that diversity is not driven by oxygenation alone. We identify two hypoxic associated species (*B. spissa* and *U. peregrina*), one oxic associated species (*G. subglobosa*) and one OMZ edge-associated species (*B. argentea*). Down core analysis of indicator species reveal variability in the upper margin of the OMZ (528 m water depth) while the core of the OMZ (800 m) and below the OMZ (1175 m) remained stable in the last 1.5 ka. We document expansion of the upper margin of the OMZ beginning 400 ybp on the San Diego margin that is synchronous with other regional records of oxygenation.

**1 Introduction**

Ocean oxygenation is declining globally; rising ocean temperatures decrease oxygen solubility at the sea surface and increased stratification inhibits ventilation, leading to decreased oxygen at depth (Breitburg et al., 2018; Levin et al., 2009; Stramma et al., 2010). Expansions of oxygen minimum zones (OMZ) have already been documented and further expansions are predicted (Bograd et al., 2008; Schmidtko et al., 2017; Stramma et al., 2010). Within the California Current system, a decline in dissolved oxygen (DO) concentration, shoaling of the hypoxic boundary, and decreased pH have been documented (Bakun, 2017; Bograd et al., 2008). The intensity and geographic extent of the CA margin oxygen minimum zone has oscillated in response to past changes in climate and ocean temperatures on millennial timescales, weakening during cool periods and strengthening during warm periods (Cannariato and Kennett, 1999; Jaccard et al., 2014; Moffitt et al., 2014, 2015a; Ohkushi et al., 2013). Determination of timing and drivers of past expansions and contractions of oxygen minimum zones is critical to developing accurate predictions of future change (Jaccard et al., 2014).

Continental margin biogeochemical dynamics structure shelf ecosystems across space and time (Levin et al., 2009; Levin and Dayton, 2009). In particular, oxygenation is a key determinant of benthic zonation; seafloor ecosystems are subject to major turnover in response to relatively minor inferred changes in oxygenation (Levin, 2003; Levin and Dayton, 2009; Moffitt et al., 2015b). Areas of low oxygen availability typically contain low abundance and diversity of organisms (Levin, 2003; Levin and Dayton, 2009). However, several species of benthic foraminifera are adapted to survive in low-oxygen conditions and are thus present, and often abundant, in such environments (Bernhard and Gupta, 1999; Gooday et al., 2000; Kaiho, 1994, 1999; Keating-Bitonti and Payne, 2016).

**1.1 Benthic foraminifera record changes in coastal margin biogeochemistry**

Microfossil records from the Southern California Borderlands are a critical tool for understanding changes in productivity (Cannariato and Kennett, 1999; Emmer and Thunell, 2000; Stott et al., 2000), orbital and millennial scale climate changes (Hendy, 2010; Hendy and Kennett, 2000; Taylor et al., 2015), and climate change through the Holocene (Balmaki et al., 2019; Fisler and Hendy, 2008; Friddell et al., 2003; Roark et al., 2003). Benthic foraminiferal assemblages are widely used as a proxy for changes in oxygenation through time (Balestra et al., 2018; Bernhard et al., 1997; Bernhard and Gupta, 1999; Cannariato and Kennett, 1999; Gooday, 2003; Jorissen et al., 2007; Moffitt et al., 2014; Ohkushi et al., 2013; Shibahara et al., 2007; Tetard et al., 2017). Previous work (through analysis of benthic foraminifera along environmental depth gradients and in laboratory culturing studies) documented relationships between benthic foraminiferal taxa and water depth, oxygen concentration, sediment substrate, position in the sediment matrix, nitrate availability, and organic matter availability (Bernhard et al., 1997; Bernhard and Bowser, 1999; Bernhard and Gupta, 1999; Caulle et al., 2014; Douglas, 1981; Douglas and Heitman, 1979; Erdem et al., 2019; Jorissen et al., 2007; Kaiho, 1994, 1999; Mallon et al., 2012; Mazumder and Nigam, 2014; Mullins et al., 1985).

**Moved down [1]:** 1.1 Microfossil record of coastal margin biogeochemical change ¶

**Commented [h7]:** RC1: -Line 42: "declines in dissolved oxygen," awkward. Please rephrase.

**Moved (insertion) [3]**

**Commented [h8]:** RC1: -Line 43: determination of

**Commented [h9]:** RC1: -Line 49: "Low-oxygen zones typically contain both low abundance and low diversity of" change to "Areas of low oxygen availability typically contain low abundance and diversity of "

**Moved (insertion) [1]**

**Commented [h11]:** RC2: paragraph starting with line 67: this section could be improved significantly by including previous observations from other OMZs which should be included in discussion where Bolivinids and nitrate availability are discussed.
RC2: RC2: 2. Introduction and discussion should be improved in terms of using literature and previous work from different OMZs. For instance there are significant amount of work from the Peruvian and Arabian OMZs focusing on similar oxygen gradient and benthic foraminiferal assemblages.

**Commented [h12]:** RC2: page 2, line 60: please check Tetard et al., 2017.

**Commented [h13]:** Added Tetard et al 2017

**Commented [h14]:** Added Caulle et al 2014, Erdem et al 2019, Mallon et al 2012, Mazumder and Nigam 2014

**Moved (insertion) [2]**

[revised manuscript text omitted]

**Commented [h38]:** RC1: -Line 156: no units on salinity. Remove "with a total" replace "range of" with "ranging from"

AC: We chose to report salinity in "practical salinity units." We will make the additional language change.

**Commented [h39]:** Rc2: RC2: 4. Definition of an OMZ: please introduce OMZ already in introduction. This MS uses certain terms such as OMZ edge, suboxia, hypoxic boundary and so on; to eliminate the confusion, edge or boundary of an OMZ considered here should be introduced as early in the text as possible.

**Commented [h40]:** RC1: -Line 157: (DO) once you have introduced an abbreviation you do not have to reference it again and you can then use the shorthand in the text. I suggest you do a search of the manuscript and identify duplicates of instances such as this. I suggest a rewriting of this section for clarity. Either report based on water column depth or minimums. As is, it is confusing. –Rephrasing suggestion: —"Water column dissolved oxygen (DO) concentration documents a low oxygen zone, with a minimum occurring at 700 m water depth (0.26 ml/L; Figure 2, Table 1), compared to 1.54 ml/L at 300 m and 0.58 ml/L[6]

**Commented [h41]:** RC2: section 3.1.: this section is confusing. please be clear with what is presented here. I ... [7]

**Commented [h42]:** RC1: ." This needs to be identified in Figure 3.

**Commented [h43]:** RC1: -Line 173: Remove "The site" and begin the sentence with "At". Change "at the core of the ... [9]

**Commented [h44]:** RC1: 3. A better discussion of how agglutinated foraminifera are reported, or not, in the total ... [11]

**Commented [h45]:** RC1: -Lines 180-181: "These dominant taxa make up more than 84% of all foraminifera counted ... [12]

**Commented [h46]:** RC1: -Line 181: "All other species each account for <4% of total assemblage across all core tops ... [13]

**Commented [h47]:** RC1: -Line 179: "sp1" can this just be written as "sp."?

First, we report the benthic foraminiferal assemblage from the >63 μm size fraction; we then report on a comparison between the 63 - 150 μm and >150 μm size fractions. The assemblage at 300 m is dominated by *G. subglobosa* (28%), *B. argentea* (25%), *U. peregrina* (10%), *Epistominella* sp. (8%), and *Bolivina spissa* (6%); species richness is 24 and diversity (H) is 2.133. The assemblage at 528 m is dominated by *B. argentea* (37%), *U. peregrina* (23%), *Epistominella* sp. (15%), *C. carinata* (6%) and *G. subglobosa* (5%); species richness is 25 and diversity (H) is 1.910. The assemblage at 700 m is dominated by *Epistominella* sp. (29%), *C. carinata* (15%), *U. peregrina* (13%), *G. subglobosa* (11%), *B. argentea* (10%), and *B. spissa* (7%); species richness is 23 and diversity (H) is 2.249. The assemblage at 800 m is dominated by *B. spissa* (16%), *U. peregrina* (16%), *Epistominella* sp. (13%), *C. carinata* (10%), *Cibicidoides* sp. (10%), and *Globobulimina ovata* (13%); species richness is 25 and diversity (H) is 2.586. The assemblage at 1175 m is dominated by *C. carinata* (25%), *Epistominella* sp. (20%), *G. subglobosa* (8%), *B. spissa* (6%); species richness is 25 and diversity (H) is 2.389 (Figure 2).

**3.3 Comparison of benthic foraminifera in two size fractions**

Comparison of foraminiferal abundance between the 63-150 μm and >150 μm shows higher abundance in the small fraction at 300 m, 700 m and 1175 m, and higher abundance in the large size fraction at 528 m and 800 m. Several taxa are found in both size fractions at all five water depths: *Globocassidulina subglobosa* and *Epistominella* sp. Four species have significantly different relative abundances between size classes; three are more likely to be found in the 63-150 μm (*Cassidulina carinata*, *Epistominella* sp. and *G. subglobosa*) and one species (*U. peregrina* ) is significantly more likely to be found in the 150 μm size fraction (ANOVA, p<0.05 for all, Figure 3).

In the >150 μm size fraction, species diversity (H) ranges from 1.316 - 2.700; minimum diversity (H) is found at 528 m (1.316) and maximum diversity (H) is found at 1175 m (2.700). In comparison, in the 63-150 μm size fraction, species diversity (H) ranges from 1.710 - 2.042; minimum diversity (H) is found at 700 m (1.710) and maximum diversity (H) is found at 800 m (2.042) (Figure 2). Species diversity is greater in the >150 μm size fraction relative to the 63-150 μm size fraction at all sites except the site at 528 m (Figure 2).

When we consider the complete assemblage (>63μm) we can classify the most abundant species into four groups based on their trends relative to the oxygen minimum zone (Figure 3). Two species are more abundant within the OMZ: *B. spissa,* and *U. peregrina*; we identify these species as dysoxic indicator species. One species is less abundant within the OMZ relative to sites outside of the range of the OMZ: *G. subglobosa*; we identify this species as an oxic indicator species (Kaiho, 1999). Two species increase in abundance with water depth: *C. carinata* and *Epistominella* sp. One species is most abundant near the uppermost edge of the OMZ: *B. argentea*; this species may be edge-associated (Mullins et al., 1985). Importantly, when we consider only the >150 μm size fraction, we observe the same trends: high abundance in OMZ (*B. spissa, U. peregrina*), low abundance in OMZ (*G. subglobosa*), increased abundance with depth (*C. carinata* and *Epistominella* sp.), and OMZ edge-associated (*B. argentea*) (Figure 3). Generally, we find that trends across depth are similar between the complete (>63 μm) and large size fraction (>150 μm) or are more pronounced in the >150 μm size fraction compared to the 63-150 μm size fraction

**Commented [h48]:** RC1: -Line 200: "of" repeated. Could streamline to, "Comparisons of foraminifera abundances between. . ."

**Commented [h49]:** RC1: -Line 200: change 150 to >150 microns.

**Commented [h50]:** RC1: -Lines 202-204: "Three species have significantly different relative abundances between size classes; three are more likely to be found in the 63-150 μm (Cassidulina carinata, Epistominella sp1 and G. subglobosa)". When you repeat "three" are you referring to the same 3 species? If so, please rephrase.

**Commented [h51]:** RC1: -Line 207: I don't think it is necessary to repeat "Shannon Index" after you describe it in the methodology.

**Commented [h52]:** RC1: -Line 217: When you say above and below the OMZ do you mean on the seafloor? Not in the water column?

**Commented [h53]:** RC1: Please clarify. G. subglobosa as oxic indicator- This assertion needs a discussion or reference. Skipped to discussion

AC: When we refer to above and below the OMZ we are referring to within the water column, not above/below the sediment surface. We are indicating that G. subglobosa is more abundant at the sites above (300m) and below (1175m) the OMZ. As such, it is from our own data that we identify

**Commented [h54]:** RC2: line 222: the term edge domina

**Commented [h55]:** RC1: -Lines 200- Section 3.3: Based

(Figure 3). In some taxa, trends in both size fractions are similar across depth (*B. spissa, Epistominella* sp., *G. subglobosa, C. carinata*) (Figure 3). For other taxa, we observe a low relative abundance of a species in the (63-150 μm) fraction throughout all water depths, while for the same species in the >150 μm size fraction, we observe a pronounced trend through depth (*U. peregrina, B. argentea*) (Figure 3).

To further analyze these trends, we completed pairwise analysis of relative abundances of benthic foraminifera and environmental parameters. DO concentrations and pH are correlated at all water depths; here we chose to compare foraminiferal abundances to dissolved oxygen, yet we acknowledge that these affiliations may be driven by the combined effect of the oxygen minimum/carbon maximum zone. When we analyze the complete assemblage (>63 μm) we identify a significant positive correlation between *G. subglobosa* and dissolved oxygen ($r^2$=0.76, $p<0.05$) and temperature ($r^2$=0.64, $p<0.05$) and a significant negative correlation between *G. subglobosa* and total organic matter ($r^2$=-0.72, $p<0.05$). If we analyze the >150 μm size fraction only, we identify the same significant positive correlation between DO and *G. subglobosa* ($r^2$=0.96, $p<0.05$) and also identify a positive correlation between *C. carinata* and water depth; abundance of this species increases with depth ($r^2$=0.93, $p<0.05$). When we analyze the 63-150 μm size fraction alone, we identify the same trends as observed in the >150 μm fraction: a significant positive correlation between *G. subglobosa* and dissolved oxygen ($r^2$=0.90, $p<0.05$). and a significant positive correlation between *C. carinata* and water depth ($r^2$=0.88, $p<0.05$). In the 63-150 μm fraction we also identify a significant negative correlation between *U. peregrina* and water depth ($r^2$=-0.95, $p<0.05$). We do not identify significant correlations between any other taxa and environmental parameter.

**3.4 Multivariate analyses of benthic foraminiferal assemblage**

[revised manuscript text omitted]

**Commented [h57]:** RC1: -Line 262: "see Discussion section" can you refer to a specific section number e.g., 4.3?

**Commented [h58]:** RC1: -Line 263: "core top[s]"

**Commented [h59]:** RC1: -Line 270: What do you mean by equal in magnitude? Can you elaborate on that? Perhaps by giving total abundance or percent abundance examples?

-Lines 268-269: please rephrase this sentence for clarity. It is hard to understand your meaning.

**Commented [h60]:** RC2: paragraph starting from line 268: this could be written much simpler, I am not sure I understand the information provided here.

**Commented [h61]:** RC1: -Lines 273- 279: shorten this section to make your findings more impactful. The word [17]

**Commented [h62]:** RC2: Line 301: what does "...size [19]

**Commented [h63]:** RC1: -Lines 300-302: Where can the [20]

**Moved (insertion) [4]**

[revised manuscript text omitted]

**Commented [h70]:** RC1: -Lines 442 and 443: change "classes" to "fractions"

**Commented [h71]:** RC1: -Line 447: "variability in upper margin of the OMZ" should be "variability in [the] upper margin of the OMZ"

**Commented [h72]:** RC1: -Line 447: by "core" do you mean center? Is there another word you can use here so as not to confuse it with sediment cores?

**Commented [h73]:** RC1: -Lines 448-449: "We document expansion of upper margin of OMZ beginning 400 ybp on San Diego Margin that is synchronous with regional records of oxygenation." Should be re-written. Perhaps, "In this study, upper margin OMZ expansion beginning 400 ybp on San Diego Margin 
[revised manuscript text omitted]

[Figure]

Commented [h76]: RC2: Figure 1: please give more information in the figure caption including which sites have what kind of results in the text. what are the depths of these sites?

[Figure]

[Figure]

**Figure 2.** Profiles of temperature (a), dissolved oxygen (b), pH (c), and total organic matter (% by weight) (d) across depth transect. Foraminiferal abundance (total calcareous foraminifera) (e) and diversity (Shannon Index, H) (f) are shown for two size fractions. In panels (e) and (f), large black dots are >150 μm size fraction, small black dots are 63-150 μm and black line on diversity plot represents trends from the complete assemblage (>63 μm). Assemblage counts are standardized to the volume (63.62 cm³) of the sampled cylinder of the sediment (core). Gray dashed line shows approximate boundaries of the oxygen minimum zone.

925

**Commented [h77]:** RC1: -Based on the OMZ bounds of Figure 2 it seems the majority of the foraminiferal abundance plots resides in what you have defined at the OMZ. So how can you see increases if you have no "background" to compare it to? Please clarify.
AC: Updated Figure 2 to show defined OMZ.

RC2: Figure 2: water depth on y axis? station names could be implemented.

**Commented [h78]:** AC: Figure 2 Diversity panel updated to alleviate graphical error (same error that occurred in Figure 3)

**Commented [h79]:** RC1: -Figure 2: Extra ")";

[Figure]

**Figure 3:** Relative abundance of foraminiferal species (percent of total calcareous taxa) in core top sample (0-2 cm) vs. water depth (m). Large black dots are >150 μm size fraction, small black dots are 63-150 μm size fraction and black lines represent trends considering the complete assemblage (>63 μm).

**Commented [h80]:** RC1: ." This needs to be identified in Figure 3.

**Commented [h81]:** RC1: -Figure 3: "General observations discussed in the text are noted here, e.g., species that increase in abundance in the OMZ, appear associated with the "edge" of the OMZ, etc. Note difference in x-axis in B. argentea plot" is not very useful information for a figure caption. Please describe the structure of the graph and summarize what you observed or reference to the section ... [21]

**Commented [h82]:** RC2: Figure 3 caption: "General observations .... " this is not needed here. Figure should be cited in the text more often.

**Commented [h83]:**

[Figure]

[Figure]

**Figure 4:** Top 3 panels show relative abundance of two species of low-oxygen foraminifera (*B. spissa* and *U. peregrina*, gray lines) and one species of oxic foraminifera (*G. subglobosa*, blue lines) from the >150 μm size fraction down core through time, in years before present for cores from 3 water depths (528 m, 800 m, 1175 m). Top panel also includes relative abundance of *B. argentea*. Bottom panel shows diversity (Shannon's Index, H) through time for 3 cores.

**Commented [h84]:** RC1: -Figure 4: The Bolivina spissa and U. peregrina lines are very similar in color and hard to distinguish. Can the point shapes be changed to better facilitate reading?
AC: changed color of line

RC2: Figure 4: y axis please mention Rel. Abundance (%) instead.

950
955

960 **Tables**

| Core Name | Water Depth (m) | Latitude | Longitude | Core Length (cm) | Temperature (ºC) | Dissolved Oxygen (mL/L) | pH | Salinity (psu) | Total organic matter (% wt) |
|---|---|---|---|---|---|---|---|---|---|
| MV1217-5-2 | 300 | 32.8100166 | 117.4681 | 16 | 8.6137 | 1.54 | 7.65 | 34.145 | 6.8 |
| MV1217-2-3 | 528 | 32.8100333 | 117.416583 | 26 | 6.6217 | 0.35 | 7.57 | 34.313 | 11.7 |
| MV1217-3-3 | 700 | 32.8099666 | 117.450966 | 20 | 5.8975 | 0.26 | 7.56 | 34.348 | 13.9 |
| MV1217-1-3 | 800 | 32.8095166 | 117.506933 | 20 | 5.0491 | 0.29 | 7.56 | 34.405 | 14.2 |
| MV1217-4-3/1 | 1175 | 32.6333333 | 117.499883 | 16 | 3.8231 | 0.58 | 7.59 | 34.501 | 14.7 |

**Table 1:** Data for cores used in this study. Temperature, dissolved oxygen, pH and salinity were measured in bottom water directly above each coring site.

| Core | Sample Interval | Age ($^{14}$C years) | ± | 1 Sigma Maximum Calendar Age Range | 1 Sigma Minimum Calendar Age Range | Age in Calendar Years | Sedimentation Rate (cm/ka) | ± |
|---|---|---|---|---|---|---|---|---|
| MV1217-2-3 | 11-12 cm | 1230 | 30 | 1403 | 1319 | 1361 | 16.9 | 1.0 |
| MV1217-2-3 | 16-17 cm | 2085 | 30 | 602 | 474 | 538 | 6.1 | 0.2 |
| MV1217-2-3 | 25-26 cm | 2405 | 35 | 237 | 107 | 172 | 24.6 | 0.1 |
| MV1217-4-3 | 5-6 cm | 670 | 35 | 1950 | 1837 | 1893.5 | 42.2 | 11.3 |
| MV1217-4-1 | 10-11 cm | 960 | 30 | 1630 | 1518 | 1574 | 15.6 | 0.2 |
| MV1217-4-1 | 20-21 cm | 1840 | 35 | 817 | 698 | 757.5 | 12.3 | 0.4 |

965 **Table 2.** Radiocarbon ages of mixed planktonic foraminifera from MV1217-2-3 (528 m water depth), MV1217-4-3 (1175 m water depth) and MV1217-4-1 (1175 m water depth).

**Commented [h87]:** RC2: Table 1: please add salinity and the captions should be more informative including where this information comes from. CTD? porewater? what is TOM?

**Commented [h85]:** RC1: -Table 1: Include salinity in the table

**Commented [h86R85]:** RC2: and salinity should be included as well in the table.

**Formatted Table**

**Commented [h88]:** RC1: 14 in the 14C needs to be superscripted

**Commented [h89]:** RC2: Table 2 caption: mixed planktonic foraminifera species. please remove bulk

| Page 2: [1] Deleted | hmpalmer4@gmail.com | 2/25/20 10:25:00 AM |
|---|---|---|

| Page 3: [2] Commented [h17] | hmpalmer4@gmail.com | 2/25/20 10:36:00 AM |
|---|---|---|

RC1: RC1: 4. A more complete literature review is needed in the discussion section in order to support the findings of the authors

RC2: paragraph starting with line 67: this section could be improved significantly by including previous observations from other OMZs which should be included in discussion where Bolivinids and nitrate availability are discussed.

RC2: 2. Introduction and discussion should be improved in terms of using literature and previous work from different OMZs. For instance there are significant amount of work from the Peruvian and Arabian OMZs focusing on similar oxygen gradient and benthic foraminiferal assemblages. These studies should be included in terms of benthic foraminifera habitat in relation with oxygen and nitrate availability etc. This will improve the MS significantly. It is a pity that the species are not stained limiting the comparison with previous studies, and yet I believe the information presented here is really valuable.

| Page 3: [3] Deleted | hmpalmer4@gmail.com | 2/25/20 9:13:00 AM |
|---|---|---|

| Page 5: [4] Deleted | hmpalmer4@gmail.com | 2/24/20 10:04:00 AM |
|---|---|---|

| Page 5: [5] Commented [h36] | hmpalmer4@gmail.com | 2/24/20 12:35:00 PM |
|---|---|---|

RC2: Line 205: is ANOVA introduced already in methods?

| Page 6: [6] Commented [h40] | hmpalmer4@gmail.com | 2/24/20 11:24:00 AM |
|---|---|---|

RC1: -Line 157: (DO) once you have introduced an abbreviation you do not have to reference it again and you can then use the shorthand in the text. I suggest you do a search of the manuscript and identify duplicates of instances such as this. I suggest a rewriting of this section for clarity. Either report based on water column depth or minimums. As is, it is confusing. –Rephrasing suggestion: —"Water column dissolved oxygen (DO) concentration documents a low oxygen zone, with a minimum occurring at 700 m water depth (0.26 ml/L; Figure 2, Table 1), compared to 1.54 ml/L at 300 m and 0.58 ml/L at 1175 m. Minimum pH is documented at 700 m (7.55) and is higher above (300 m, 7.65) and below (1175 m, 7.59) the intermediate depth low pH zone (Figure 2, Table 1)." —Water column DO measurements indicate areas of low oxygen availability from 300 m (1.54 ml/L) to 1175 m (0.58 ml/L) with lowest oxygen availability at 700 m (0.26 ml/L). Although not greatly variable, pH minima also occur at 700 m (7.55) and is higher at 300 m (7.65) and 1175 m, (7.59). In this section, why not report as hypoxic, anoxic, as outlined by Bernhard et al?

AC: We will improve upon the introduction of abbreviations earlier on in the text as suggested by both reviewers. We chose not to report oxygen as hypoxic, anoxic, as outlined by Bernhard et al because we later argue that in the environment we studied, these categories of foraminifera by oxygenation do not accurately reflect the foraminifera we sampled. Thus, it is more useful to describe each site using the measured dissolved oxygen.

| **Page 6: [7] Commented [h41]** | **hmpalmer4@gmail.com** | **2/24/20 12:29:00 PM** |

RC2: section 3.1.: this section is confusing. please be clear with what is presented here. I assume these are the deepest points CTD measured? is there any pore water measurements or are these only water column?

| **Page 6: [8] Deleted** | **hmpalmer4@gmail.com** | **2/24/20 11:24:00 AM** |

| **Page 6: [9] Commented [h43]** | **hmpalmer4@gmail.com** | **2/24/20 10:08:00 AM** |

RC1: -Line 173: Remove "The site" and begin the sentence with "At". Change "at the core of the OMZ" to "within the core of the OMZ, the". Add "occurred" to the end of the sentence. Alternatively, this sentence could be shortened and combined with the previous sentence: Calcareous taxa dominated the assemblage at every site; agglutinated foraminifera made up 0 (e.g., XXX m) to 17.7% of the assemblage at 700 m, within the core of the OMZ.

| **Page 6: [10] Deleted** | **hmpalmer4@gmail.com** | **2/24/20 10:07:00 AM** |

| **Page 6: [11] Commented [h44]** | **hmpalmer4@gmail.com** | **2/24/20 9:24:00 AM** |

RC1: 3. A better discussion of how agglutinated foraminifera are reported, or not, in the total foraminiferal counts should be included.

| **Page 6: [12] Commented [h45]** | **hmpalmer4@gmail.com** | **2/24/20 11:25:00 AM** |

RC1: -Lines 180-181: "These dominant taxa make up more than 84% of all foraminifera counted across all core top samples." Does this include the agglutinates or just 84% of the calcareous foraminiferal population?

| **Page 6: [13] Commented [h46]** | **hmpalmer4@gmail.com** | **2/24/20 10:14:00 AM** |

RC1: -Line 181: "All other species each account for <4% of total assemblage across all core tops." This sentence is confusing. Please rephrase.

| **Page 7: [14] Commented [h53]** | **hmpalmer4@gmail.com** | **2/24/20 11:29:00 AM** |

RC1: Please clarify. G. subglobosa as oxic indicator- This assertion needs a discussion or reference. Skipped to discussion

AC: When we refer to above and below the OMZ we are referring to within the water column, not above/below the sediment surface. We are indicating that *G. subglobosa* is more abundant at the sites above (300m) and below (1175m) the OMZ. As such, it is from our own data that we identify this species as an oxic indicator, rather than from previous work. But, additionally, we can add citations showing this species as an oxic indicator (Kaiho 1999).

| **Page 7: [15] Commented [h54]** | **hmpalmer4@gmail.com** | **2/25/20 10:43:00 AM** |

RC2: line 222: the term edge dominant.. what does this actually mean? According to which previous work edge of the OMZ is considered?

| **Page 7: [16] Commented [h55]** | **hmpalmer4@gmail.com** | **2/24/20 11:27:00 AM** |

RC1: -Lines 200- Section 3.3: Based on the results of this section why are you concluding that it is better to look at the >150micron size fraction as you state in the conclusion?

AC: We conclude that it is effective to look at the > 150 micron size fraction because we find that trends across depth are similar between the complete (>63 µm) and large size fraction (>150 µm) or are more pronounced in the >150 µm size fraction compared to the 63-150 µm size fraction. Further, we utilized this size fraction to remain consistent with other regional studies. These trends are further elucidated by the corrected Figure 3 we have uploaded as part of this response.

| Page 9: [17] Commented [h61] | hmpalmer4@gmail.com | 2/25/20 11:41:00 AM |
|---|---|---|

RC1: -Lines 273- 279: shorten this section to make your findings more impactful. The word "document" is repeated and could be eliminated completely.

| Page 9: [18] Deleted | hmpalmer4@gmail.com | 2/25/20 11:41:00 AM |
|---|---|---|

| Page 9: [19] Commented [h62] | hmpalmer4@gmail.com | 2/25/20 10:47:00 AM |
|---|---|---|

RC2: Line 301: what does "...size fraction or 2" mean? paragraph starting at line 312: for such a discussion based on specific species, authors should provide a reference list including species names as mentioned earlier.

| Page 9: [20] Commented [h63] | hmpalmer4@gmail.com | 2/24/20 11:46:00 AM |
|---|---|---|

RC1: -Lines 300-302: Where can the readers see this stated relationship? "...and the >150 µm size fraction or 2) trends in the >150 µm size fraction are more pronounced than in the complete assemblage"

AC: This relationship can be seen in the updated Figure 3; due to the graphical error in Figure 3, this was not clear in the original submission.

| Page 23: [21] Commented [h81] | hmpalmer4@gmail.com | 2/24/20 11:50:00 AM |
|---|---|---|

RC1: -Figure 3: "General observations discussed in the text are noted here, e.g., species that increase in abundance in the OMZ, appear associated with the "edge" of the OMZ, etc. Note difference in x-axis in B. argentea plot" is not very useful information for a figure caption. Please describe the structure of the graph and summarize what you observed or reference to the section of the paper where it is discussed. Please also clearly state in this figure that "core tops" are the 0-2cm intervals.